# SMART: Scalable Mesh-free Aerodynamic Simulations from Raw Geometries using a Transformer-based Surrogate Model

**Jan Hagnberger** [1]  **Mathias Niepert** [1 2 3]

## Abstract

Machine learning–based surrogate models have emerged as more efficient alternatives to numerical solvers for physical simulations over complex geometries, such as car bodies. Many existing models incorporate the simulation mesh as an additional input, thereby reducing prediction errors. However, generating a simulation mesh for new geometries is computationally costly. In contrast, mesh-free methods, which do not rely on the simulation mesh, typically incur higher errors. Motivated by these considerations, we introduce SMART, a neural surrogate model that predicts physical quantities at arbitrary query locations using only a point-cloud representation of the geometry, without requiring access to the simulation mesh. The geometry and simulation parameters are encoded into a shared latent space that captures both structural and parametric characteristics of the physical field. A physics decoder then attends to the encoder's intermediate latent representations to map spatial queries to physical quantities. Through this cross-layer interaction, the model jointly updates latent geometric features and the evolving physical field. Extensive experiments show that SMART is competitive with and often outperforms existing methods that rely on the simulation mesh as input, demonstrating its capabilities for industry-level simulations.

## 1. Introduction

Physical simulations are essential for evaluating and optimizing technical properties during product development. For example, the design and optimization of car and air-

[1]Machine Learning and Simulation Lab, University of Stuttgart, Germany [2]International Max Planck Research School for Intelligent Systems [3]Stuttgart Center for Simulation Science. Correspondence to: Jan Hagnberger <j.hagnberger@gmail.com>.

*Proceedings of the $43^{rd}$ International Conference on Machine Learning*, Seoul, South Korea. PMLR 306, 2026. Copyright 2026 by the author(s).

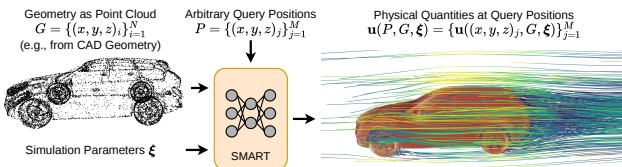

*Figure 1.* The proposed SMART model maps a geometry G, represented as a point cloud, and simulation parameters $\boldsymbol{\xi}$ to physical quantities (e.g., surface pressure and velocity in the volume) for arbitrary query positions $P$ without relying on the simulation mesh.

craft bodies rely on Computational Fluid Dynamics (CFD) simulations to assess aerodynamic properties such as drag and lift coefficients. Traditional design pipelines depend on numerical solvers for these simulations, which are computationally expensive and time-consuming, significantly slowing the iterative design process.

Machine Learning (ML) is increasingly used to accelerate the solution process of Partial Differential Equations (PDEs) and physical simulations (Lu et al., 2019; Li et al., 2021; 2023c; Serrano et al., 2024; Hagnberger et al., 2026). ML-based methods offer several advantages over traditional numerical solvers, including efficient inference and applicability even when the underlying physics is partially unknown (Tompson et al., 2017; Li et al., 2021). Recent advances have scaled ML-based models to industry-level settings, enabling simulations over complex and varying geometries of cars and airplanes with tens of thousands of mesh points (Li et al., 2023b; Wu et al., 2024; Ranade et al., 2025; Alkin et al., 2025a). These models predict physical fields such as surface pressure and velocity in the surrounding volume from geometric inputs. Compared to numerical solvers, they can achieve orders-of-magnitude faster inference while maintaining acceptable accuracy, thereby simplifying and accelerating the design processes.

Recall that simulations involve two different meshes: the cheap geometry mesh and an expensive simulation mesh. State-of-the-art architectures, such as Transolver (Wu et al., 2024) and the mesh-dependent AB-UPT (Alkin et al., 2025a), directly use the simulation mesh of each geometry as input. Conditioning on the simulation mesh substantially reduces prediction error, as the mesh provides fine-grained spatial resolution in important regions. However, this accu-

racy comes at a practical cost: before inference on a new geometry, a simulation mesh must be generated, a process that can take up to 30 minutes (Ashton et al., 2024) and, in the case of adaptive meshing, may even require running a full simulation. This mesh-generation step significantly slows iterative design workflows. Importantly, these models use the simulation mesh not merely as a set of independent query locations for evaluating the solution, but as an input feature that actively shapes the learned representation. In contrast, approaches that treat the simulation mesh solely as query points typically suffer from substantially higher error.

Motivated by this trade-off, we propose SMART (**S**calable **M**esh-free **A**erodynamic simulations from **R**aw geometries using a **T**ransformer-based surrogate model), a model that predicts physical quantities for varying and complex geometries from a point-cloud representation of the geometry, without relying on the simulation mesh. The SMART model takes a geometry represented as a point cloud and arbitrary query positions as input and maps them to the physical quantities at those positions, as depicted in the overview in Figure 1. SMART compresses the input geometry and additional parameters (e.g., yaw angle) into a shared latent space. This reduces dimensionality, cleanly separates the geometric representation from the output query locations, and allows the model to learn the structure of the physical field in this compressed latent space. A physics decoder then attends to the encoder's intermediate latent representations to map arbitrary query points to physical quantities. Through this cross-layer interaction, the model tightly couples the latent geometric representation with the evolving physical field.

We evaluate SMART on multiple large-scale aerodynamic simulation problems from automotive and aerospace engineering. Across these tasks, SMART matches or outperforms strong mesh-dependent baselines, while eliminating the need for simulation meshes during inference. Our contributions are summarized as follows:

1. We introduce a mesh-free neural surrogate model for large-scale aerodynamic simulations that supports arbitrary, independent spatial queries and removes the dependency on simulation-specific meshes.

2. We propose a geometry encoder that compresses point-cloud representations of the geometry and the simulation parameters into a compact latent representation, referred to as a latent geometry, which captures simulation-relevant geometric structure.

3. We design a cross-attention decoder that uses the encoder's intermediate latent geometries to map spatial queries to physical quantities. Through this cross-layer interaction, the model jointly updates the latent geometry and the evolving physics field, tightly coupling geometric context with physical predictions.

## 2. Simulations directly from Raw Geometries

In this section, we define the problem of aerodynamic simulations and state the desiderata of a neural surrogate.

### 2.1. Problem Definition

We consider the problem of learning a surrogate model for 3D aerodynamic simulations. The input is a 3D geometric representation of an object, given as a point cloud $G = \{(x, y, z)_i\}_{i=1}^{N} \in \mathcal{G}$ with $N$ points. The goal is to predict physical quantities, such as surface pressure and velocity field, throughout the 3D spatial domain $\Omega \subset \mathbb{R}^3$. The physical quantities are governed by underlying PDEs, such as the Navier-Stokes equations, with the geometry defining boundary conditions that influence the solution. Simulation parameters $\boldsymbol{\xi} \in \mathbb{R}^{N_p}$ further influence the solution. In practice, time-resolved simulation data is often averaged, since engineering applications typically require time-averaged solutions. Thus, the problem can be defined as learning the time-independent solution function $\mathbf{u} : \Omega \times \mathcal{G} \times \mathbb{R}^{N_p} \rightarrow \mathbb{R}^{N_c} : (x, y, z), G, \boldsymbol{\xi} \mapsto \mathbf{u}\big((x, y, z), G, \boldsymbol{\xi}\big)$ that maps a geometry $G$ from all possible geometries $\mathcal{G}$ and arbitrary queries $(x, y, z)$ from the spatial domain $\Omega$ to $N_c$ physical quantities for that position. Usually, we are interested in the physical quantities for multiple query points, which we denote as the query point set $P = \{(x, y, z)_j\}_{j=1}^{M}$ containing $M$ query points. Traditional numerical solvers first discretize the spatial domain into a simulation mesh, which is then used to solve the PDE. More advanced solvers employ adaptive meshing, refining the mesh dynamically in regions of interest to improve accuracy. Our goal is to approximate the solution function $\mathbf{u}\big((x, y, z), G, \boldsymbol{\xi}\big) \approx f_\theta\big((x, y, z), G, \boldsymbol{\xi}\big)$ with a neural network $f$ with parameters $\theta$, given training data from a numerical solver. The goal is for the neural surrogate to generalize across geometries and parameters, thereby serving as an efficient alternative to conventional solvers. Section A contains details on the problem definition.

### 2.2. Desiderata for a Neural Surrogate Model

We outline a set of desiderata for a neural surrogate model, which guide the development of the SMART model. Ideally, a neural surrogate should take as input only the geometry and output physical quantities at arbitrary and independent query points in the spatial domain.

**(D1) Efficiency and Accuracy.** A neural surrogate must offer substantially lower computational cost than traditional solvers, while maintaining acceptable prediction errors.

**(D2) Independence of Arbitrary Queries.** Neural surrogates should accept arbitrary query coordinates and treat each query independently, meaning the prediction at one

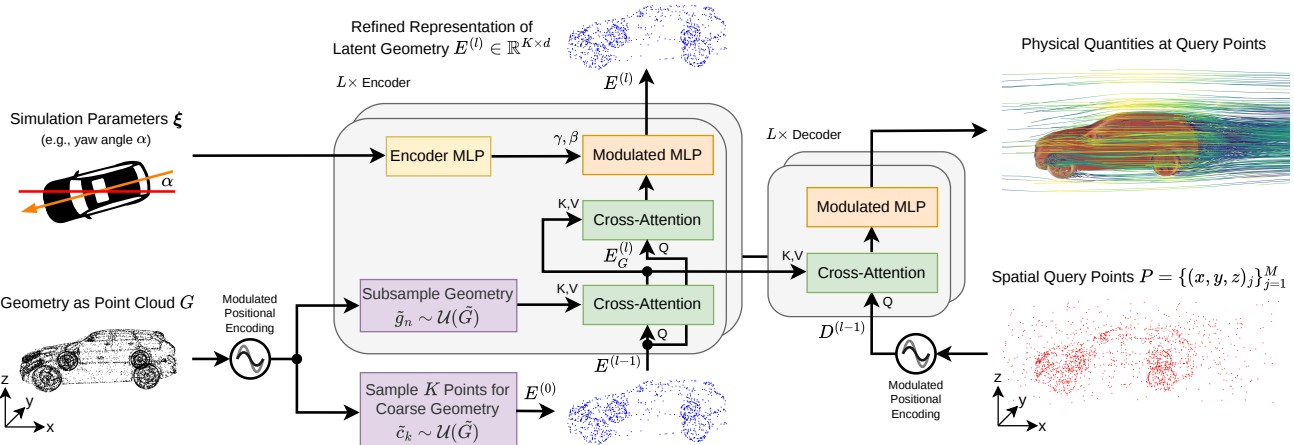

*Figure 2.* SMART is an encoder-decoder architecture that encodes the geometry $G$ and simulation parameters $\boldsymbol{\xi}$ into a shared latent space. The decoder attends to the encoder's intermediate latent geometries $E_G^{(l)}$ to map arbitrary spatial coordinates $P$ to physical quantities $\mathbf{u}(P, G, \boldsymbol{\xi})$. This cross-layer interaction enables a joint update of the latent geometry and the evolving physics field.

location must not depend on other points in the query set $P$. This reflects the core principle that querying a physical field should not alter the underlying solution. Although mechanisms like self-attention over queries can exploit spatial correlations and reduce error (Hao et al., 2023; Hagnberger et al., 2024), they break this independence. Practically, query independence enables dividing the domain into sub-regions and evaluating them sequentially to reduce GPU memory usage during inference (Alkin et al., 2025a).

**(D3) No Dependency on Simulation Mesh.** The neural surrogate should not rely on the simulation mesh as an input feature. Although meshes provide a useful prior, they are not available for new geometries at inference time and must be generated beforehand, which is computationally expensive. For example, Ashton et al. (2024) report that meshing a single sample of the AhmedML dataset requires approximately 30 minutes. The limitation is even more severe for simulators with adaptive-meshing, such as the Luminary adaptive mesh refinement used for SHIFT-Wing (Luminary Cloud, 2025b), where the mesh evolves during the simulation itself. In such cases, using the mesh as input would require running the full simulation in advance, defeating the purpose of a surrogate model. Please note that models such as the mesh-dependent AB-UPT (Alkin et al., 2025a) satisfy query independence and still rely on the simulation mesh.

## 3. Method

In this section, we present SMART, a neural surrogate model for 3D aerodynamic simulations designed to meet the criteria outlined in the previous section. The model predicts physical quantities solely from the geometry and supports arbitrary spatial queries, treating each query independently

so that none of them alters the model's internal representation. As a result, no simulation mesh generation is required when evaluating new geometries at inference time.

### 3.1. SMART: Mesh-free Aerodynamic Simulations

The model compresses the geometry, represented as a 3D point cloud $G = \{(x, y, z)_i\}_{i=1}^N$, and additional simulation parameters $\boldsymbol{\xi}$ into a sequence of latent representations, capturing both geometric and simulation-specific features. We also refer to the latent representations as latent geometries. Rather than producing a single global latent representation, the encoder outputs progressively refined latent geometries across $L$ blocks. A decoder mirrors the structure of the encoder with $L$ blocks and maps arbitrary queries, organized in a set of query points $P = \{(x, y, z)_j\}_{j=1}^M$, to physical quantities using the latent geometries. Crucially, each decoder block attends to the intermediate latent geometries produced by its corresponding encoder block, enabling a coupled update of both the latent geometry and the evolving physical field. This cross-layer interaction is central to SMART's simulation capability and similar to the anchor attention proposed by Alkin et al. (2025a). Figure 2 illustrates the encoder-decoder architecture of the model, given as

$$f_\theta(P, G, \boldsymbol{\xi}) = \mathcal{D}_\theta\big(\tilde{P}, \underbrace{\mathcal{E}_\theta(\tilde{G}, \boldsymbol{\xi})}_{\text{Latent Geometries}}, \boldsymbol{\xi}\big) \qquad (1)$$

where $\tilde{\cdot}$ represents embedded coordinates, $\mathcal{E}$ denotes the geometry encoder, and $\mathcal{D}$ the physics decoder.

#### 3.1.1. MODULATED POSITIONAL ENCODING

The coordinates of the geometry point cloud $G$ and the query coordinates $P$ are embedded into a $d$-dimensional feature space using sinusoidal functions with varying frequencies

(Vaswani et al., 2017). While multi-frequency embeddings provide multi-scale capabilities, smooth regions in aerodynamic simulations (e.g., air inlet) exhibit only low-frequency variation. To adaptively emphasize relevant frequency bands for each coordinate, we modulate the sinusoidal functions via a learned scaling and shifting, which is given as $\texttt{MPE}(x, 2i) = \sin\left(\left(\gamma_{2i}(x) \cdot x/10000^{2i/d}\right) + \beta_{2i}(x)\right)$ where $\gamma_{2i}(x)$ and $\beta_{2i}(x)$ denote the learned scaling and shifting for the $2i^{th}$ dimension that are computed using an MLP. An analogous formulation is applied to the cosine terms. Similar to prior work (Parmar et al., 2018; Carion et al., 2020; Alkin et al., 2025a), the 3D positional encoding is constructed by embedding each dimension independently and concatenating the resulting vectors into a single representation, denoted as $\texttt{MPE}(x, y, z) = \left(\texttt{MPE}(x)||\texttt{MPE}(y)||\texttt{MPE}(z)\right)$ where $||$ represents the concatenation. We denote the embedded geometry point cloud as $\tilde{G} = \{\texttt{MPE}((x, y, z)_i)\}_{i=1}^{N} \in \mathbb{R}^{N \times d}$ and the embedded query points as $\tilde{P} \in \mathbb{R}^{M \times d}$. Figure 2 shows the modulated positional encoding applied to the geometry point cloud $G$ and query position set $P$.

### 3.1.2. GEOMETRY ENCODER

The dense embedded geometry $\tilde{G} \in \mathbb{R}^{N \times d}$, with a variable size $N$ depending on the resolution, is mapped to a sequence of coarser latent geometries $E_G^{(l)} \in \mathbb{R}^{K \times d}$ of a fixed size $K$, typically with $N \gg K$. This dimensionality reduction decouples the geometry representation from the number of spatial queries, enabling independent scaling of geometry resolution and number of queries. Additionally, simulation parameters $\boldsymbol{\xi}$ are injected into the latent representation, enriching it with simulation-specific features. Formally, the encoder maps a geometry point cloud $\tilde{G}$ and simulation parameters $\boldsymbol{\xi}$ to a sequence of latent representations

$$\mathcal{E}(\tilde{G}, \boldsymbol{\xi}) = \left(E_G^{(1)}, \ldots, E_G^{(L)}\right); \quad E_G^{(l)} \in \mathbb{R}^{K \times d} \quad (2)$$

where each $E_G^{(l)}$ corresponds to the output of the geometry cross-attention layer of the $l^{th}$ encoder block.

**Sampling Initial Coarse Geometry Point Cloud.** As a first step, $K$ points are uniformly sampled from the input geometry to construct a coarse geometry $\tilde{C} = \{\tilde{c}_k\}_{k=1}^{K}$ with $\tilde{c}_k \sim \mathcal{U}(\tilde{G})$. This coarse set provides an initial geometric scaffold for the latent representations, serving as the input to the first encoder block, thus $E^{(0)} := \tilde{C} \in \mathbb{R}^{K \times d}$. Using a coarse point cloud has the advantage that it directly provides geometric information, prior to the application of the geometry encoder. Instead of random sampling, more advanced sampling strategies (e.g., curvature-aware or farthest-point sampling) could be used, but are not required for SMART.

**Geometry Cross-Attention.** Each encoder block refines the latent geometry with cross-attention by attending to a subsampled set of points from the full geometry $\tilde{G}$. For the $l^{th}$ geometry encoder block, the block first uniformly samples $N'$ points from the geometry point cloud

$$\tilde{G}_{sub} = \{\tilde{g}_n\}_{n=1}^{N'} \quad \tilde{g}_n \sim \mathcal{U}(\tilde{G}) \quad (3)$$

and computes cross-attention between the latent geometry $E^{(l-1)}$ and the subsampled geometry $\tilde{G}_{sub}$, given as

$$E_G^{(l)} = \texttt{CrossAttn}(Q = E^{(l-1)}, KV = \tilde{G}_{sub}) \quad (4)$$

where $\texttt{CrossAttn}$ computes multi-head cross-attention (Vaswani et al., 2017) between the query $Q$ and keys and values $KV$. Resampling $\tilde{G}_{sub}$ at each block allows covering large geometries while keeping attention costs manageable. Similar to constructing the initial coarse geometry, more sophisticated methods for sampling could be used.

**Refinement of Latent Geometry.** The cross-attended latent geometry $E_G^{(l)}$ is incorporated into $E^{(l-1)}$ with another cross-attention layer, denoted as $E_{CA}^{(l)} = \texttt{CrossAttn}(Q = E^{(l-1)}, KV = E_G^{(l)})$, which captures spatial dependencies within the coarse latent geometry.

**Simulation-Parameter Modulated MLP.** Similar to a Transformer, a pointwise MLP is applied to update the representation of each latent token. To incorporate additionally parametric information $\boldsymbol{\xi}$ (e.g., PDE parameters, yaw angle, or angle of attack), we modulate the intermediate activations of the MLP by scaling and shifting them as proposed in Perez et al. (2018), which is given as

$$E^{(l)} = \mathbf{W_2}\left(\sigma(\mathbf{W_1} E_{CA}^{(l)} + \mathbf{b_1})\gamma(\boldsymbol{\xi}) + \beta(\boldsymbol{\xi})\right) + \mathbf{b_2} \\ \gamma(\boldsymbol{\xi}), \beta(\boldsymbol{\xi}) = \texttt{MLP}(\boldsymbol{\xi}) \quad (5)$$

where $\mathbf{W}.$ and $\mathbf{b}.$ represent the weights and biases, and $\gamma(\boldsymbol{\xi})$ and $\beta(\boldsymbol{\xi})$ the shift and scaling computed by a shallow encoder MLP. This step injects simulation-specific information into the latent geometry.

**Encoder Outputs.** Each encoder block outputs (i) an intermediate representation $E_G^{(l)} \; \forall l = 1, \ldots, L$ after the geometry cross-attention layer, which is used by the decoder and (ii) a final representation $E^{(l)} \; \forall l = 1, \ldots, L$ after the modulated MLP, which is passed to the next encoder block. Both representations contain information about the geometry and the simulation parameters. Across $L$ blocks, the encoder progressively integrates geometric structure and simulation parameters into a compact latent geometry suitable for downstream physics field decoding.

### 3.1.3. PHYSICS DECODER

Given the latent geometries of each encoder block $E_G^{(1)}, \ldots, E_G^{(L)}$ and the embedded query coordinates $\tilde{P} \in$

$\mathbb{R}^{M \times d}$, the decoder maps each query to the corresponding physical quantities at these locations. To satisfy the requirement of query independence (desideratum 2), the decoder must not allow information exchange between queries, nor may queries modify the latent geometries. Consequently, self-attention among queries is prohibited, while pointwise transformations and cross-attention from the latent geometries to the queries remain permissible. Formally, the decoder implements a mapping

$$\mathcal{D}_\theta \big( \tilde{P}, \underbrace{(E_G^{(1)}, \ldots, E_G^{(L)})}_{=\mathcal{E}_\theta(\tilde{G}, \boldsymbol{\xi})}, \boldsymbol{\xi} \big) \approx \{\mathbf{u}(p_j, G, \boldsymbol{\xi})\}_{j=1}^M \quad (6)$$

where $\{\mathbf{u}(p_j, G, \boldsymbol{\xi})\}_{j=1}^M = \mathbf{u}(P, G, \boldsymbol{\xi})$ denotes the true physical field for all query coordinates $p_j$.

**Latent Geometry Cross-Attention.**  Each decoder block updates the query embeddings $D^{(l-1)} \in \mathbb{R}^{M \times d}$ by attending to the latent geometry $E_G^{(l)}$ produced at the corresponding encoder block $l$. The first decoder block receives the embedded query positions $D^{(0)} := \tilde{P} \in \mathbb{R}^{M \times d}$ as input. At block $l$, this update is implemented through cross-attention

$$D_{CA}^{(l)} = \texttt{CrossAttn}(Q = D^{(l-1)}, KV = E_G^{(l)}) \quad (7)$$

where the keys and values correspond to the intermediate latent geometry of the $l^{th}$ encoder block. This mechanism enables each query to extract geometric and simulation-dependent information from the latent geometry without interacting with other queries. Following cross-attention, a simulation-parameter modulated MLP is applied to incorporate the simulation parameters and update each query embedding independently. The cross-attention layer and MLP can share their weights with the cross-attention layer and MLP from the corresponding encoder block.

**Decoder Output.**  Each decoder block produces an updated query representation $D^{(l)}$ $\forall l = 1, \ldots, L$, which is used as input for the next decoder block. After $L$ blocks, the query embeddings are evolved to latent physical quantities, yielding the final representation $D^{(L)} \in \mathbb{R}^{M \times d}$. A final MLP maps the latent physics to the predicted physical quantities $\mathbf{u}(P, G, \boldsymbol{\xi}) \in \mathbb{R}^{M \times N_c}$ for all query locations.

### 3.1.4. CROSS-LAYER GEOMETRY-PHYSICS UPDATE

A central component of SMART is the cross-layer interaction between the encoder and decoder. Instead of attending only to the final encoder output $E^{(L)}$, each decoder block uses the intermediate latent geometry $E_G^{(l)}$ of the corresponding encoder block $l$ as key and values. This cross-layer interaction allows the encoder and decoder to jointly update the latent geometry and the latent physics, mirroring the anchor attention in AB-UPT (Alkin et al., 2025a). This design allows for (i) joint refinement of the geometry and physics and (ii) multi-scale geometric grounding.

**Joint Refinement of Geometry and Physics.**  The latent geometry evolves across encoder layers, while the decoder's physics-carrying query embeddings evolve across decoder layers. Cross-attention at every depth couples these two processes, enabling a joint update of the latent geometry and the emerging physics field.

**Multi-scale Geometric Grounding.**  Early encoder layers may capture coarse geometric structure, while deeper layers encode increasingly fine-grained details (cf. hierarchical feature learning). By attending to all intermediate representations, the decoder integrates geometric information across scales, improving physical fidelity.

This cross-layer geometry-physics update mechanism is crucial for SMART's simulation capabilities. Physics is not solely encoded in the encoder nor decoder, but emerges through iterative encoder-decoder interaction across all layers.

### 3.2. Training and Inference Procedure

Industrial-scale CFD simulations typically contain tens of millions of surface and volume points. Training neural surrogate models on all available mesh points simultaneously would require prohibitive memory and compute resources, often necessitating multi-GPU setups (Luo et al., 2025). Because SMART treats spatial queries independently, we can exploit this property to make training tractable. During training, we randomly subsample query points from the full set and compute the loss only on this subset. This substantially reduces memory consumption and accelerates training, while ensuring that the model observes the entire spatial domain over the course of training in expectation. At inference time, SMART can evaluate the full-resolution field without retraining. To keep GPU memory usage constant, we partition the full set of query points into smaller subregions and evaluate them sequentially, following the strategy proposed by Alkin et al. (2025a). Since queries do not interact, this sequential evaluation produces identical results to evaluating all points simultaneously, but with significantly reduced memory requirements.

## 4. Related Work

We briefly elaborate on ML for solving PDEs in general. Furthermore, we provide an overview of related work of neural surrogates for industry-level fluid dynamics simulations, which we divide into mesh-free models and mesh-dependent models that rely on the simulation mesh as input.

**Machine Learning for PDE-Solving.**  ML is increasingly used to approximate solution functions of PDEs. Among the leading methodologies are neural operators (Kovachki et al., 2023), with notable variants such as the Graph Neural

Operator (GNO; Li et al. (2020)) and the Fourier Neural Operator (FNO; Li et al. (2021)), alongside several extensions (Li et al., 2023a; Tran et al., 2023). Transformer-based architectures have also gained traction for PDE solving (Cao, 2021; Li et al., 2023c; Hao et al., 2023; Wu et al., 2024), typically leveraging attention mechanisms over the spatial domain. Another emerging direction involves neural fields, which approximate PDE solution functions through implicit representations (Yin et al., 2023; Serrano et al., 2023; Hagnberger et al., 2024; Knigge et al., 2024). Recent models compress the spatial domain into compact latent structures, often using predefined or learned latent meshes and aggregation operations, and solve the PDE in this latent space (Chen & Wu, 2024; Hagnberger et al., 2026; Wen et al., 2025).

**Mesh-free Neural Surrogates for Industry-level Problems.** GINO (Li et al., 2023b) combines GNO and FNO for simulations involving complex and varying geometries. The model uses GNO layers with message-passing (Gilmer et al., 2017) to transform the input geometry into a uniform latent mesh, FNO layers as a processor, and GNO layers to map the latent representation back for arbitrary queries. Geometry-Preserving Universal Physics Transformer (GP-UPT; Bleeker et al. (2025)) extends the Universal Physics Transformer (UPT; Alkin et al. (2024)) to large-scale aerodynamic simulations. The architecture employs a geometry-aware encoder with message-passing and attention to derive a compressed latent representation of the CFD surface mesh, which is directly used for surface predictions. A field-based decoder utilizing cross-attention allows for arbitrary queries in the simulation domain, including the surface and volume. In a second step, GP-UPT is fine-tuned on the geometry to remove the dependency on the CFD surface mesh.

**Mesh-dependent Neural Surrogates for Industry-level Problems.** GP-UPT was originally introduced by Bleeker et al. (2025) and later revised into the Anchored-Branched Universal Physics Transformer (AB-UPT; Alkin et al. (2025a)). The updated model combines message passing and attention to build a compressed latent representation of the input geometry, which is then processed by a physics Transformer with separate branches for surface and volume predictions. Each branch operates on a set of randomly sampled mesh points (anchors) together with the corresponding surface or volume query coordinates. Self-attention over the anchors captures spatial dependencies but limits scalability, while cross-attention between anchors and queries enables query independence and supports inference on millions of points. Transolver (Wu et al., 2024) proposes physics-attention, which is jointly applied over the geometry point cloud and the query points. To reduce the quadratic cost of full self-attention, physics-attention introduces slice tokens. The model performs cross-attention between the input and the slice tokens, self-attention only

*Table 1.* Used datasets with their average number of points.

| Dataset | Geometry Points | Surface Points | Volume Points |
|---|---|---|---|
| ShapeNetCar | 3,682 | 3,682 | 29,498 |
| AhmedML | 101,405 | 1,069,004 | 21,195,967 |
| SHIFT-SUV | 2,548,037 | 2,521,717 | 50,507,366 |
| SHIFT-Wing | 1,623,086 | 3,246,168 | 5,970,264 |

among the slice tokens, and final cross-attention back to the input. This design preserves global information exchange as in self-attention while avoiding the full quadratic complexity. Transolver++ (Luo et al., 2025) extends this approach to large-scale simulations by introducing a local adaptive mechanism to improve the representation of the slice tokens and distributing computation across multiple GPUs.

## 5. Experiments and Evaluation

We design our experiments to answer the following research questions. ● **RQ1:** How effective is SMART trained and tested on subsampled aerodynamic simulations with 16k mesh points? ● **RQ2:** Which errors does the model achieve on industry-level simulations with millions of mesh points? ● **RQ3:** How does model performance change when the CFD-optimized surface and volume meshes are replaced at inference with the CAD surface mesh and a uniformly sampled volume mesh? ● **RQ4:** What happens to the model if a distribution shift in the query coordinates happens for inference? ● **RQ5:** Which components contribute to the performance of SMART?

### 5.1. Datasets

We train and evaluate the models on three datasets for automotive aerodynamic simulations. In particular, we use the ShapeNetCar (Umetani & Bickel, 2018), AhmedML (Ashton et al., 2024), and SHIFT-SUV (Luminary Cloud, 2025a) datasets. Additionally, we test and train the models on the SHIFT-Wing dataset (Luminary Cloud, 2025b) to evaluate the models for aerospace applications. The ShapeNetCar dataset consists of car bodies without details and tens of thousands of mesh points. The AhmedML dataset uses simple Ahmed bodies as geometries and has millions of mesh points. The SHIFT-SUV and SHIFT-Wing datasets are industry-level datasets with car bodies and aircraft designs with millions of points and fine details. Table 1 shows the average number of points for each dataset, and Section F provides details about the datasets.

### 5.2. Baseline Models

We compare the proposed model against GINO (Li et al., 2023b), OFormer (Li et al., 2023c), GNOT (Hao et al., 2023), Transolver (Wu et al., 2024), LNO (Wang & Wang, 2024), and AB-UPT (Alkin et al., 2025a). We include two

*Table 2.* Rel. L2 errors of models trained and tested on **subsampled** datasets with 16k points on the surface and volume. Values in parentheses indicate the percentage deviation to SMART, and underlined values indicate the second-best errors.

| Model | Relative L2 Error $\times 10^{-2}$ ($\downarrow$) | | | | | | | |
| --- | --- | --- | --- | --- | --- | --- | --- | --- |
| | ShapeNetCar | | AhmedML | | SHIFT-SUV | | SHIFT-Wing | |
| | Surface | Volume | Surface | Volume | Surface | Volume | Surface | Volume |
| GINO | 14.4208 (+113%) | 10.2862 (+105%) | 16.0332 (+414%) | 17.3267 (+236%) | 0.1215 (+594%) | 51.5540 (+723%) | 4.3333 (+7680%) | 75.1867 (+413%) |
| OFormer | 9.1896 (+36%) | 7.3458 (+46%) | 5.0421 (+62%) | 7.3925 (+44%) | 0.0224 (+28%) | 10.9833 (+75%) | 0.2637 (+373%) | 43.2504 (+195%) |
| GNOT | 11.2701 (+66%) | 9.1364 (+82%) | 3.7068 (+19%) | 5.8419 (+13%) | 0.0188 (+7%) | 7.1901 (+15%) | 0.0749 (+34%) | 18.5032 (+26%) |
| Transolver | 8.4825 (+25%) | 7.1897 (+43%) | 3.4441 (+10%) | 5.4736 (+6%) | 0.0183 (+5%) | 6.8487 (+9%) | 0.0708 (+27%) | 17.8178 (+22%) |
| LNO | 12.5501 (+85%) | 10.4892 (+109%) | 5.1230 (+64%) | 10.2931 (+100%) | 0.0334 (+91%) | 18.8967 (+202%) | 0.3747 (+573%) | 46.2939 (+216%) |
| GP-UPT | 6.9672 (+3%) | 5.5198 (+10%) | 3.3879 (+9%) | 5.6393 (+9%) | 0.0206 (+18%) | 9.0023 (+44%) | 0.1537 (+176%) | 39.8774 (+172%) |
| MD AB-UPT | 6.9959 (+3%) | 5.2457 (+4%) | 3.4478 (+11%) | 5.1812 (+1%) | 0.0181 (+3%) | 6.5665 (+5%) | 0.0696 (+25%) | 16.1130 (+10%) |
| MF AB-UPT | 6.9406 (+2%) | 5.2709 (+5%) | 3.2166 (+3%) | 5.1712 (+0%) | 0.0181 (+3%) | 6.4180 (+2%) | 0.0774 (+39%) | 39.4740 (+169%) |
| SMART | **6.7734** | **5.0215** | **3.1195** | **5.1511** | **0.0175** | **6.2627** | **0.0557** | **14.6639** |

variants of AB-UPT: The Mesh-Dependent AB-UPT (MD AB-UPT), which uses the simulation mesh as anchors, and the Mesh-Free AB-UPT (MF AB-UPT), which does not utilize the simulation mesh. We also include GP-UPT (Bleeker et al., 2025), the predecessor of AB-UPT, which we have reimplemented according to the description in their paper. GNOT, Transolver, and MD AB-UPT leverage the simulation mesh as input, while the other models are mesh-free.

## 5.3. Results

For each model, we train and evaluate multiple models with different initializations and report the mean values of the relative L2 error. We report the errors for the surface prediction, which includes the pressure, and the volume prediction, including the velocity in $x, y, z$ directions. We refer to Section L for the full results with the standard deviations and for several qualitative results. Additionally, we refer to Section I for details on the setup used for the experiments.

**RQ1.** In the first experiment, we train and evaluate all models on randomly subsampled data consisting of 16k surface points and 16k volume points. As shown in Table 2, SMART consistently outperforms all baselines across all datasets. Mesh-dependent models such as GNOT, Transolver, and MD AB-UPT also achieve strong performance across all datasets, and the mesh-free variant of AB-UPT also performs well on the automotive datasets. However, MF AB-UPT fails to predict the volume field on SHIFT-Wing, making SMART the only model capable of accurately predicting the full flow field on this dataset without access to the simulation mesh. This aspect is especially important for this dataset, as its adaptively refined mesh cannot be used for inference, as it would require a complete simulation to generate it. Overall, these results demonstrate that SMART provides accurate, mesh-free predictions for aerodynamic simulations, even in settings where other mesh-free methods struggle.

**RQ2.** Next, we evaluate the models on industry-scale simulations by querying the models with the full spatial resolution (i.e., millions of points on the surface and volume).

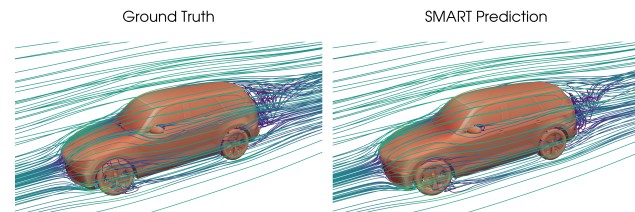

*Figure 3.* True and predicted pressure field and velocity field (as streamlines) for a random sample from the SHIFT-SUV dataset.

We train the models on randomly subsampled data with 16k points, as in (RQ1), and evaluate them on the full resolution. As denoted in Section 3.2 about the training and inference, training on randomly subsampled data significantly reduces the training time, while still allowing the model to see enough points on the surface and volume to learn the full fields accurately. Table 3 shows the errors of the models evaluated on the full resolution. Similar to the results on the subsampled data, SMART outperforms all baselines except for MF AB-UPT on the volume field of the AhmedML dataset. Inference on full resolution with GNOT and Transolver yields an out-of-memory error on an NVIDIA A100 80GB GPU due to the use of self- or physics-attention on the query coordinates. For the other query-independent models, we can divide the queries into subregions and query the models sequentially on the subregions to reduce the memory consumption. Figure 3 shows the prediction of a sample from the SHIFT-SUV dataset.

**RQ3.** In the previous experiments, we assumed that the simulation mesh was available for all geometries. In practice, however, a CFD mesh is rarely provided for new geometries. We therefore evaluate the models in a more realistic setting where no simulation mesh is given. Instead, the models operate on (i) a surface mesh obtained directly from the CAD geometry, which is not optimized for CFD, and (ii) a volume discretization obtained from a uniform sampling of the domain. Details on how we construct the corresponding ground-truth fields are provided in Section G.1. Table 4, dataset (a) shows that mesh-free models (i.e., models that do

*Table 3.* Rel. L2 errors of models tested on **full-resolution** datasets with millions of points on the surface and volume. Values in parentheses indicate the percentage deviation to SMART, and underlined values indicate the second-best errors.

| Model | Relative L2 Error $\times 10^{-2}$ ($\downarrow$) | | | | | | | |
| | ShapeNetCar | | AhmedML | | SHIFT-SUV | | SHIFT-Wing | |
| | Surface | Volume | Surface | Volume | Surface | Volume | Surface | Volume |
|---|---|---|---|---|---|---|---|---|
| GINO | 14.4206 (+113%) | 10.3250 (+105%) | 16.1834 (+391%) | 17.4560 (+229%) | 0.1218 (+596%) | 51.5854 (+720%) | 4.3343 (+7654%) | 75.2558 (+411%) |
| OFormer | 9.1898 (+36%) | 7.3559 (+46%) | 5.1494 (+56%) | 7.5140 (+42%) | 0.0225 (+29%) | 10.9924 (+75%) | 0.2653 (+375%) | 43.3266 (+194%) |
| GNOT | 13.0965 (+93%) | 11.2348 (+123%) | | | Out of memory (A100 80GB GPU) | | | |
| Transolver | 9.6288 (+42%) | 8.9447 (+78%) | | | Out of memory (A100 80GB GPU) | | | |
| LNO | 12.5501 (+85%) | 10.5334 (+109%) | 5.2975 (+61%) | 10.3790 (+96%) | 0.0335 (+91%) | 18.9424 (+201%) | 0.3834 (+586%) | 46.3603 (+215%) |
| GP-UPT | 6.9671 (+3%) | 5.5764 (+11%) | 3.4917 (+6%) | 5.7412 (+8%) | 0.0206 (+18%) | 9.0318 (+43%) | 0.1567 (+180%) | 39.9266 (+171%) |
| MD AB-UPT | 7.0654 (+4%) | 5.3283 (+6%) | 3.6450 (+11%) | 5.3253 (+0%) | 0.0181 (+3%) | 6.5956 (+5%) | 0.0705 (+26%) | 16.1691 (+10%) |
| MF AB-UPT | 6.9814 (+3%) | 5.3298 (+6%) | 3.3310 (+1%) | 5.3012 (-0%) | 0.0181 (+3%) | 6.4445 (+2%) | 0.0775 (+39%) | 39.5142 (+168%) |
| SMART | 6.7734 | 5.0357 | 3.2938 | 5.3050 | 0.0175 | 6.2946 | 0.0559 | 14.7187 |

not rely on the CFD-optimized simulation mesh) maintain their accuracy or even improve in this setting. In contrast, models such as GNOT, Transolver, and MD AB-UPT exhibit a substantial increase in error. This degradation occurs because these methods rely on the structure of the CFD mesh, which is no longer present when using CAD-derived surface meshes and uniformly sampled volume points. Where competing mesh-free models suffer from higher errors, SMART achieves lower errors, showing that it removes the dependency on simulation meshes and can be deployed directly in practical applications.

**RQ4.** Similar to the previous experiment, we examine how the models behave under a distribution shift in the query coordinates. During training, all models see points randomly sampled from the entire simulation domain as in (RQ1), but at test time, we evaluate them only on a subset of the domain. Specifically, we consider a practical scenario in which only the aerodynamics at the rear of the car are of interest, for example, when comparing different spoiler designs. In such cases, it is sufficient to query the models only at rear-region coordinates, which induces a clear shift between the training and test query distributions. We emulate this setup on the SHIFT-SUV dataset using models trained on the full spatial domain but evaluated only on mesh points located at the rear of the vehicle. As shown in Table 4, dataset (b), mesh-free models, including SMART, remain robust under this query distribution shift, exhibiting no or only negligible increases in error. In contrast, mesh-dependent models, such as Transolver, show a noticeable degradation in accuracy.

**RQ5.** Lastly, we conduct an ablation study in Section K to investigate which components contribute to the model's performance. The results demonstrate that increasing the size of the latent geometries consistently reduces errors. Furthermore, the geometry cross-attention and the use of a coarse geometry as a scaffold in the geometry encoder significantly improve the errors. Additionally, the joint geometry-physics update via the cross-layer interaction of the encoder and decoder significantly improves the performance of SMART.

*Table 4.* Rel. L2 errors of models tested on two variations of the SHIFT-SUV dataset. (a) Using the **CAD-mesh** as the surface mesh (600k points) and an **uniform mesh** as the volume mesh (2M points). (b) Using only the mesh points corresponding to the **rear region** of the vehicles (1.5M points on surface and volume).

| Model | Relative L2 Error $\times 10^{-2}$ ($\downarrow$) | | | |
| | (a) SHIFT-SUV Uniform | | (b) SHIFT-SUV Rear | |
| | Surface | Volume | Surface | Volume |
|---|---|---|---|---|
| GINO | 0.1225 (+596%) | 46.9 (+688%) | 0.1001 (+581%) | 57.5 (+870%) |
| OFormer | 0.0225 (+28%) | 12.5 (+110%) | 0.0181 (+23%) | 11.4 (+92%) |
| GNOT | 0.0765 (+335%) | 46.0 (+673%) | 0.0326 (+122%) | 22.9 (+286%) |
| Transolver | 0.0622 (+253%) | 25.1 (+322%) | 0.0321 (+118%) | 18.0 (+204%) |
| LNO | 0.0334 (+90%) | 25.6 (+330%) | 0.0257 (+75%) | 20.7 (+249%) |
| GP-UPT | 0.0207 (+18%) | 11.4 (+92%) | 0.0174 (+18%) | 9.26 (+56%) |
| MD AB-UPT | 0.1172 (+566%) | 141 (+2270%) | 0.0660 (+349%) | 33.7 (+468%) |
| MF AB-UPT | 0.0181 (+3%) | 7.35 (+24%) | 0.0156 (+6%) | 6.27 (+6%) |
| SMART | 0.0176 | 5.95 | 0.0147 | 5.93 |

## 6. Limitations

While SMART demonstrates strong performance on large-scale time-independent aerodynamic simulations, several limitations remain. First, our evaluation focuses on external aerodynamics in automotive and aerospace settings. Extending the benchmark to additional domains, such as turbomachinery and biomedical fluid dynamics, would provide a broader assessment of generalization capabilities. Second, the current model targets time-independent simulations and does not handle time-dependent flows. Incorporating temporal dynamics, for example, through latent time-stepping models, is an important direction for future work and would enable applications in transient CFD. Third, SMART does not enforce physical laws such as conservation laws. Incorporating physics explicitly in the architecture or training objective may improve accuracy and robustness. Finally, the currently implemented training strategy relies on random subsampling of query points. While this enables efficient training on large domains and works surprisingly well, it may underrepresent rare or highly localized flow phenomena unless sampling strategies are adapted accordingly.

## 7. Conclusion

We showed that existing surrogate models such as GNOT, Transolver, and the mesh-dependent AB-UPT rely heavily

on the simulation mesh, not just as a set of spatial query locations, but as an active input that shapes the prediction. As a result, these models struggle when evaluated on meshes that differ from those seen during training. This dependence poses a practical barrier for real-world deployment. Generating high-quality meshes for new geometries is computationally expensive, and adaptive meshes are only available after running a full CFD simulation. SMART addresses this limitation by providing a fully mesh-free surrogate that performs reliably across all tested scenarios. Despite not using the simulation mesh, SMART matches or surpasses the accuracy of mesh-dependent models. Its performance stems from three key components: a geometry encoder, a cross-attention-based decoder, and a joint geometry-physics update mechanism that couples both representations throughout the model. In this work, we focused on time-independent aerodynamic simulations due to their central role in engineering applications. Future work includes extending SMART to time-dependent flows and exploring more principled strategies for constructing the initial coarse latent geometry.

## Code Availability

The implementation of SMART, together with all associated experiment scripts, is available on GitHub at: **https://github.com/jhagnberger/smart**

## Impact Statement

Cutting-edge neural surrogates greatly reduce the time and cost associated with simulations used in engineering, such as design optimization. Consequently, this helps to accelerate the design process and reduce energy usage and carbon emissions. Yet, a notable concern is the possibility of misuse by bad actors, since fluid dynamics simulations can also be applied to the development of military technologies such as missiles.

## Acknowledgements

We gratefully acknowledge Luminary Cloud for providing access to the SHIFT-SUV and SHIFT-Wing datasets. We further thank Michael Emory (Luminary Cloud) for supplying detailed information regarding the SHIFT datasets. This work was funded by Deutsche Forschungsgemeinschaft (DFG, German Research Foundation) under Germany's Excellence Strategy - EXC 2075 – 390740016. We acknowledge the support of the Stuttgart Center for Simulation Science (SimTech). The authors thank the International Max Planck Research School for Intelligent Systems (IMPRS-IS) for supporting Mathias Niepert. Additionally, we acknowledge the support of the German Federal Ministry of Research, Technology and Space (BMFTR) as part of InnoPhase (funding code: 02NUK078). Lastly, we acknowledge the support of the European Laboratory for Learning and Intelligent Systems (ELLIS) Unit Stuttgart.

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

# APPENDIX FOR SMART

# A. Details on the Problem Definition

We consider the problem of learning a neural surrogate model for simulating the air flow around geometries such as car and aircraft bodies. First, we elaborate on the different representations of the geometry, followed by a brief explanation about generating ground truth data with a numerical solver, and conclude by specifying the training objective.

**Representations of Geometries.**  Geometries are typically created in CAD software, which stores them in application-specific formats. For use in other tools, they are commonly exported as STL files, which approximate the shape with a surface mesh composed of triangles. STL export is broadly supported and remains computationally inexpensive compared to generating a CFD-optimized mesh. A point cloud can be further extracted from the mesh by either computing the cell center and representing each cell center as a point or by using the mesh vertices as points. We denote the point cloud representation of the geometry as $G = \{(x, y, z)_i\}_{i=1}^N \in \mathbb{R}^{N \times 3}$ consisting of $N$ points. The higher the resolution of the mesh, the higher the number of points. Depending on the downstream task, either the mesh or the point cloud representation of the geometry is more suitable. For instance, CFD-mesh generation pipelines rely on the STL representation, whereas ML models typically operate on the point cloud representation. Figure 4 illustrates different representations and the steps of obtaining them.

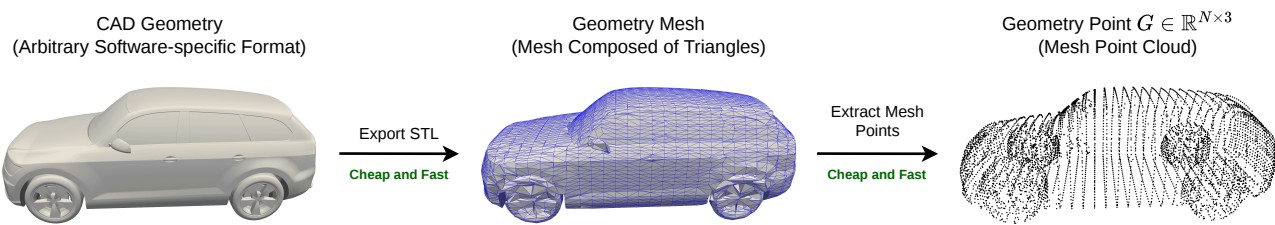

*Figure 4.* Geometries can be either represented in an application-specific CAD format, a derived geometry mesh (STL file), or a point cloud. The geometry mesh and point cloud are subsampled solely for visualization purposes.

**Generating Ground Truth Data with Numerical Solvers.**  The flow around the geometry is governed by underlying PDEs such as the Navier-Stokes equations, with the geometry defining boundary conditions that influence the solution (i.e., physical quantities of the flow around the geometry). Additional simulation parameters $\boldsymbol{\xi} \in \mathbb{R}^{N_p}$ (e.g., yaw angle or angle of attack) further influence the solution. To obtain a solution for a given geometry and parameters, the PDE will be solved numerically. This requires computing a CFD-optimized simulation mesh for the surface and volume. Mesh-generation tools take the STL representation as input and compute a CFD-optimized simulation mesh, refining regions where strong flow gradients are expected, such as sharp edges or narrow gaps, independent of how the STL is tessellated. OpenFOAM's snappyHexMesh and Gmsh (Geuzaine & Remacle, 2009) are examples of such tools. The resulting simulation mesh consists of a surface mesh $P_S$ and a volume mesh $P_V$, which we denote using a point cloud notation consistent with the geometry $G$. A numerical solver, such as OpenFOAM (Weller et al., 1998) or SU2 (Economon et al., 2016), resolves the governing physics on this given mesh, producing physical quantities for each surface cell and volume point. Depending on the solver, the output may be a directly computed time-averaged field (e.g., steady RANS), or it may result from a transient simulation with physical timesteps whose instantaneous fields are averaged in time. We denote the time-averaged physical quantities for a specific location given the geometry and parameters as $\mathbf{u}((x, y, z), G, \boldsymbol{\xi}) \in \mathbb{R}^{N_c}$, and over all surface and volume locations as $\mathbf{u}(P, G, \boldsymbol{\xi}) = \{\mathbf{u}((x, y, z)_j, G, \boldsymbol{\xi})\}_{j=1}^M \in \mathbb{R}^{M \times N_c}$ with $P = P_S \cup P_V \in \mathbb{R}^{M \times 3}$. We merge the solution from the surface and volume into a single solution function for notational convenience. The final sample is a quadruple $\left(G, \boldsymbol{\xi}, P, \mathbf{u}(P, G, \boldsymbol{\xi})\right)$ consisting of the geometry $G \in \mathbb{R}^{N \times 3}$, simulation parameters $\boldsymbol{\xi} \in \mathbb{R}^{N_p}$, query positions $P \in \mathbb{R}^{M \times 3}$ obtained from the simulation mesh, and the physical quantities for the positions $\mathbf{u}(P, G, \boldsymbol{\xi}) \in \mathbb{R}^{M \times N_c}$. Figure 5 illustrates the workflow of a numerical solver, including the meshing stage.

**Dataset.**  A dataset consists of many samples, each represented as a quadruple $\left(G, \boldsymbol{\xi}, P, \mathbf{u}(P, G, \boldsymbol{\xi})\right)$ containing a geometry point cloud $G \in \mathbb{R}^{N \times 3}$, its simulation parameters $\boldsymbol{\xi} \in \mathbb{R}^{N_p}$, the corresponding CFD mesh points $P \in \mathbb{R}^{M \times 3}$, and the resulting physical fields $\mathbf{u}(P, G, \boldsymbol{\xi}) \in \mathbb{R}^{M \times N_c}$. Across the dataset, both the geometries and the simulation parameters vary, as each geometry is generated from a parametric shape model with different parameter settings. This variation produces a broad range of flow configurations with distinct boundary conditions and operating regimes, which is essential for training surrogate models that generalize beyond a single shape or parameter choice. Because each geometry induces its own

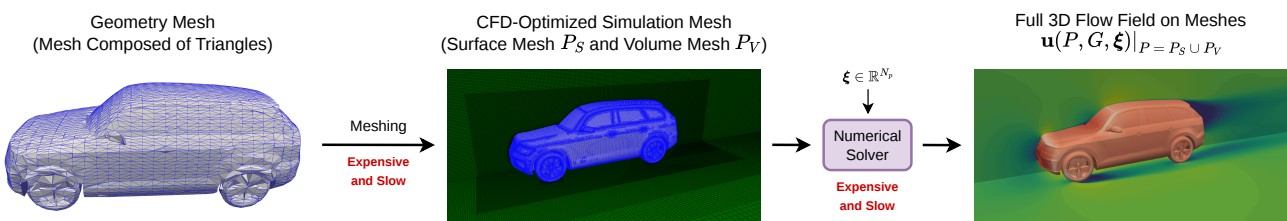

*Figure 5.* The geometry mesh (i.e., STL file) is used as input to generate the simulation mesh consisting of the surface mesh $P_S$ and volume mesh $P_V$. The simulation mesh generation is both computationally expensive and slow. A numerical solver resolves the governing physics on the simulation mesh, producing physical quantities $\mathbf{u}((x, y, z), G, \boldsymbol{\xi})$ for each of the mesh locations.

CFD-optimized mesh, the number of mesh points $M$ and the spatial distribution of $P$ differ from sample to sample, reflecting the adaptive refinement strategies of the meshing tools. We denote the full dataset as $\mathcal{D} = \left\{ \left( G, \boldsymbol{\xi}, P, \mathbf{u}(P, G, \boldsymbol{\xi}) \right)_s \right\}_{s=1}^{N_{data}}$ and use $\mathcal{D}_{train}$ and $\mathcal{D}_{test}$ to denote the training and test splits, respectively.

**Training Objective.** The training objective is to optimize the parameters $\theta$, comprising the weights and biases of the model $f_\theta$, so that it best approximates the true solution function $\mathbf{u}((x, y, z), G, \boldsymbol{\xi})$ by minimizing the empirical risk over the dataset $\mathcal{D}_{train}$ as

$$\underset{\theta \in \Theta}{\operatorname{argmin}} \, \mathcal{L}_\theta = \underset{\theta \in \Theta}{\operatorname{argmin}} \, \mathbb{E}_{\left( G, \boldsymbol{\xi}, P, \mathbf{u}(P, G, \boldsymbol{\xi}) \right) \sim \mathcal{D}_{train}} \left[ L \Big( \underbrace{f_\theta(P, G, \boldsymbol{\xi})}_{\text{Prediction}}, \underbrace{\mathbf{u}(P, G, \boldsymbol{\xi})}_{\text{Ground Truth}} \Big) \right] \tag{8}$$

where $L$ is a suitable loss function such as MSE or relative L2 error and $f_\theta(P, G, \boldsymbol{\xi})$ is the model's prediction. In this work, we use the sum of the relative L2 errors computed independently over the surface and volume points as a loss function, as described in Section I about the experiment details. Industry-scale datasets have millions of surface and volume mesh points, which makes training on the full resolution expensive. Thus, the empirical risk can be approximated by training on a smaller number of randomly sampled mesh points (i.e., query points) as

$$\underset{\theta \in \Theta}{\operatorname{argmin}} \, \mathcal{L}_\theta = \underset{\theta \in \Theta}{\operatorname{argmin}} \, \mathbb{E}_{\left( G, \boldsymbol{\xi}, P, \mathbf{u}(P, G, \boldsymbol{\xi}) \right) \sim \mathcal{D}_{train}, \, p_1, \ldots, p_{M'} \sim \mathcal{U}(P)} \left[ L \big( f_\theta(P, G, \boldsymbol{\xi}), \mathbf{u}(P, G, \boldsymbol{\xi}) \big) \big|_{P = \{p_1, \ldots, p_{M'}\}} \right] \tag{9}$$

where a sample from the dataset is selected, as well as a random subset of $M'$ query points where the loss is evaluated. In expectation, the model will still see all mesh points to learn the full physics field.

## B. Independence of Arbitrary Queries

We identify the independence of arbitrary spatial queries as a key desideratum for neural surrogate models. Figure 6 illustrates this concept in a 2D spatial domain, contrasting query-independent and query-dependent formulations. In a query-independent setup, each query is allowed to attend only to a global representation (e.g., via cross-attention) and not to other queries. Crucially, queries may read from this global representation but must not modify it. In contrast, query dependence arises when queries also attend to one another (e.g., through self-attention over the query set), thereby violating independence. Formally, query independence requires $\forall p \in \Omega \quad \forall P = \{p_1, \ldots, p_{j-1}, p, p_{j+1}, \ldots\} \subseteq \Omega : f_\theta(p, G, \boldsymbol{\xi}) = f_\theta(P, G, \boldsymbol{\xi})_j$ which means that querying the model with only one arbitrary query point $p$ yields the same output as querying $p$ together with an arbitrary query set $P$. As a consequence, queries can be arbitrarily partitioned, ordered, or batched, since each query in $P$ is independent of the others. For the proposed SMART model, query independence requires that queries do not interact and do not alter the latent geometry. Therefore, only the decoder must be analyzed to verify query independence. Using only pointwise operations (e.g., pointwise positional encodings and MLPs) on the queries $P$ and cross-attention between the queries and latent geometries automatically ensures that SMART satisfies query independence.

## C. Mesh-free vs. Mesh-dependent Neural Surrogate Models

We highlight mesh-free or mesh-independent surrogate models as a central advantage of neural approaches to CFD simulations. Unlike mesh-dependent models, which take the CFD-optimized simulation mesh as an input that directly influences their predictions, mesh-free models operate on arbitrary query points $P$ without assuming any spatial structure.

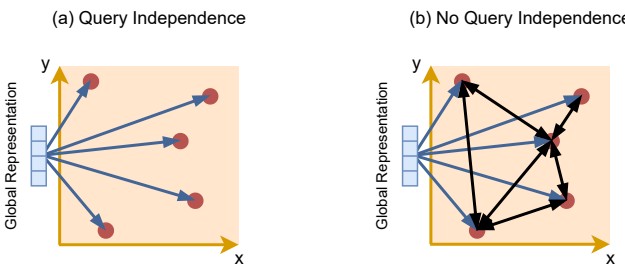

*Figure 6.* Example of query independence and query dependence. (a) Query points attend only to information from a global representation, without changing it, which ensures query independence. (b) Query points additionally share information (e.g, with self-attention), which breaks query independence.

**Mesh-free Neural Surrogates.** During training, both model types learn to predict physical quantities at locations on the simulation mesh, where ground-truth values from the solver are available. The crucial difference is that mesh-free models use these mesh points only as supervised samples. They do not encode the mesh topology or let it shape their internal representation. As a result, they can be queried at any set of points during inference. This removes the need to generate a CFD-optimized mesh at test time, and inexpensive geometry meshes or uniformly sampled points in the volume are sufficient. Consequently, the entire meshing and numerical-solver pipeline can be bypassed, as illustrated in Figure 7.

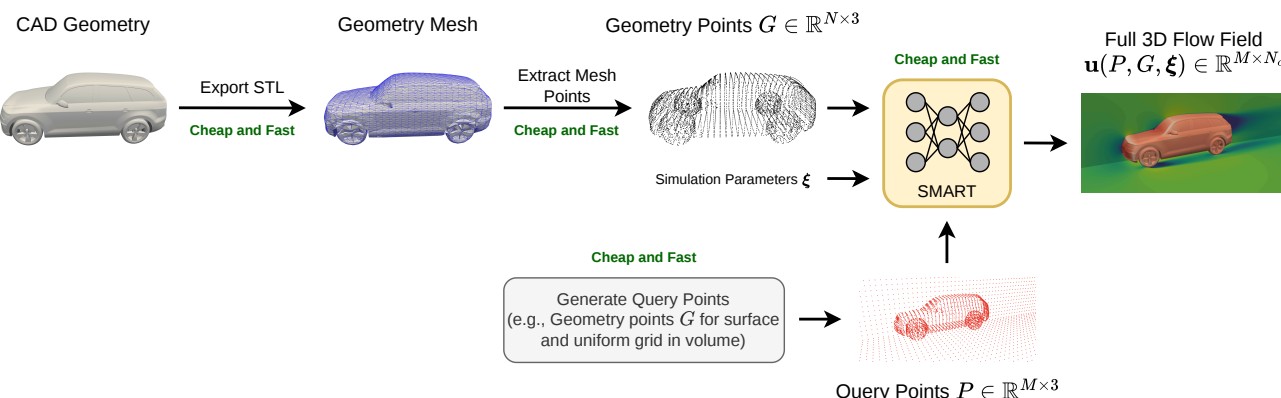

*Figure 7.* Mesh-free models require as input only the geometry, which is typically represented as a point cloud, and a set of arbitrary query points at which the physical field is evaluated. For instance, query points can be generated by using the geometry point cloud for the surface and a uniform grid in the volume.

**Mesh-dependent Neural Surrogates.** Mesh-dependent models behave differently. They rely on the CFD-optimized mesh as a structural input, and mechanisms such as self-attention over mesh points exploit spatial relationships and implicitly encode the mesh topology. This biases the model toward the solver's discretization and forces it to use the same CFD-optimized mesh during inference, as shown in Figure 8. When these models are queried with alternative point sets, such as a coarse geometry mesh or a uniform volumetric grid, their performance deteriorates sharply. Our experiments confirm this behavior.

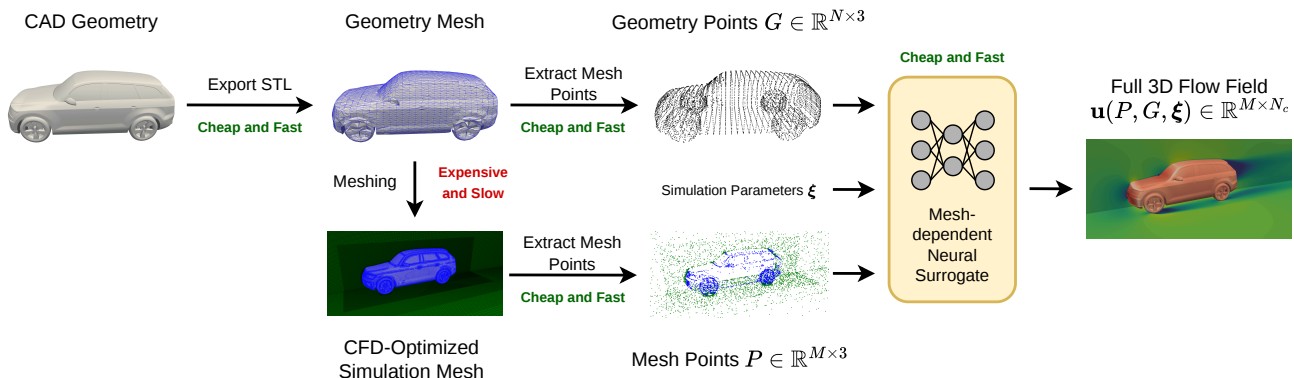

*Figure 8.* Mesh-dependent neural surrogates require as input the geometry, often given as a point cloud, and the CFD-optimized simulation mesh, which is also typically represented as a point cloud. Generating this simulation mesh is both computationally expensive and time-consuming. The mesh points $P$ are subsampled solely for visualization purposes.

## D. Continuation of Related Work

We provide additional related work on neural surrogates, including point-cloud neural networks and graph neural networks.

**Point-cloud Neural Networks.**   Point-cloud architectures such as PointNet (Qi et al., 2017a) and its successor PointNet++ (Qi et al., 2017b), originally developed for classification and semantic segmentation of point clouds, have also been explored for 3D aerodynamic surrogate modeling (Elrefaie et al., 2024b; Qiu et al., 2025). DoMINO (Ranade et al., 2025) represents a model specifically tailored for aerodynamic predictions that employs a multi-scale mechanism, based on point convolution evaluated at multiple ball radii, to extract a global representation of the geometry in the spatial domain. For each spatial query, a local representation is extracted from the global representation, which is then used to predict the physical quantities at this location. Given that both geometry and CFD data are commonly represented as point clouds, point-cloud architectures align well with the structure of the problem.

**Graph Neural Networks.**   Graph Neural Networks (GNNs) are also leveraged for aerodynamic surrogate modeling, as the geometries, simulation meshes, and local neighborhoods of point clouds can be represented as graphs. Elrefaie et al. (2024a) introduces RegDGCNN, a dynamic graph convolutional neural network model, which predicts aerodynamic performance measures such as drag coefficients from car bodies. RegDGCNN builds on the DGCNN architecture (Wang et al., 2019), which integrates PointNet-style spatial encoding with GNN operations, and adapts it to regress the aerodynamic drag of cars. Elrefaie et al. (2024b) further investigates the use of graph convolution neural networks (Kipf & Welling, 2017) for this setting and extends RegDGCNN to predict full pressure fields over car surfaces.

## E. Comparison to Related Models

In this section, we compare the related baseline models with the proposed SMART model. Table 5 shows a comparison based on the following properties.

- Independent queries: Indicates whether the model supports independent spatial queries. Independent queries allow querying the model sequentially with subregions, which drastically reduces the GPU memory consumption for large-scale simulations with millions of points. We define query independence as a desideratum of a neural surrogate model.

- Mesh-free: Indicates whether the model requires the CFD-optimized simulation mesh as an input feature (e.g., as anchors). Using the simulation mesh requires time-consuming meshing for new geometries during inference. We define mesh-free operation as a desideratum of a neural surrogate model.

- Decoupled geometry and queries: Models that decouple the input geometry from the spatial queries in the physical field can scale the geometry resolution independently of the number of spatial queries.

- Joint geometry-physics update: Indicates whether the model refines geometric and physical representations jointly throughout the network, allowing queries to attend to intermediate representations, or whether queries are only incorporated at the final decoding stage following a strict encode-process-decode paradigm.

- Compressed latent space: The model operates in a compressed latent space and not on the original input point cloud to reduce computational costs.

- Geometry-informed latent space: The model incorporates geometry information into the latent space before the application of an encoder.

- Iterative geometry encoding: Indicates whether the model iteratively attends to the geometry to integrate geometric information into a latent representation of the geometry or physical field.

- Unified branch: Indicates whether the model employs a single and unified branch for surface and volume prediction or two separate branches. A unified branch enables the model to learn the physical field jointly, which may improve performance. Separate branches may also improve the predictions by enabling each branch to specialize, but they typically need a mechanism to exchange information between the branches (e.g., cross-attention), which adds additional components to the model architecture.

- Requires neighborhood graph: Indicates whether the model needs to construct a neighborhood graph for GNN-based encoder or decoder layers, which adds additional hyperparameters (e.g., radius) and components to the model.

*Table 5.* Overview of properties of related models and the proposed SMART model.

| Property | GINO | OFormer | GNOT | Transolver | LNO | GP-UPT | MD AB-UPT | MF AB-UPT | SMART |
|---|---|---|---|---|---|---|---|---|---|
| Independent queries (D2) | ✓ | ✓ | ✗ | ✗ | ✓ | ✓ | ✓ | ✓ | ✓ |
| Mesh-free (D3) | ✓ | ✓ | ✗ | ✗ | ✓ | ✓ | ✗ | ✓ | ✓ |
| Decoupled geometry and queries | ✓ | ✗ | ✗ | ✗ | ✓ | ✓ | ✓ | ✓ | ✓ |
| Joint geometry-physics update | ✗ | ✗ | ✗ | ✓ | ✗ | ✗ | ✓ | ✓ | ✓ |
| Compressed latent space | ✓ | ✗ | ✗ | ✗ | ✓ | ✓ | ✓ | ✓ | ✓ |
| Geometry-informed latent space | ✗ | ✓ | ✓ | ✓ | ✗ | ✓ | ✓ | ✓ | ✓ |
| Iterative geometry encoding | ✗ | ✗ | ✓ | ✗ | ✗ | ✗ | ✗ | ✗ | ✓ |
| Unified branch | ✓ | ✓ | ✓ | ✓ | ✓ | ✓ | ✗ | ✗ | ✓ |
| No neighborhood graph required | ✗ | ✓ | ✓ | ✓ | ✓ | ✗ | ✗ | ✗ | ✓ |

# F. Additional Details on the Datasets

We provide a detailed characterization of the datasets utilized in our experiments, namely the ShapeNetCar, AhmedML, SHIFT-SUV, and SHIFT-Wing datasets. Additionally, we provide a visualization of one sample from each dataset.

## F.1. ShapeNetCar

The ShapeNetCar dataset (Umetani & Bickel, 2018) consists of 889 different car shapes of the class "car" from the ShapeNet dataset (Chang et al., 2015). For each car body, details such as the side mirrors, spoilers, and tires are manually removed. To obtain the surface pressure and velocity field for each car shape, the Navier-Stokes equations are solved on a fine spatial grid over multiple timesteps. Each run models 10 seconds of air flow with a speed of 20 m/s ($Re = 5 \times 10^6$), with the final 4 seconds averaged to generate the time-averaged solution. The total computational time per simulation is approximately 50 minutes. The final data has 3,682 cells for the car geometry and surface pressure, as well as 29,498 points in the simulation volume. We use the first 100 samples for testing and the remaining ones for training. Figure 9 shows a random sample from the dataset with the geometry and simulated surface pressure and velocity.

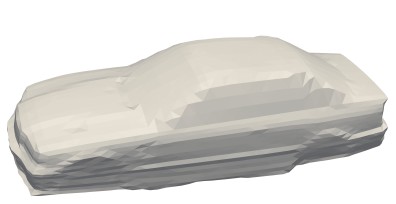
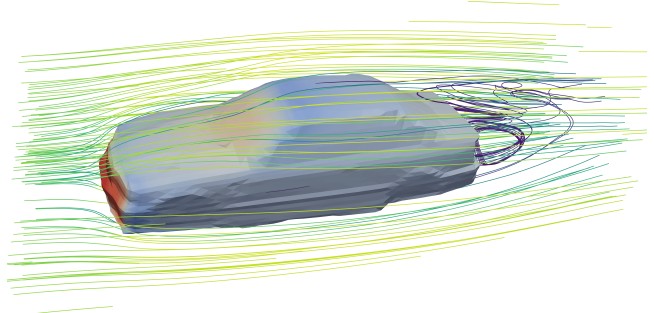

*(a)* CAD ShapeNetCar Geometry  *(b)* Simulated Pressure and Velocity Field

*Figure 9.* First sample from the ShapeNetCar dataset consisting of (a) the CAD geometry and (b) the physical simulation with the pressure field on the surface and velocity field in the volume.

### F.2. AhmedML

Ahmed bodies are simplified geometries that capture many of the characteristic flow topologies observed in road vehicles, such as cars (Ahmed et al., 1984). For this reason, they are widely employed as benchmark test cases for numerical solvers. The AhmedML dataset (Ashton et al., 2024) comprises 500 distinct Ahmed body variations along with simulated airflow data, including pressure and velocity fields, serving as reference cases for neural surrogates. Each configuration is simulated using a high-fidelity, time-resolved Detached-Eddy Simulation (DES), a hybrid turbulence modeling approach that combines Reynolds-Averaged Navier-Stokes (RANS) and Large-Eddy Simulation (LES). Meshes are generated with OpenFOAM's snappyHexMesh, requiring roughly 30 minutes per geometry, while the full simulation time amounted to approximately 48 hours per case. To reduce storage requirements and align with engineering practice, the results are time-averaged. On average, each sample has 101,405 cells for the geometry, 1,069,004 cells on the surface of the geometry, and 21,195,967 points in the surrounding volume. We generate a random train and test split and use 100 samples for testing and 400 samples for training. Figure 10 shows the first sample from the dataset with the geometry and simulated flow field.

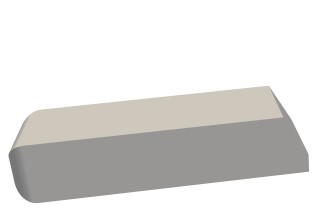
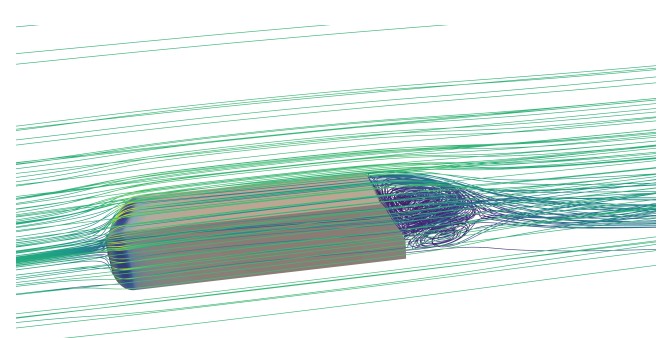

*(a)* CAD Ahmed Geometry  *(b)* Simulated Pressure and Velocity Field

*Figure 10.* First sample from the AhmedML dataset consisting of (a) the CAD geometry and (b) the physical simulation with the pressure field on the surface and velocity field in the volume.

### F.3. SHIFT-SUV

The SHIFT-SUV dataset (Luminary Cloud, 2025a) provides high-fidelity car geometries and corresponding aerodynamic simulations. It comprises hundreds of parametrically morphed variants of the AeroSUV platform, developed by FKFS (Forschungsinstitut für Kraftfahrwesen und Fahrzeugmotoren Stuttgart). The dataset includes two vehicle types (estate and fastback) and two scales (full and quarter). The quarter-scale version, obtained by uniformly scaling all geometries by a factor of 0.25, was employed by FKFS for wind tunnel validation. The simulations are performed with the Luminary

Cloud platform solver, a GPU-native solver utilizing a transient, scale-resolving Detached-Eddy Simulation (DES). For the full-scale datasets, a uniform inflow velocity of 30 m/s was applied. The time-resolved simulations were subsequently averaged to produce time-averaged fields. In our experiments, we restrict the experiments to the estate, full-scale subset, which yields 998 samples in total. On average, each geometry consists of 2,548,037 cells, with 2,521,717 cells on the surface simulation mesh and 50,507,366 points in the surrounding volume. A random split was performed, with 798 samples for training and 200 for testing. Figure 11 shows the first sample from the dataset with the geometry and simulated surface pressure and velocity.

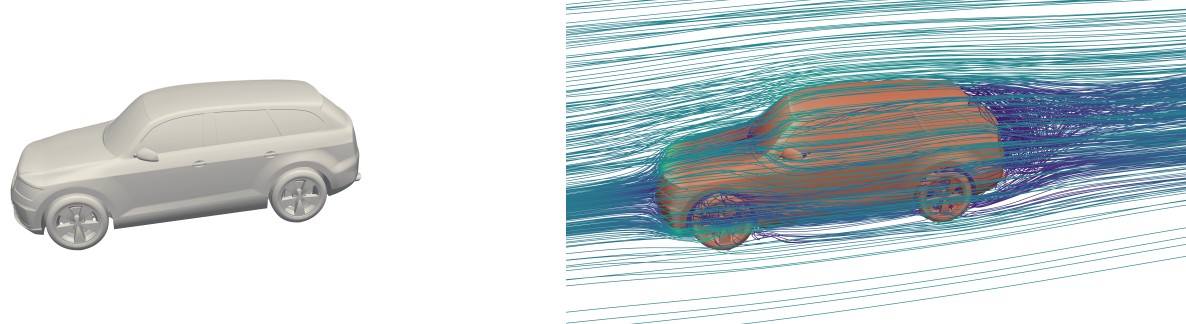

*(a)* CAD SUV Geometry                 *(b)* Simulated Pressure and Velocity Field

*Figure 11.* First sample from the SHIFT-SUV dataset consisting of (a) the CAD geometry and (b) the physical simulation with the pressure field on the surface and velocity field in the volume.

## F.4. SHIFT-Wing

The SHIFT-Wing dataset (Luminary Cloud, 2025b) comprises high-fidelity aerodynamic simulations of airplanes, derived from hundreds of parametric variants of the NASA Common Research Model (CRM). The parametric CRM framework defines both fuselage characteristics (e.g., tube diameter and length) and wing properties (e.g., position and rotation of the wing profile). In addition to geometric variations, the dataset spans different flow speeds (Mach numbers) and angles of attack. As with the SHIFT-SUV dataset, all cases are simulated using the Luminary Cloud platform solver. Distinctively, the SHIFT-Wing dataset employs Luminary Mesh Adaptation, which adaptively refines the mesh to resolve thin and sharp features such as transonic shocks without manual intervention. In total, the dataset contains 1,698 geometries, each simulated across multiple flow conditions. To incorporate these parameters, all models are conditioned on the Mach number and angle of attack. For our experiments, we restrict the dataset to the first 1000 samples to reduce storage requirements. On average, each sample consists of 1,623,086 geometry cells, 3,246,168 cells on the surface simulation mesh, and 5,970,264 volume points. A random split was applied, with 800 samples used for training and 200 for testing. Figure 12 shows the first sample from the dataset with the geometry and simulated flow field.

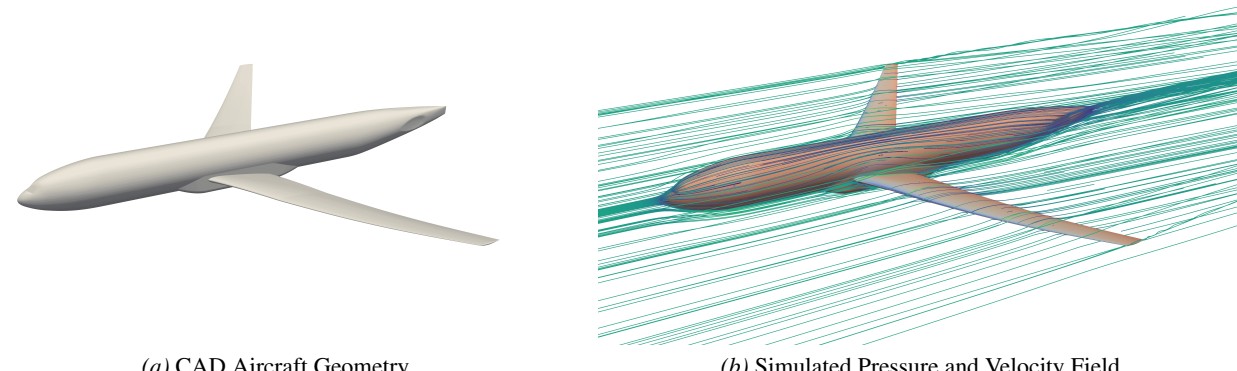

*(a)* CAD Aircraft Geometry                                      *(b)* Simulated Pressure and Velocity Field

*Figure 12.* First sample from the SHIFT-Wing dataset consisting of (a) the CAD geometry and (b) the physical simulation with the pressure field on the surface and velocity field in the volume.

## G. Additional Details on the Derived Datasets

For the final two experiments, we construct two datasets derived from the SHIFT-SUV dataset. In particular, we derive the SHIFT-SUV Uniform dataset, which replaces the CFD-optimized simulation meshes with uniformly resampled meshes that are not tailored to the numerical solver. The second is the SHIFT-SUV Rear dataset, which restricts the pressure and velocity fields to the rear of the car. Both datasets aim to evaluate how models perform under a distribution shift of the spatial query points.

### G.1. SHIFT-SUV Uniform

To evaluate neural surrogates, it is usually assumed that the simulation mesh for the test set is already available. Since the numerical solver is used to generate the ground-truth fields of the test set, the simulation mesh on the surface and volume must be generated beforehand, and is naturally available. In practical applications, however, the situation is different. When the model is expected to predict pressure and velocity fields for new geometries, no simulation mesh exists beforehand. Creating such a mesh is computationally expensive and time-consuming. A desirable surrogate model should therefore not depend on a simulation mesh and should support arbitrary spatial queries. This makes it possible to query the model with the CAD geometry mesh as the surface mesh and a regular grid as the volume mesh.

**Resampling Simulation Mesh.**   The SHIFT-SUV Uniform dataset is designed to evaluate this setting. We test the models on a mesh that consists of (i) the CAD-derived surface cells as the surface mesh and (ii) a regular grid in the volume. To obtain ground-truth fields for this configuration, we relax the problem and construct an approximately uniform volume mesh. For each point $p_{uni}$ on the uniform mesh, we obtain the nearest point $p_{nearest}$ from the original simulation mesh and replace $p_{uni}$ with the position and value of that nearest point $p_{nearest}$. This produces an approximately regular mesh while avoiding interpolation artifacts that could distort the physical fields. A similar nearest-cell sampling is applied to the CAD geometry mesh to obtain a surface mesh that approximates the CAD geometry mesh.

**Eliminating Solver-Induced Mesh Biases.**   Figure 13 compares the original CFD-optimized volume mesh and the uniformly resampled mesh. The optimized mesh becomes increasingly fine near the boundary of the car and forms a shell-like structure that follows the car body. The uniformly resampled mesh does not contain such adaptive refinement. As a result, the uniform mesh removes biases of the CFD-optimized simulation meshes, such as the exact position of the car or the correlation between the mesh density, high gradients, and salient flow patterns. On average, each sample of the SHIFT-SUV Uniform dataset contains approximately 600k surface points and 2M volume points.

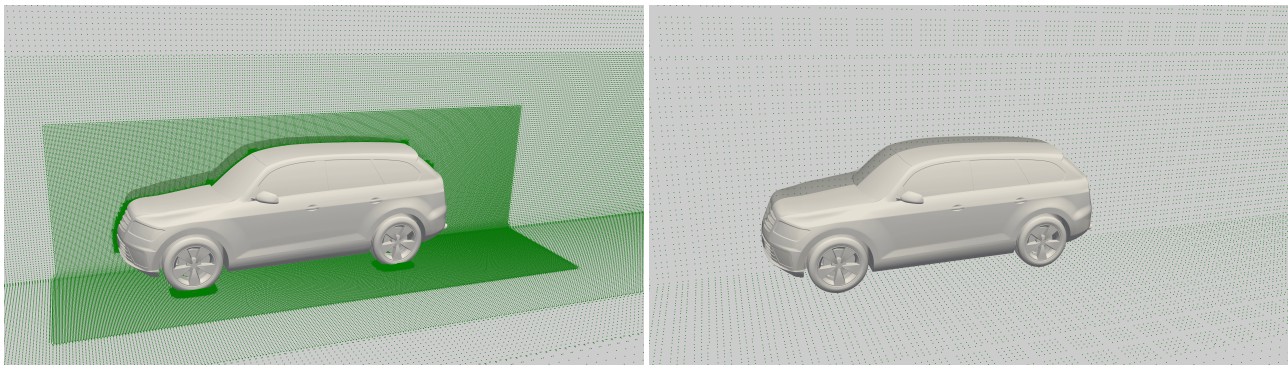

*(a)* Original CFD Volume Mesh     *(b)* Uniformly Resampled Volume Mesh

*Figure 13.* A sample from (a) the original SHIFT-SUV dataset with a CFD-optimized volume mesh and (b) the corresponding sample in the SHIFT-SUV Uniform dataset with a more uniform mesh. The original CFD mesh contains fine details close to the car boundary and looks like a shell, which is not the case for the uniformly resampled mesh.

### G.2. SHIFT-SUV Rear

Another practical scenario arises when engineers are only interested in the aerodynamics of specific spatial regions of the simulation domain. For instance, a car designer may be interested only in the flow field in the rear of the vehicle (the wake region) when evaluating different spoiler designs. To reflect this use case, we construct a dataset that restricts the pressure and velocity fields to the rear region of the car. Importantly, only the target pressure and velocity field are spatially restricted, while the geometry itself remains unchanged. Figure 14 illustrates one sample from the dataset. On average, each sample has approximately 1.5M points on the surface and 1.5M points in the volume.

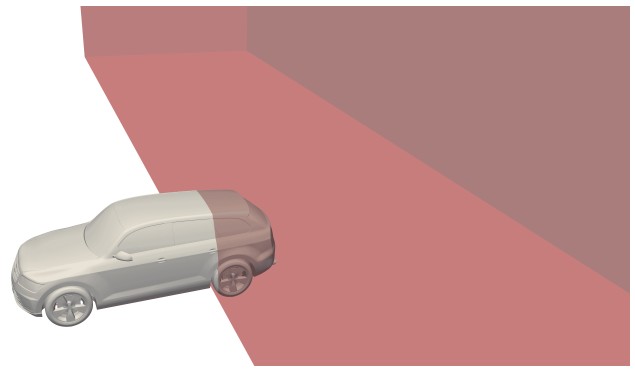

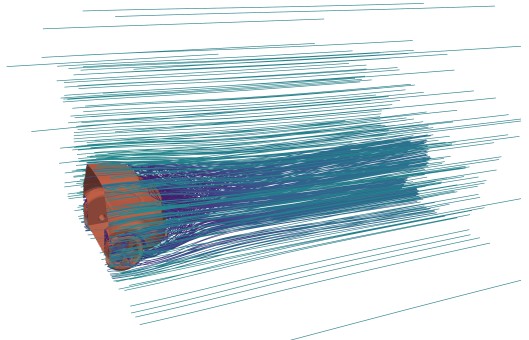

*(a)* CAD SUV Geometry and Rear Region (Red Box)     *(b)* Simulated Pressure and Velocity Field

*Figure 14.* A sample from the derived dataset consisting of (a) the CAD geometry and (b) the spatially restricted physical simulation with the pressure field on the surface and velocity field in the volume.

## H. Additional Details on the Baselines

**Geometry-informed Neural Operator (GINO).**    The Geometry-informed Neural Operator (GINO; Li et al. (2023b)) is a surrogate model for 3D simulations that integrates the Graph Neural Operator (GNO; Li et al. (2020)) with the Fourier Neural Operator (FNO; Li et al. (2021)). GINO takes a geometry, represented as a point cloud, as input and predicts physical quantities at arbitrary query locations within the simulation domain. Specifically, a GNO layer maps the input point cloud onto a predefined latent uniform mesh. To enrich this representation, each latent mesh point also incorporates the signed distance to the geometry boundary as an additional feature, providing spatial context. The resulting latent mesh is then processed by FNO layers, which leverage spectral convolutions to efficiently capture global interactions on the structured grid. Finally, another GNO layer projects the latent mesh back to physical quantities at the desired query positions. Additional simulation-specific parameters, such as angle of attack, are included into the model with adaptive instance

normalization as implemented in the NeuralOperator library (Kossaifi et al., 2025).

**Operator Transformer (OFormer).** The Operator Transformer (OFormer; Li et al. (2023c)) is designed to solve both time-dependent and time-independent PDEs. It leverages linear self-attention to encode the initial condition and geometry into a latent representation, enabling the model to capture spatial dependencies effectively. A cross-attention-based decoder then maps arbitrary query points to the corresponding quantities of the learned physical field using this latent representation. For time-dependent problems, the physical field is evolved iteratively through a pointwise MLP. In our experiments, OFormer is employed to predict physical fields conditioned on varying geometries. Specifically, the encoder processes the input geometry point cloud, while the decoder outputs the physical quantities at queried positions given the geometry and coordinates. We implement OFormer for 3D problems since the authors do not provide an implementation of the model for 3D. To reduce computational overhead, we subsample the geometry during training and inference. Additionally, we encode the coordinates for the encoder and decoder with the positional encoding from Transformers (Vaswani et al., 2017), which reduces the errors compared to using a linear projection and random Fourier features (Tancik et al., 2020) for coordinate encoding as proposed by the authors. We include additional simulation-specific parameters into the model with adaptive Layer Normalization with zero initialization (adaLN-Zero; Peebles & Xie (2023)) similar to the mechanism introduced in AB-UPT (Alkin et al., 2025b).

**General Neural Operator Transformer (GNOT).** The General Neural Operator Transformer (GNOT; Hao et al. (2023)) is a Transformer-based neural operator framework designed for a wide range of simulation tasks. It is capable of modeling both time-dependent and time-independent problems, handling complex geometries, and solving parametric PDEs. Additionally, the model supports arbitrary query coordinates. GNOT operates on a set of query coordinates, which serve as flexible probes into the simulation domain. These coordinates query the initial conditions, geometric information, and PDE parameters through heterogeneous normalized cross-attention, allowing the model to integrate diverse sources of context. Following this, a geometric gating mechanism, based on a mixture-of-experts (MoE), combines multiple pointwise MLP outputs weighted by spatial position, effectively processing pointwise information and serving as a soft decomposition of the domain. Normalized self-attention across the query set enables GNOT to capture spatial dependencies and correlations between different query points. Another geometric gating mechanism refines the representation further. By stacking multiple blocks of cross-attention, MoE-based gating, MLPs, and self-attention, GNOT builds a deep architecture that unifies operator learning across simulation types and domains, offering both flexibility and generalization. The original implementation uses an MLP to encode the coordinates, which we replace with positional encoding, which improves the error of the model.

**Transolver.** Transolver (Wu et al., 2024) is a Transformer-based model for solving time-dependent and time-independent PDEs on general geometries that applies attention over the spatial domain. To mitigate the quadratic costs of self-attention, the authors propose physics-attention, which removes the quadratic complexity while preserving the important property of learning dependencies between all tokens. In particular, physics-attention first maps the input tokens to a fixed number of slice tokens (i.e., computes cross-attention between the input tokens and the slice tokens). Subsequently, self-attention between the slice tokens is computed. After that, the slice tokens are mapped back to the input tokens with cross-attention. For time-dependent problems, Transolver takes the initial condition as input and applies physics-attention, followed by a pointwise MLP, on the initial condition. After several blocks with physics-attention and the pointwise MLP, the model outputs the solution for the subsequent timestep. Applying the model in an autoregressive fashion allows predicting multiple timesteps. For time-independent problems with varying geometries, Transolver uses the geometry as well as the query positions from the surface and volume mesh as input. Physics-attention on these allows the model to learn local and global patterns with reduced computational costs. After several blocks with physics-attention and pointwise MLPs, the model outputs the physical quantities for the query positions. The original Transolver model uses an MLP to encode the coordinates, which we replace with positional encoding, which reduces the errors. Additional parameters are encoded using adaLN-Zero (Peebles & Xie, 2023) in each block.

**Latent Neural Operator (LNO).** Wang & Wang (2024) propose the Latent Neural Operator (LNO) for solving time-dependent and time-independent problems. The model employs an encoder based on physics attention (cross-attention between input positions and latent positions) to encode the initial condition, which can contain information about the geometry, into a compressed and fixed latent representation. A Transformer-based processor computes the solution in that latent space, which is decoded by a physics-attention-based decoder that maps arbitrary queries to the solution for that position. For time-dependent problems, the model is applied in an autoregressive fashion. We use LNO to predict physical quantities from geometries as input. Similar to OFormer, the encoder maps the geometry to a latent representation, which

is used by the decoder to map arbitrary queries to the physical quantity at that position. In contrast to OFormer, LNO compresses the geometry into a latent representation with a fixed size, decoupling the geometry from the queries. We also encode the coordinates using positional encoding, which reduces the errors of the model compared to using MLPs for encoding. We adopt adaLN-Zero (Peebles & Xie, 2023) to add additional simulation parameters to the model.

**Geometry-Preserving Universal Physics Transformer (GP-UPT).**    The Geometry-Preserving Universal Physics Transformer (GP-UPT; Bleeker et al. (2025)) extends the Universal Physics Transformer (UPT; (Alkin et al., 2024)) for large-scale aerodynamic simulations. The model employs a geometry-aware encoder to encode the CFD-optimized surface mesh. The encoder follows the paradigm proposed for UPT, consisting of supernode pooling followed by a Perceiver (Jaegle et al., 2021). As a first step, a fixed set of supernodes is sampled from the surface mesh. These supernodes aggregate information from their local neighborhoods via message passing (Gilmer et al., 2017), producing latent supernode representations. A Perceiver then models spatial dependencies among these supernodes, producing a final representation for the CFD-optimized surface mesh. To support predictions at arbitrary spatial locations, a field decoder maps arbitrary query points on the surface and in the volume to physical quantities by attending to the latent supernode representations. This architecture enables direct supervision on the supernodes for surface predictions while maintaining flexibility through the decoder. To remove the dependency on the CFD-optimized surface mesh, the model is fine-tuned using the geometry point cloud instead of the simulation mesh. We implement the GP-UPT based on the description provided in their paper. However, we directly train GP-UPT using the geometry as input and do not utilize the supervision signal on the supernodes, as no ground truth is available for the supernodes sampled from the geometry point cloud. Furthermore, we add adaLN-Zero (Peebles & Xie, 2023) to each encoder and decoder block to condition the model on additional simulation-specific parameters. In the newest revision of the paper, the authors have replaced GP-UPT with AB-UPT.

**Mesh-Dependent Anchored-Branched Universal Physics Transformer (MD AB-UPT).**    In the first revision of the work, the authors proposed GP-UPT (Bleeker et al., 2025), which was later replaced by the Anchored-Branched Universal Physics Transformer (AB-UPT) in their latest revision (Alkin et al., 2025a). AB-UPT is specifically designed for industry-scale aerodynamic simulations. Like UPT and GP-UPT, it incorporates the message-passing-based supernode pooling and a Transformer for encoding. AB-UPT begins by encoding the input geometry through supernode pooling: a fixed number of points is sampled from the geometry, and each supernode aggregates information from its local neighborhood via message passing. The resulting supernode tokens are then refined using a Transformer. This stage mirrors GP-UPT, with the key difference that AB-UPT operates directly on the geometry rather than on the CFD-optimized surface mesh. A dedicated physics Transformer processes the encoded geometry. Its inputs consist of the supernode tokens (representing the geometry), a set of randomly sampled points from the surface and volume simulation meshes (anchors), and arbitrary surface and volume query coordinates. The physics Transformer is organized into two branches, one for surface predictions and one for volume predictions, and consists of physics and decoder blocks. Physics blocks support multiple configurations: (i) Cross-attention between the supernodes (as key and values in attention) and the anchors and query coordinates (as queries in attention) to inject geometric information into both branches; (ii) Self-attention among the set of anchors to capture spatial dependencies, followed by cross-attention between the anchors (as key and values in attention) and the query coordinates (as queries in attention) to enable scalability and query independence; and (iii) Cross-attention between the anchors and the opposite branch's anchors and queries to enable information exchange between the surface and volume branches. Several physics blocks with different configurations are applied, followed by separate decoder blocks for volume and surface predictions that compute cross-attention between the anchors and query coordinates. The model computes the physical quantities for the anchors and queries after several physics and decoder blocks. AB-UPT maintains query independence and supports scaling the queries to millions of points due to the anchor attention mechanism. However, the anchors are sampled from the simulation meshes, which means that AB-UPT still requires the simulation mesh as additional input. We follow Alkin et al. (2025b) and include additional simulation-specific parameters into AB-UPT with adaLN-Zero (Peebles & Xie, 2023).

**Mesh-Free AB-UPT (MF AB-UPT).**    The mesh-free AB-UPT uses the model architecture of AB-UPT but does not rely on the surface and volume simulation meshes as additional input. As proposed in Section "Training from CAD" in Alkin et al. (2025a), the surface anchors are sampled from the geometry mesh instead of the surface simulation mesh, and the volume anchors are sampled from a regular 3D grid, which makes AB-UPT a mesh-free model.

# I. Additional Details on the Experiments

The following section provides additional details on the hardware, loss function, evaluation metric, and hyperparameters used for the baseline and SMART models.

## I.1. Hardware

All experiments are carried out on an HPC cluster with NVIDIA A100 SXM4 GPUs with either 40GB or 80GB of memory. Training SMART requires between 3 and 16 hours on a single A100 GPU, depending on the dataset and the used precision.

## I.2. Loss Function

Following prior work (Li et al., 2021; 2023c;d; Chen & Wu, 2024; Hagnberger et al., 2026), we train the models using the relative L2 loss. This formulation ensures that channels with small magnitudes are weighted equally to those with large magnitudes (e.g., velocity in $z$-direction and velocity in $x$-direction), leading to balanced contributions across all channels. Formally, let $\boldsymbol{Y} \in \mathbb{R}^{M \times N_c}$ denote the ground truth, obtained with a numerical solver, and $\hat{\boldsymbol{Y}} \in \mathbb{R}^{M \times N_c}$ the model's prediction of $N_c$ physical quantities at $M$ positions in the spatial domain. Then, the relative L2 loss for a single sample is defined as

$$L(\hat{\boldsymbol{Y}}, \boldsymbol{Y}) = \frac{1}{N_c} \sum_{c=1}^{N_c} \frac{\left\| \hat{\boldsymbol{Y}}_{\cdot,c} - \boldsymbol{Y}_{\cdot,c} \right\|}{\left\| \boldsymbol{Y}_{\cdot,c} \right\|}, \tag{10}$$

where $\|\cdot\|$ denotes the $\mathcal{L}_2$ norm. For training, we compute a relative L2 loss for both surface and volume physical quantities and minimize the sum of these losses. The surface loss includes only the pressure channel, and the volume consists of the $x, y, z$ components of the velocity.

## I.3. Evaluation Metric

For evaluation, we employ the relative L2 error as defined above. Beyond the benefits discussed in the previous section, this metric can also be interpreted as a percentage error, offering an intuitive measure of the prediction accuracy. We provide the L2 error for the physical quantities in the volume ($x, y, z$ components of the velocity) and on the surface (only pressure channel).

## I.4. Hyperparameters

In this section, we list the hyperparameters used for the baseline and SMART models to ensure reproducibility. The baseline models include GINO, OFormer, GNOT, Transovler, LNO, GP-UPT, and AB-UPT.

**GINO.** Table 6 lists the hyperparameters of GINO used in the experiments. We follow the hyperparameters provided in the NeuralOperator library (Kossaifi et al., 2025) and use a latent mesh of $64 \times 64 \times 64$. To better match the other models in terms of trainable parameters and computational costs, we reduce the FNO dimension from $d = 64$ to $d = 32$. All GINO models are trained in `float32`, as mixed-precision with `bfloat16` or `float16` resulted in numerical instabilities.

*Table 6.* Hyperparameters of GINO used in the experiments.

| Parameter | Dataset | | | |
|---|---|---|---|---|
| | ShapeNetCar | AhmedML | SHIFT-SUV | SHIFT-Wing |
| In GNO Dimensions | [80, 80, 80] | | | |
| Out GNO Dimensions | [512, 256] | | | |
| GNO Dimension | 32 | | | |
| GNO Radius | 0.033 | | | |
| Latent Mesh | $64 \times 64 \times 64$ | | | |
| FNO Dimension | 32 | | | |
| FNO Modes | 16 | | | |
| # FNO Layers | 4 | | | |
| Learning Rate | 1e-3 | | | |
| Batch Size | 1 | | | |
| Epochs | 200 | | | |
| Optimizer | AdamW | | | |
| Gradient Norm Clipping | ✗ | ✗ | ✗ | ✗ |
| Precision | float32 | | | |
| Parameters | 19,270,759 | 19,270,759 | 19,270,759 | 19,545,703 |

**OFormer.** Table 7 outlines the OFormer hyperparameters used in the experiments. Since the authors did not test OFormer on the datasets considered in this work or on comparable ones, we obtain the hyperparameters through a hyperparameter search. We use a hidden dimension of $d = 256$ for all experiments, and employ 4 and 6 cross-attention layers in the encoder for ShapeNetCar and the remaining datasets, respectively, which have proven to be effective. Additionally, we train the automotive models with mixed-precision and `bfloat16` and the aerospace model with `float32`.

*Table 7.* Hyperparameters of OFormer used in the experiments.

| Parameter | Dataset | | | |
|---|---|---|---|---|
| | ShapeNetCar | AhmedML | SHIFT-SUV | SHIFT-Wing |
| Dimension | 256 | | | |
| # Encoder Cross-Attention Layers | 4 | 6 | 6 | 6 |
| Decoder MLP Dimension | 512 | | | |
| Learning Rate | 1e-5 | | | |
| Batch Size | 1 | | | |
| Epochs | 200 | | | |
| Optimizer | LION | | | |
| Gradient Norm Clipping | ✗ | ✗ | ✗ | 2.0 |
| Precision | bfloat16 | bfloat16 | bfloat16 | float32 |
| Parameters | 5,847,556 | 7,293,444 | 7,293,444 | 10,062,084 |

**GNOT.** Table 8 presents the hyperparameters of GNOT used in the experiments. As the original work does not report results for GNOT on the benchmarks used in this work or on comparable benchmarks, we determine suitable hyperparameters through a hyperparameter search. For ShapeNetCar, we use a hidden dimension of $d = 128$ and employ 2 GNOT blocks. The AhmedML and SHIFT-SUV datasets use a larger configuration with $d = 256$ and 4 blocks, and the model is further scaled to 6 blocks for SHIFT-Wing. These configurations have shown strong empirical performance across the respective benchmarks. Additionally, we employ mixed-precision training with `bfloat16` for the automotive models, while the aerospace model is trained in `float32`.

*Table 8.* Hyperparameters of GNOT used in the experiments.

| Parameter | Dataset | | | |
|---|---|---|---|---|
| | ShapeNetCar | AhmedML | SHIFT-SUV | SHIFT-Wing |
| Dimension | 128 | 192 | 192 | 192 |
| # Blocks | 2 | 4 | 4 | 6 |
| # Heads | | | 1 | |
| # MLP Layers | | | 2 | |
| # Experts | | | 2 | |
| Learning Rate | 1e-5 | 1e-4 | 1e-4 | 1e-4 |
| Batch Size | 1 | 2 | 2 | 2 |
| Epochs | | | 200 | |
| Optimizer | | | LION | |
| Gradient Norm Clipping | ✗ | ✗ | ✗ | 2.0 |
| Precision | `bfloat16` | `bfloat16` | `bfloat16` | `float32` |
| Parameters | 2,034,056 | 8,716,620 | 8,716,620 | 13,429,456 |

**Transolver.** Table 9 summarizes the hyperparameters of Transolver used in the experiments. We adopt the hyperparameters provided by the authors for the ShapeNetCar dataset (Wu et al., 2024), but increase the MLP ratio from 2 to 4. Moreover, we increase the number of slice tokens to 128 for the AhmedML, SHIFT-SUV, and SHIFT-Wing datasets to accommodate the larger point clouds, which has proven to be effective. We employ mixed-precision training with `bfloat16` for the automotive models, whereas the aerospace model is trained using `float32` to avoid training instabilities.

*Table 9.* Hyperparameters of Transolver used in the experiments.

| Parameter | Dataset | | | |
|---|---|---|---|---|
| | ShapeNetCar | AhmedML | SHIFT-SUV | SHIFT-Wing |
| Dimension | | | 256 | |
| # Layers | | | 8 | |
| # Slice Tokens | 32 | 128 | 128 | 128 |
| MLP Ratio | | | 4 | |
| # Heads | | | 8 | |
| Learning Rate | 1e-3 | 1e-3 | 1e-3 | 5e-4 |
| Batch Size | | | 1 | |
| Epochs | | | 200 | |
| Optimizer | | | AdamW | |
| Gradient Norm Clipping | ✗ | ✗ | ✗ | ✗ |
| Precision | `bfloat16` | `bfloat16` | `bfloat16` | `float32` |
| Parameters | 5,851,972 | 5,851,972 | 5,851,972 | 7,968,580 |

**LNO.** Table 10 provides the LNO hyperparameters used in the experiments. Since the original work did not cover the benchmark cases considered here, we identify suitable settings through a hyperparameter search. Across all datasets, we use a hidden dimension of $d = 256$ and 512 modes. Compared to the configuration used for ShapeNetCar, we increase both the number of projector layers and processor blocks for the remaining datasets. The training is performed in mixed precision using `float16` for ShapeNetCar and `bfloat16` for AhmedML and SHIFT-SUV. For SHIFT-Wing, we train the model in `float32` to prevent numerical instabilities.

*Table 10.* Hyperparameters for LNO used in the experiments.

| Parameter | Dataset | | | |
|---|---|---|---|---|
| | ShapeNetCar | AhmedML | SHIFT-SUV | SHIFT-Wing |
| Dimension | | | 256 | |
| Modes | | | 512 | |
| # Projector Layers | 4 | 6 | 6 | 6 |
| # Processor Blocks | 4 | 8 | 8 | 8 |
| # Heads | | | 4 | |
| Learning Rate | 1e-5 | 5e-5 | 5e-5 | 1e-5 |
| Batch Size | | | 1 | |
| Epochs | | | 200 | |
| Optimizer | | | LION | |
| Gradient Norm Clipping | ✗ | ✗ | ✗ | 2.0 |
| Precision | `float16` | `bfloat16` | `bfloat16` | `float32` |
| Parameters | 4,148,996 | 6,915,332 | 6,915,332 | 10,079,492 |

**GP-UPT.** The hyperparameters of GP-UPT are depicted in Table 11. The authors provide hyperparameters for the ShapeNetCar and DrivAerML (Ashton et al., 2025) datasets in Bleeker et al. (2025). For the number of encoder Perceiver blocks and decoder cross-attention blocks, we adopt the number proposed by the authors. However, we reduce the hidden dimension to $d = 128$ for the ShapeNetCar dataset and use $d = 192$ for the Shift-SUV and Shift-Wing datasets, which have demonstrated strong empirical performance on their respective benchmarks. Additionally, we train the automotive models with mixed-precision and `float16` and the aerospace model with `float32`.

*Table 11.* Hyperparameters for GP-UPT used in the experiments.

| Parameter | Dataset | | | |
|---|---|---|---|---|
| | ShapeNetCar | AhmedML | SHIFT-SUV | SHIFT-Wing |
| Dimension | 128 | 192 | 192 | 192 |
| # Supernodes | 3,682 | 16,384 | 16,384 | 8,192 |
| Supernode Pooling Radius | | | 0.25 | |
| # Encoder Perceiver Blocks | | | 3 | |
| # Decoder Cross-Attention Blocks | 2 | 2 | 2 | 4 |
| Learning Rate | | | 5e-5 | |
| Batch Size | | | 1 | |
| Epochs | | | 200 | |
| Optimizer | | | LION | |
| Gradient Norm Clipping | 2.0 | 2.0 | 2.0 | ✗ |
| Precision | `float16` | `float16` | `float16` | `float32` |
| Parameters | 4,005,701 | 7,110,405 | 7,110,405 | 12,646,149 |

**AB-UPT.** We use the hyperparameter denoted in Table 12 for AB-UPT in the experiments. We adopt the hyperparameters provided by the authors in Alkin et al. (2025a). However, we halve the number of supernodes and anchors to reduce computational cost. In addition, we decrease the latent dimension for the smaller ShapeNetCar dataset to reduce overfitting, which has proven to be effective. The physics block order specifies how attention is applied. We follow the notation used in their implementation. P denotes cross-attention between the geometry and the volume and surface anchors and queries (Perceiver), S denotes cross-attention between the anchors and the anchors and queries within a branch (Split), and C denotes cross-attention between the anchors and the queries and anchors of the opposite branch (Cross). We train the automotive models with mixed-precision and `float16` and the aerospace model with `float32` following Alkin et al. (2025a;b).

*Table 12.* Hyperparameters of AB-UPT used in the experiments.

| Parameter | Dataset | | | |
|---|---|---|---|---|
| | ShapeNetCar | AhmedML | SHIFT-SUV | SHIFT-Wing |
| Dimension | 128 | 192 | 192 | 192 |
| # Surface Anchors | 1,841 | 8,192 | 8,192 | 4,096 |
| # Volume Anchors | 8,192 | 8,192 | 8,192 | 4,096 |
| # Supernodes | 3,682 | 16,384 | 16,384 | 8,192 |
| Supernode Pooling Radius | | 0.25 | | |
| # Geometry Blocks | | 1 | | |
| # Physics Blocks | | 6 | | |
| # Physics Blocks Order | | PSCSCS | | |
| # Volume Decoder Blocks | | 6 | | |
| # Surface Decoder Blocks | | 6 | | |
| Learning Rate | 1e-4 | 5e-5 | 1e-4 | 5e-5 |
| Batch Size | | 1 | | |
| Epochs | | 200 | | |
| Optimizer | | LION | | |
| Gradient Norm Clipping | ✗ | ✗ | ✗ | ✗ |
| Precision | `float16` | `float16` | `float16` | `float32` |
| Parameters | 3,899,908 | 8,749,828 | 8,749,828 | 12,762,244 |

**SMART.** Table 13 shows the hyperparameters of SMART used in the experiments. We use a latent dimension of $d = 128$ for the ShapeNetCar dataset and $d = 256$ for the AhmedML, SHIFT-SUV, and SHIFT-Wing datasets. For the automotive datasets, we employ 6 encoder and decoder blocks, while for the aerospace dataset, we increase this number to 8, a configuration that has proven to be effective. Automotive models are trained with mixed precision and `float16`, whereas the aerospace model is trained in `float32` due to numerical instabilities observed with mixed precision and `bfloat16` or `float16`.

*Table 13.* Hyperparameters of SMART used in the experiments.

| Parameter | Dataset | | | |
|---|---|---|---|---|
| | ShapeNetCar | AhmedML | SHIFT-SUV | SHIFT-Wing |
| Dimension $d$ | 128 | 256 | 256 | 256 |
| # Points of Latent Geometry $K$ | 3,682 | 2,048 | 2,048 | 4,096 |
| # Points in Sampled Geometry $\tilde{G}_{sub}$ | 3,682 | 16,384 | 8,192 | 4,096 |
| # Encoder Blocks | 6 | 6 | 6 | 8 |
| # Decoder Blocks | 6 | 6 | 6 | 8 |
| Shared Attention Layer and MLP | | ✓ | | |
| Learning Rate | 1e-4 | 1e-4 | 1e-4 | 5e-5 |
| Batch Size | 1 | 1 | 1 | 1 |
| Epochs | | 200 | | |
| Optimizer | | LION | | |
| Gradient Norm Clipping | ✗ | ✗ | ✗ | 2.0 |
| Precision | `float16` | `float16` | `float16` | `float32` |
| Parameters | 2,009,024 | 7,937,468 | 7,937,468 | 12,426,172 |

# J. Hyperparameter Description

We provide additional information on the key hyperparameters of SMART and clarify how they should be selected for different simulation scenarios.

**Latent Geometry Size $K$.** The size of the latent geometries $K$ is a task-dependent hyperparameter that potentially must be adapted to the geometric complexity, flow physics, and effective size of the simulation domain. For instance, in the SHIFT-Wing dataset, the domain is large and the wake exhibits long, coherent trailing-vortex structures. Capturing these persistent 3D flow features requires a larger number of latent tokens. In contrast, the SHIFT-SUV dataset represents a

bluff-body flow with a short, rapidly breaking turbulent wake, so fewer tokens are sufficient. For these reasons, K scales with the spatial extent and the underlying flow structures, not merely with the number of simulation mesh points. The following ablation study also investigates the effect of different $K$ on the model's performance. A small $K$ of 128 already yields errors smaller compared to most of the baselines, and increasing $K$ consistently decreases the errors. This indicates that SMART can capture the important geometric structures with relatively few latent tokens, while larger $K$ values provide additional capacity for more complex simulations.

## K. Ablation Study

In this section, we analyze how the model scales with respect to its latent space and investigate which components contribute most to the model's performance through an ablation study. In particular, we investigate the impact of (i) the modulated positional encoding, (ii) using a coarse geometry point cloud as initial input for the encoder instead of learnable tokens, (iii) geometry cross-attention within the encoder blocks, (iv) the cross-layer geometry-physics update that incorporates the encoder's intermediate representations into the decoder, and (v) the sampling strategy. All experiments are conducted on randomly subsampled SHIFT-SUV and SHIFT-Wing datasets with 16k points on the surface and volume. Each ablation study provides an updated architecture diagram to illustrate the changes made to the model.

### K.1. Size of Latent Space (Latent Geometries)

The size of the model's latent space is defined by the dimension $d$ (number of channels) and the number of points or tokens $K$ of the latent geometries. We investigate the effect of the latent space size by scaling the number of points $K$ of the latent geometries, starting from a model (a) with a small number of points and incrementally increasing it up to a model (f) with a large number of points. Table 14 shows that increasing the number of points consistently reduces the errors for both surface and volume predictions on the SHIFT-SUV dataset.

*Table 14.* Relative L2 errors of SMART models trained and tested on the **subsampled** SHIFT-SUV dataset with 16k points on the surface and volume. The values in parentheses indicate the percentage deviation to the best model, which uses the largest latent space with $K = 4096$ points.

| | Model Configuration | Relative L2 Error $\times 10^{-2}$ ($\downarrow$) | |
| --- | --- | --- | --- |
| | # Points of Latent Geometry $K$ | SHIFT-SUV | |
| | | Surface | Volume |
| (a) | 128 | $0.0181^{\pm 0.0000}$ (+4%) | $6.6090^{\pm 0.0331}$ (+6%) |
| (b) | 256 | $0.0180^{\pm 0.0000}$ (+3%) | $6.5126^{\pm 0.0152}$ (+5%) |
| (c) | 512 | $0.0178^{\pm 0.0001}$ (+2%) | $6.4245^{\pm 0.0311}$ (+3%) |
| (d) | 1024 | $0.0176^{\pm 0.0001}$ (+1%) | $6.2957^{\pm 0.0297}$ (+1%) |
| (e) | 2048 | $\underline{0.0175}^{\pm 0.0001}$ (+1%) | $\underline{6.2627}^{\pm 0.0265}$ (+1%) |
| (f) | 4096 | $\mathbf{0.0174}^{\pm 0.0001}$ | $\mathbf{6.2076}^{\pm 0.0215}$ |

### K.2. Modulated Positional Encoding

Next, we conduct an ablation study to investigate the effect of the modulation positional encoding for embedding the 3D spatial coordinates of the geometry point cloud and query points. We compare a model (a) that uses modulated positional encoding against a model (b) with standard Transformer positional encoding. Figure 15 shows the updated model (b) with the standard Transformer positional encoding instead of modulated positional encoding. Table 15 shows that the modulated positional encoding slightly contributes to the model performance across several datasets, including the SHIFT-SUV and SHIFT-Wing datasets.

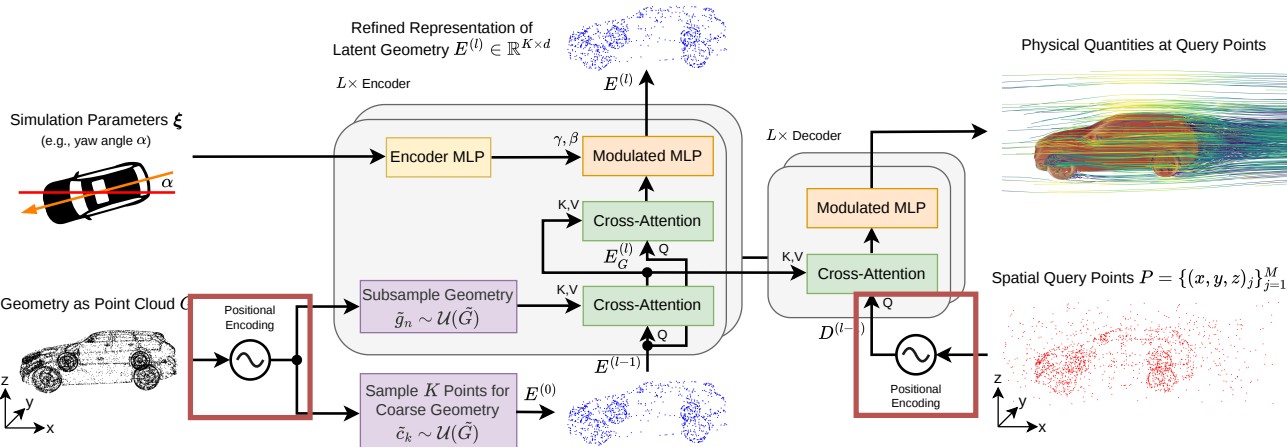

*Figure 15.* The updated model uses the standard Transformer positional encoding instead of modulated positional encoding. The red rectangle highlights the changes made to the proposed model.

*Table 15.* Relative L2 errors of SMART models trained and tested on **subsampled** SHIFT-SUV and SHIFT-Wing datasets with 16k points on the surface and volume. The values in parentheses indicate the percentage deviation to the model that uses modulated positional encoding.

| | Model Configuration | Relative L2 Error $\times 10^{-2}$ ($\downarrow$) | | | |
|---|---|---|---|---|---|
| | Modulated PE | SHIFT-SUV | | SHIFT-Wing | |
| | | Surface | Volume | Surface | Volume |
| (a) | ✓ | $\mathbf{0.0175}^{\pm 0.0001}$ | $\mathbf{6.2627}^{\pm 0.0265}$ | $\mathbf{0.0557}^{\pm 0.0004}$ | $\mathbf{14.6639}^{\pm 0.0315}$ |
| (b) | ✗ | $0.0178^{\pm 0.0000}$ (+2%) | $6.4383^{\pm 0.0293}$ (+3%) | $0.0570^{\pm 0.0004}$ (+2%) | $14.9724^{\pm 0.0071}$ (+2%) |

## K.3. Coarse Geometry as Initial Input for the Encoder

Recent latent neural surrogate models, such as AROMA (Serrano et al., 2024) and LNO (Wang & Wang, 2024), compress the input into a fixed latent space using learnable query tokens combined with cross-attention. In contrast, SMART uses randomly sampled points from the input geometry (i.e., a coarse geometry) as input for the encoder, providing an initial geometric scaffold which will be refined using cross-attention. In this experiment, we compare a model (a) that uses a coarse geometry as input for the encoder against a model (b) that uses learnable query tokens. Figure 16 shows the model (b) with the learnable tokens as initial input to the encoder. The experiment is conducted on the SHIFT-Wing dataset. As shown in Table 16, using a coarse geometry as initial encoder input substantially reduces prediction errors, as it supplies additional geometric information that learnable queries alone cannot capture.

*Table 16.* Relative L2 errors of SMART models trained and tested on the **subsampled** SHIFT-Wing dataset with 16k points on the surface and volume. The values in parentheses indicate the percentage deviation to the model that uses a coarse geometry as initial encoder input.

| | Model Configuration | Relative L2 Error $\times 10^{-2}$ ($\downarrow$) | |
|---|---|---|---|
| | Coarse Geometry as Initial Input | SHIFT-Wing | |
| | | Surface | Volume |
| (a) | ✓ | $\mathbf{0.0557}^{\pm 0.0004}$ | $\mathbf{14.6639}^{\pm 0.0315}$ |
| (b) | ✗ | $0.1564^{\pm 0.0048}$ (+181%) | $36.4043^{\pm 0.1814}$ (+148%) |

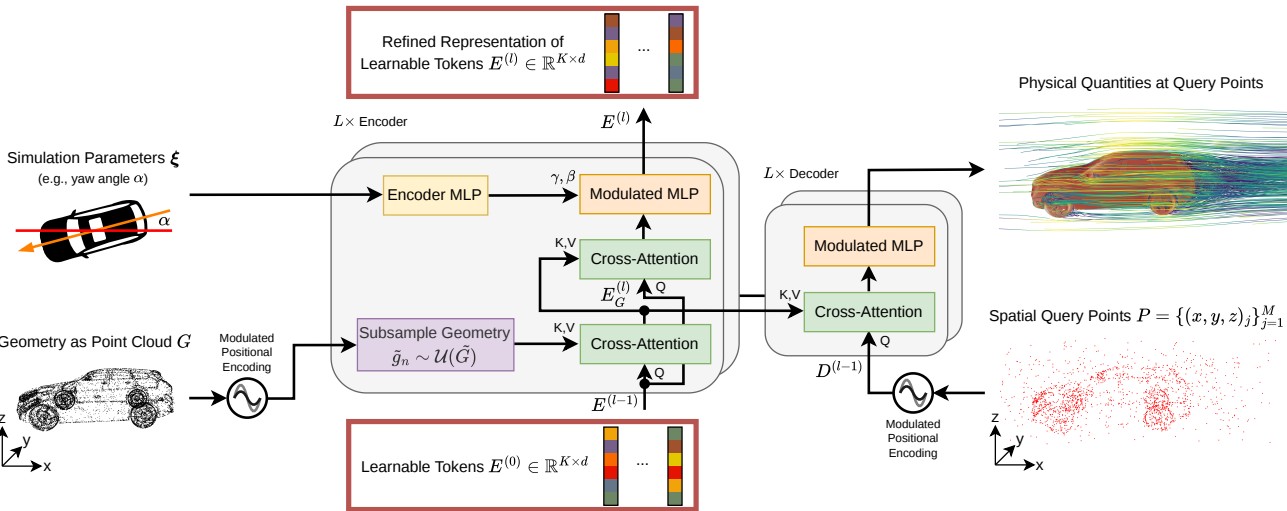

*Figure 16.* The encoder uses learnable query tokens instead of a coarse geometry as initial input (red rectangle). These tokens are subsequently refined using geometric features extracted from the input geometry via cross-attention.

## K.4. Geometry Cross-Attention within the Encoder Blocks

The encoder blocks of SMART employ geometry cross-attention, enabling the latent geometric representation (i.e., the latent geometries) to attend to randomly subsampled points from the input geometry point cloud to refine their features. In this experiment, we compare a model (a) that includes geometry cross-attention within the encoder with a model (b) that omits geometry cross-attention within the encoder. In contrast to the model configuration used for the main experiments, we decrease the size of the latent geometries to $K = 256$ points. Figure 17 illustrates model (b) that omits geometry cross-attention in the encoder. As shown in Table 17, geometry cross-attention reduces prediction errors by enriching the latent geometries with additional geometric information from the input geometry.

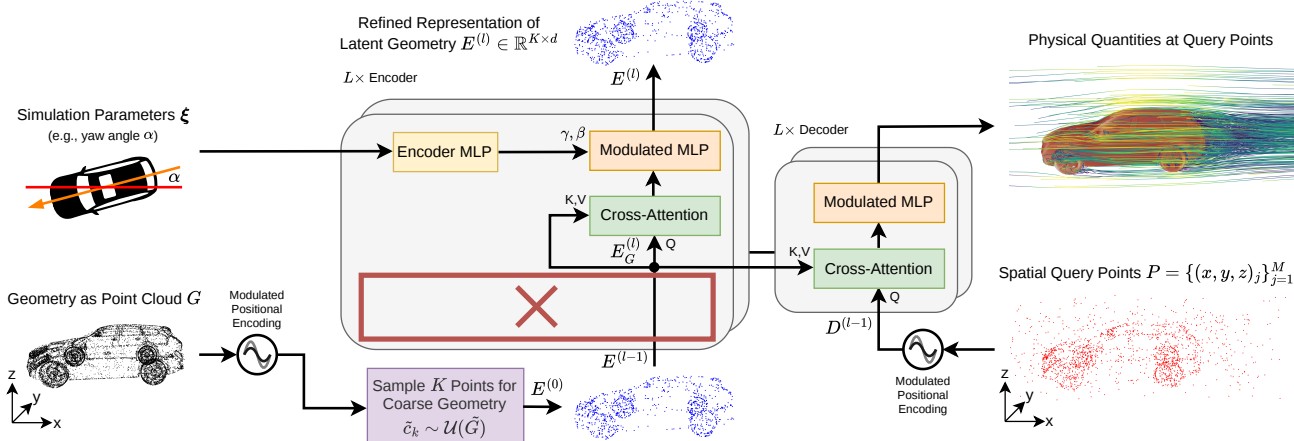

*Figure 17.* The encoder does not utilize geometry cross-attention between the latent geometries and randomly sampled points from the point cloud, illustrated by the missing geometry cross-attention component in the red rectangle.

*Table 17.* Relative L2 errors of SMART models trained and tested on the **subsampled** SHIFT-SUV dataset with 16k points on the surface and volume. The values in parentheses indicate the percentage deviation to the model that uses geometry cross-attention within the encoder blocks.

| Model Configuration | Relative L2 Error $\times 10^{-2}$ ($\downarrow$) | | |
| --- | --- | --- | --- |
| Geometry Cross-Attention | | SHIFT-SUV | |
| | | Surface | Volume |
| (a) | ✓ | $\mathbf{0.0180}^{\pm 0.0000}$ | $\mathbf{6.5126}^{\pm 0.0152}$ |
| (b) | ✗ | $0.0188^{\pm 0.0001}$ (+4%) | $7.1569^{\pm 0.0222}$ (+10%) |

## K.5. Cross-layer Geometry-Physics Update

The decoder of SMART does not rely solely on the final encoder output but instead attends to the intermediate latent geometries produced at each encoder block. This cross-layer interaction enables joint refinement of the geometric and physical field representations. To evaluate the importance of this mechanism, we compare a model that (a) employs cross-layer geometry-physics update by incorporating the encoder's intermediate representations into the decoder with a model (b) that uses only the final latent representation of the encoder as input to the decoder. The experiment is conducted on the SHIFT-Wing dataset. Figure 18 illustrates the modified model (b) that uses only the final representations of the encoder instead of the intermediate representations. Table 18 shows that the model that leverages intermediate representations performs significantly better, confirming the importance of cross-layer geometry-physics update for accurate simulations.

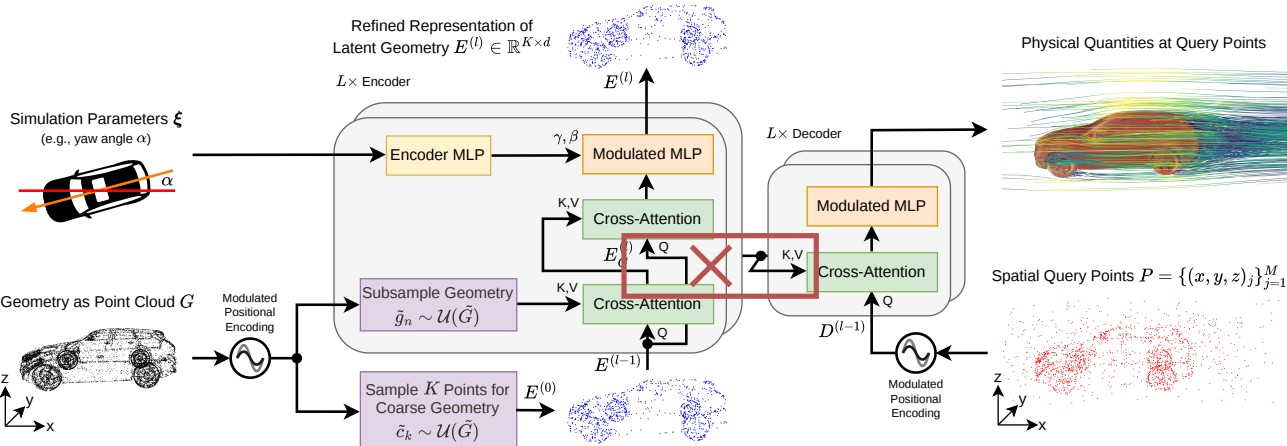

*Figure 18.* The decoder uses only the final encoder representation as input, omitting all intermediate encoder features. The red rectangle highlights that the first decoder block also receives only the final encoder output and that the connection from the first encoder block to the first decoder block is removed.

*Table 18.* Relative L2 errors of SMART models trained and tested on the **subsampled** SHIFT-Wing dataset with 16k points on the surface and volume. The values in parentheses indicate the percentage deviation to the model that uses the cross-layer geometry-physics update by attending to the intermediate representations of the encoder.

| Model Configuration | Relative L2 Error $\times 10^{-2}$ ($\downarrow$) | | |
| --- | --- | --- | --- |
| Cross-layer Geometry-Physics Update | | SHIFT-Wing | |
| | | Surface | Volume |
| (a) | ✓ | $\mathbf{0.0557}^{\pm 0.0004}$ | $\mathbf{14.6639}^{\pm 0.0315}$ |
| (b) | ✗ | $0.1337^{\pm 0.0049}$ (+140%) | $35.8533^{\pm 0.3180}$ (+145%) |

### K.6. Sampling Strategy

Finally, we examine the effect of the sampling strategy on model accuracy by replacing (a) uniform sampling with (b) farthest-point sampling (FPS) during training and inference. The results in Table 19 show only marginal deviations across all settings, indicating that SMART is robust to the choice of the sampling strategy. Given that uniform sampling is substantially cheaper to compute, especially for large point clouds, it remains the preferred option in practice.

*Table 19.* Relative L2 errors of SMART models trained and tested on the SHIFT-SUV dataset. The values in parentheses indicate the percentage deviation to the model that uses uniform sampling.

| Model Configuration | | Relative L2 Error $\times 10^{-2}$ ($\downarrow$) | | | |
|---|---|---|---|---|---|
| Sampling Strategy | | SHIFT-SUV | | | |
| | | Subsampled | | Full Resolution | |
| | | Surface | Volume | Surface | Volume |
| (a) | Uniform | 0.0175 | 6.2627 | **0.0175** | 6.2946 |
| (b) | FPS | **0.0174** (-0.57%) | **6.2449** (-0.28%) | **0.0175** (-0.00%) | **6.2922** (-0.04%) |

# L. Additional Results

We report detailed metrics for (RQ1) subsampled datasets, (RQ2) full-resolution datasets, (RQ3) models queried on meshes not optimized for CFD, and (RQ4) models queried on a subregion of the spatial domain. In addition, we provide drag and lift forces computed using the CFD simulation and the model's surface predictions. We also provide qualitative results for randomly selected samples from each dataset.

### L.1. Quantitative Results

#### L.1.1. ERRORS ON SUBSAMPLED DATASETS (RQ1)

Tables 20 and 21 report the mean errors and standard deviations across multiple runs for models trained and evaluated on randomly subsampled data consisting of 16k points on the surface and 16k points in the volume.

*Table 20.* Rel. L2 errors of models tested on **subsampled** datasets with 16k points on the surface and volume. Values in parentheses indicate the percentage deviation to SMART, and underlined values indicate the second-best errors.

| Model | Relative L2 Error $\times 10^{-2}$ ($\downarrow$) | | | |
|---|---|---|---|---|
| | ShapeNetCar | | AhmedML | |
| | Surface | Volume | Surface | Volume |
| GINO | $14.4208^{\pm 0.0224}$ (+113%) | $10.2862^{\pm 0.1063}$ (+105%) | $16.0332^{\pm 0.0818}$ (+414%) | $17.3267^{\pm 0.1174}$ (+236%) |
| OFormer | $9.1896^{\pm 0.0326}$ (+36%) | $7.3458^{\pm 0.1758}$ (+46%) | $5.0421^{\pm 0.1470}$ (+62%) | $7.3925^{\pm 0.1584}$ (+44%) |
| GNOT | $11.2701^{\pm 0.1839}$ (+66%) | $9.1364^{\pm 0.1551}$ (+82%) | $3.7068^{\pm 0.2048}$ (+19%) | $5.8419^{\pm 0.1519}$ (+13%) |
| Transolver | $8.4825^{\pm 0.2202}$ (+25%) | $7.1897^{\pm 0.2490}$ (+43%) | $3.4441^{\pm 0.2808}$ (+10%) | $5.4736^{\pm 0.2998}$ (+6%) |
| LNO | $12.5501^{\pm 0.5861}$ (+85%) | $10.4892^{\pm 0.2193}$ (+109%) | $5.1230^{\pm 0.1588}$ (+64%) | $10.2931^{\pm 0.4578}$ (+100%) |
| GP-UPT | $6.9672^{\pm 0.1113}$ (+3%) | $5.5198^{\pm 0.1416}$ (+10%) | $3.3879^{\pm 0.0877}$ (+9%) | $5.6393^{\pm 0.1420}$ (+9%) |
| Mesh-dependent AB-UPT | $6.9959^{\pm 0.2364}$ (+3%) | $\underline{5.2457}^{\pm 0.2260}$ (+4%) | $3.4478^{\pm 0.2083}$ (+11%) | $5.1812^{\pm 0.1634}$ (+1%) |
| Mesh-free AB-UPT | $\underline{6.9406}^{\pm 0.0455}$ (+2%) | $5.2709^{\pm 0.2320}$ (+5%) | $\underline{3.2166}^{\pm 0.0476}$ (+3%) | $\underline{5.1712}^{\pm 0.1161}$ (+0%) |
| SMART | $\textbf{6.7734}^{\pm 0.0577}$ | $\textbf{5.0215}^{\pm 0.0744}$ | $\textbf{3.1195}^{\pm 0.3393}$ | $\textbf{5.1511}^{\pm 0.3392}$ |

*Table 21.* Rel. L2 errors of models tested on **subsampled** datasets with 16k points on the surface and volume. Values in parentheses indicate the percentage deviation to SMART, and underlined values indicate the second-best errors.

| Model | Relative L2 Error $\times 10^{-2}$ ($\downarrow$) | | | |
|---|---|---|---|---|
| | SHIFT-SUV | | SHIFT-Wing | |
| | Surface | Volume | Surface | Volume |
| GINO | $0.1215^{\pm 0.0004}$ (+594%) | $51.5540^{\pm 0.2144}$ (+723%) | $4.3333^{\pm 0.0050}$ (+7680%) | $75.1867^{\pm 0.0181}$ (+413%) |
| OFormer | $0.0224^{\pm 0.0001}$ (+28%) | $10.9833^{\pm 0.0365}$ (+75%) | $0.2637^{\pm 0.0090}$ (+373%) | $43.2504^{\pm 0.2000}$ (+195%) |
| GNOT | $0.0188^{\pm 0.0001}$ (+7%) | $7.1901^{\pm 0.0715}$ (+15%) | $0.0749^{\pm 0.0015}$ (+34%) | $18.5032^{\pm 0.4848}$ (+26%) |
| Transolver | $0.0183^{\pm 0.0001}$ (+5%) | $6.8487^{\pm 0.0862}$ (+9%) | $0.0708^{\pm 0.0010}$ (+27%) | $17.8178^{\pm 0.2968}$ (+22%) |
| LNO | $0.0334^{\pm 0.0007}$ (+91%) | $18.8967^{\pm 0.4781}$ (+202%) | $0.3747^{\pm 0.0158}$ (+573%) | $46.2939^{\pm 0.5873}$ (+216%) |
| GP-UPT | $0.0206^{\pm 0.0004}$ (+18%) | $9.0023^{\pm 0.3885}$ (+44%) | $0.1537^{\pm 0.0051}$ (+176%) | $39.8774^{\pm 0.1191}$ (+172%) |
| Mesh-dependent AB-UPT | $\underline{0.0181}^{\pm 0.0000}$ (+3%) | $6.5665^{\pm 0.0523}$ (+5%) | $\underline{0.0696}^{\pm 0.0016}$ (+25%) | $\underline{16.1130}^{\pm 0.1627}$ (+10%) |
| Mesh-free AB-UPT | $0.0181^{\pm 0.0002}$ (+3%) | $\underline{6.4180}^{\pm 0.0236}$ (+2%) | $0.0774^{\pm 0.0019}$ (+39%) | $39.4740^{\pm 1.9237}$ (+169%) |
| SMART | $\textbf{0.0175}^{\pm 0.0001}$ | $\textbf{6.2627}^{\pm 0.0265}$ | $\textbf{0.0557}^{\pm 0.0004}$ | $\textbf{14.6639}^{\pm 0.0315}$ |

## L.1.2. ERRORS ON FULL-RESOLUTION DATASETS (RQ2)

Tables 22 and 23 show the mean errors and standard deviations across multiple runs for models trained on randomly subsampled and tested on full-resolution data with millions of points on the surface and volume. Running GNOT and Transolver at full resolution results in out-of-memory failures on an NVIDIA A100 80 GB GPU, even with a batch size of 1, caused by their use of self- and physics-attention over the spatial query coordinates.

*Table 22.* Rel. L2 errors of models tested on **full-resolution** datasets with millions of points on the surface and volume. Values in parentheses indicate the percentage deviation to SMART, and underlined values indicate the second-best errors.

| Model | Relative L2 Error $\times 10^{-2}$ ($\downarrow$) | | | |
| --- | --- | --- | --- | --- |
| | ShapeNetCar | | AhmedML | |
| | Surface | Volume | Surface | Volume |
| GINO | $14.4206^{\pm 0.0222}$ (+113%) | $10.3250^{\pm 0.1014}$ (+105%) | $16.1834^{\pm 0.1385}$ (+391%) | $17.4560^{\pm 0.1309}$ (+229%) |
| OFormer | $9.1898^{\pm 0.0325}$ (+36%) | $7.3559^{\pm 0.1622}$ (+46%) | $5.1494^{\pm 0.1432}$ (+56%) | $7.5140^{\pm 0.1560}$ (+42%) |
| GNOT | $13.0965^{\pm 0.3429}$ (+93%) | $11.2348^{\pm 0.4904}$ (+123%) | Out of memory (A100 80GB GPU) | |
| Transolver | $9.6288^{\pm 0.4703}$ (+42%) | $8.9447^{\pm 0.4378}$ (+78%) | Out of memory (A100 80GB GPU) | |
| LNO | $12.5501^{\pm 0.5860}$ (+85%) | $10.5334^{\pm 0.2259}$ (+109%) | $5.2975^{\pm 0.1718}$ (+61%) | $10.3790^{\pm 0.5008}$ (+96%) |
| GP-UPT | $\underline{6.9671}^{\pm 0.1113}$ (+3%) | $5.5764^{\pm 0.1353}$ (+11%) | $3.4917^{\pm 0.0334}$ (+6%) | $5.7412^{\pm 0.0820}$ (+8%) |
| Mesh-dependent AB-UPT | $7.0654^{\pm 0.2371}$ (+4%) | $\underline{5.3283}^{\pm 0.2322}$ (+6%) | $3.6450^{\pm 0.3001}$ (+11%) | $5.3253^{\pm 0.1491}$ (+0%) |
| Mesh-free AB-UPT | $6.9814^{\pm 0.0598}$ (+3%) | $5.3298^{\pm 0.2143}$ (+6%) | $\underline{3.3310}^{\pm 0.0716}$ (+1%) | $\mathbf{5.3012}^{\pm 0.0291}$ (-0%) |
| SMART | $\mathbf{6.7734}^{\pm 0.0577}$ | $\mathbf{5.0357}^{\pm 0.0766}$ | $\mathbf{3.2938}^{\pm 0.3285}$ | $\underline{5.3050}^{\pm 0.3483}$ |

*Table 23.* Rel. L2 errors of models tested on **full-resolution** datasets with millions of points on the surface and volume. Values in parentheses indicate the percentage deviation to SMART, and underlined values indicate the second-best errors.

| Model | Relative L2 Error $\times 10^{-2}$ ($\downarrow$) | | | |
| --- | --- | --- | --- | --- |
| | SHIFT-SUV | | SHIFT-Wing | |
| | Surface | Volume | Surface | Volume |
| GINO | $0.1218^{\pm 0.0002}$ (+596%) | $51.5854^{\pm 0.1934}$ (+720%) | $4.3343^{\pm 0.0019}$ (+7654%) | $75.2558^{\pm 0.0395}$ (+411%) |
| OFormer | $0.0225^{\pm 0.0001}$ (+29%) | $10.9924^{\pm 0.0365}$ (+75%) | $0.2653^{\pm 0.0060}$ (+375%) | $43.3266^{\pm 0.2038}$ (+194%) |
| GNOT | Out of memory (A100 80GB GPU) | | | |
| Transolver | Out of memory (A100 80GB GPU) | | | |
| LNO | $0.0335^{\pm 0.0007}$ (+91%) | $18.9424^{\pm 0.4769}$ (+201%) | $0.3834^{\pm 0.0151}$ (+586%) | $46.3603^{\pm 0.5500}$ (+215%) |
| GP-UPT | $0.0206^{\pm 0.0004}$ (+18%) | $9.0318^{\pm 0.3935}$ (+43%) | $0.1567^{\pm 0.0066}$ (+180%) | $39.9266^{\pm 0.1018}$ (+171%) |
| Mesh-dependent AB-UPT | $\underline{0.0181}^{\pm 0.0000}$ (+3%) | $6.5956^{\pm 0.0369}$ (+5%) | $\underline{0.0705}^{\pm 0.0015}$ (+26%) | $\mathbf{16.1691}^{\pm 0.1568}$ (+10%) |
| Mesh-free AB-UPT | $0.0181^{\pm 0.0001}$ (+3%) | $\underline{6.4445}^{\pm 0.0372}$ (+2%) | $0.0775^{\pm 0.0016}$ (+39%) | $39.5142^{\pm 1.9202}$ (+168%) |
| SMART | $\mathbf{0.0175}^{\pm 0.0001}$ | $\mathbf{6.2946}^{\pm 0.0212}$ | $\mathbf{0.0559}^{\pm 0.0002}$ | $\underline{14.7187}^{\pm 0.0231}$ |

## L.1.3. ERRORS FOR MODELS QUERIED ON MESHES NOT OPTIMIZED FOR CFD (RQ3)

Table 24 shows the mean errors and standard deviations across multiple runs for models trained on points randomly sampled from the CFD-optimized simulation meshes (surface and volume) and tested on points that are sampled from meshes that are not optimized for CFD. In particular, the surface points are based on the CAD geometry, and the volume mesh is an (approximately) uniform grid. Both meshes differ substantially from the CFD-optimized meshes used during training.

*Table 24.* Rel. L2 errors of models tested on the SHIFT-SUV dataset using the **CAD-mesh** as the surface mesh (600k points) and an **uniform mesh** as the volume mesh (2M points). Values in parentheses indicate the percentage deviation to SMART, and underlined values indicate the second-best errors.

| Model | Relative L2 Error $\times 10^{-2}$ ($\downarrow$) | |
| --- | --- | --- |
| | SHIFT-SUV Uniform | |
| | Surface | Volume |
| GINO | $0.1225^{\pm 0.0002}$ (+596%) | $46.9132^{\pm 5.0717}$ (+688%) |
| OFormer | $0.0225^{\pm 0.0001}$ (+28%) | $12.5240^{\pm 0.1543}$ (+110%) |
| GNOT | $0.0765^{\pm 0.0043}$ (+335%) | $45.9890^{\pm 4.3812}$ (+672%) |
| Transolver | $0.0622^{\pm 0.0092}$ (+253%) | $25.0725^{\pm 4.4068}$ (+321%) |
| LNO | $0.0334^{\pm 0.0007}$ (+90%) | $25.6217^{\pm 1.6678}$ (+330%) |
| GP-UPT | $0.0207^{\pm 0.0004}$ (+18%) | $11.3625^{\pm 0.6649}$ (+91%) |
| Mesh-dependent AB-UPT | $0.1172^{\pm 0.0117}$ (+566%) | $140.9773^{\pm 34.2689}$ (+2268%) |
| Mesh-free AB-UPT | $\underline{0.0181}^{\pm 0.0001}$ (+3%) | $\underline{7.3500}^{\pm 0.1369}$ (+23%) |
| SMART | $\mathbf{0.0176}^{\pm 0.0001}$ | $\mathbf{5.9543}^{\pm 0.0324}$ |

## L.1.4. ERRORS FOR MODELS QUERIED ON A SUBREGION OF THE SPATIAL DOMAIN (RQ4)

Table 25 reports the mean errors and standard deviations across multiple runs for models trained on points randomly sampled from the full spatial domain and evaluated exclusively on points located in the rear region of the vehicle. This evaluation

setting reflects practical design scenarios in which interest is restricted to a specific subregion of the spatial domain. For example, when assessing the aerodynamic impact of adding a spoiler, it is sufficient to query the model only at coordinates within the rear section of the car.

*Table 25.* Rel. L2 errors of models tested on the SHIFT-SUV dataset using only the mesh points corresponding to the **rear region** of the vehicles (1.5M points on surface and volume). Values in parentheses indicate the percentage deviation to SMART, and underlined values indicate the second-best errors.

| Model | Relative L2 Error $\times 10^{-2}$ ($\downarrow$) | |
| --- | --- | --- |
| | SHIFT-SUV Rear | |
| | Surface | Volume |
| GINO | $0.1001^{\pm 0.0002}$ (+581%) | $57.5154^{\pm 0.2903}$ (+870%) |
| OFormer | $0.0181^{\pm 0.0001}$ (+23%) | $11.3537^{\pm 0.0450}$ (+91%) |
| GNOT | $0.0326^{\pm 0.0017}$ (+122%) | $22.8793^{\pm 1.8053}$ (+286%) |
| Transolver | $0.0321^{\pm 0.0019}$ (+118%) | $18.0383^{\pm 0.8143}$ (+204%) |
| LNO | $0.0257^{\pm 0.0009}$ (+75%) | $20.6669^{\pm 0.7360}$ (+248%) |
| GP-UPT | $0.0174^{\pm 0.0003}$ (+18%) | $9.2568^{\pm 0.6649}$ (+56%) |
| Mesh-dependent AB-UPT | $0.0660^{\pm 0.0121}$ (+349%) | $33.6630^{\pm 3.1073}$ (+467%) |
| Mesh-free AB-UPT | $\underline{0.0156}^{\pm 0.0001}$ (+6%) | $\underline{6.2652}^{\pm 0.0718}$ (+6%) |
| SMART | $\mathbf{0.0147}^{\pm 0.0000}$ | $\mathbf{5.9319}^{\pm 0.0237}$ |

### L.1.5. COMPUTATION OF DRAG AND LIFT FORCES

Drag and lift coefficients are key aerodynamic performance metrics in both automotive and aerospace applications. They depend directly on the aerodynamic force vector $\mathbf{F}$, which represents the net effect of how the flow interacts with the body surface. Pressure differences across the geometry generate normal forces, while viscous stresses within the boundary layer contribute tangential forces. Together, they determine the magnitude and direction of $\mathbf{F}$, which is defined as the surface integral

$$\mathbf{F} = \int_S \left( -p\,\mathbf{n} + \boldsymbol{\tau_\omega} \right) ds \approx \sum_{s_i \in S} \left( -p(s_i)\,\mathbf{n}(s_i) + \boldsymbol{\tau_\omega}(s_i) \right) A(s_i) \tag{11}$$

where $p$ denotes the surface pressure field, $\mathbf{n}$ the normal vector, and $\boldsymbol{\tau_\omega}$ the wall shear stress on the surface. The surface integral can be approximated from a discrete simulation by summing over all surface cells and their corresponding pressure acting in the normal direction and wall shear stress, multiplied by the surface cell area $A(s_i)$. Drag and lift forces correspond to the components of $\mathbf{F}$ in the freestream direction (typically the $x$-axis) and the direction perpendicular to it (typically the $z$-axis). Projecting the aerodynamic force vector onto the relevant directions yields the following expressions

**Automotive (no angle of attack)**  **Aerospace (with angle of attack $\alpha$)**

$$\text{Drag: } F_d = \mathbf{F} \begin{bmatrix} 1 \\ 0 \\ 0 \end{bmatrix} \qquad\qquad \text{Drag: } F_d = \mathbf{F} \begin{bmatrix} \cos\alpha \\ 0 \\ \sin\alpha \end{bmatrix}$$

$$\text{(12)} \qquad\qquad\qquad \text{(13)}$$

$$\text{Lift: } F_l = \mathbf{F} \begin{bmatrix} 0 \\ 0 \\ 1 \end{bmatrix} \qquad\qquad \text{Lift: } F_l = \mathbf{F} \begin{bmatrix} -\sin\alpha \\ 0 \\ \cos\alpha \end{bmatrix}$$

for the lift and drag forces in automotive and aerospace applications. The corresponding drag and lift coefficients $C_d$ and $C_l$ are computed as $C_. = \frac{2\,F_.}{\rho\,v^2\,A}$, where $\rho$ is the density, $v$ the velocity, and $A$ the reference area. In this work, we focus on comparing the forces directly rather than their non-dimensional coefficients. We evaluate the drag and lift forces computed from the CFD pressure and wall shear stress fields (ground truth) against those obtained from SMART's predicted fields. Figure 19 presents scatter plots for the SHIFT-SUV (estate, full-scale subset) and SHIFT-Wing (only first 1000 samples considered for training and testing) datasets along with their $R^2$ scores. SMART accurately predicts drag for the SHIFT-SUV dataset, and lift is well captured for values smaller than approximately $-200$ N. A similar behavior was reported for AB-UPT and can be attributed to the limited number of geometries in the dataset exhibiting lift forces greater than $-200$ N (Alkin et al., 2025b). For SHIFT-Wing, both drag and lift forces are predicted with high accuracy.

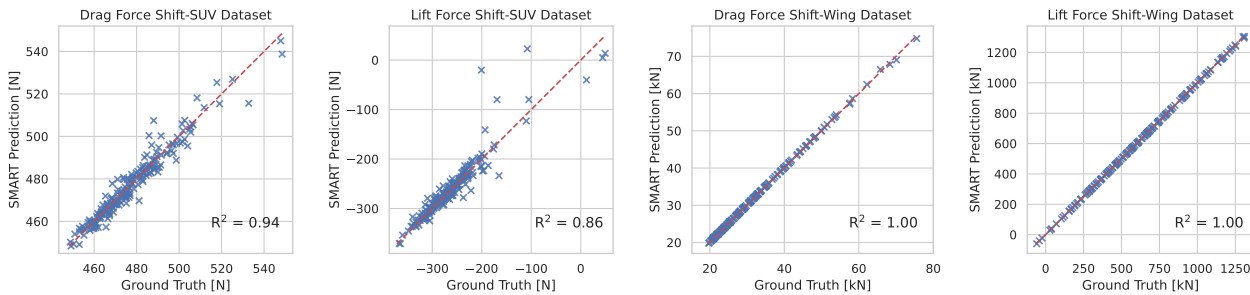

*(a)* Drag Forces for SHIFT-SUV  *(b)* Lift Forces for SHIFT-SUV  *(c)* Drag Forces for SHIFT-Wing  *(d)* Lift Forces for SHIFT-Wing

*Figure 19.* Scatter plots comparing the ground truth and predicted drag and lift forces for the SHIFT-SUV (estate, full-scale subset) and SHIFT-Wing (only first 1000 samples considered for training and testing) datasets. The red line indicates the line of best fit. The accompanying $R^2$ scores quantify prediction accuracy, with values closer to 1 indicating better agreement.

### L.1.6. GENERALIZATION TO NOVEL SHAPES

We examine how well SMART can generalize to novel shapes by training a model on the estate subset of the SHIFT-SUV dataset and evaluating it on the fastback subset. For comparison, we also train a SMART model directly on the fastback subset of SHIFT-SUV. As shown in Table 26, the errors increase when the model trained on estate SUVs is evaluated on fastback SUVs. However, this behavior is expected, as the model has only learned the aerodynamic characteristics of estate-type vehicles. These results suggest that SMART does not generalize reliably to shape categories that differ from those seen during training. Consequently, either finetuning on a small number of target-domain samples or training on a more diverse dataset spanning multiple shape types is required to enable zero-shot generalization to new geometries.

*Table 26.* Rel. L2 errors of models tested on the SHIFT-SUV fastback dataset using a subsampled resolution of 16k and the full resolution. Values in parentheses indicate the percentage deviation to the model that is trained on the fastback subset of SHIFT-SUV.

| | Relative L2 Error $\times 10^{-2}$ ($\downarrow$) | | | |
| --- | --- | --- | --- | --- |
| Training Dataset | SHIFT-SUV Fastback | | | |
| | Subsampled | | Full Resolution | |
| | Surface | Volume | Surface | Volume |
| SHIFT-SUV Estate | 0.0338 (+84%) | 17.5187 (+167%) | 0.0339 (+82%) | 17.5204 (+168%) |
| SHIFT-SUV Fastback | **0.0184** | **6.5641** | **0.0186** | **6.5489** |

### L.1.7. INFERENCE TIME AND COMPUTATIONAL COMPLEXITY

Additionally, we evaluate the wall-clock time and computational complexity of SMART during inference on the SHIFT-Wing dataset. The model is configured with 8 encoder and decoder blocks, a latent geometry size of $K = 4,096$, and `float32` precision. Under this configuration, and when evaluated with 5 million query points, the wall-clock inference time is $17.11^{\pm 0.0656}$ s. The corresponding compute cost is $60,968$ GFLOPS. Both values are measured on an A100 SXM4 80GB GPU.

### L.1.8. COMPARISON TO TRANSOLVER++ ON FULL-RESOLUTION DATASETS

We compare Transolver++ (Luo et al., 2025) with SMART on the large-scale datasets at their full resolution, each containing millions of points. For all datasets, Transolver++ runs out-of-memory on a single A100 80GB GPU. Based on the memory scaling reported in their paper (Figure 1a), a linear extrapolation suggests that approximately 20 A100 80GB GPUs would be required to run inference on the SHIFT-SUV dataset with 53M mesh points. In contrast, SMART requires only 7GB of GPU memory on this dataset.

*Table 27.* Rel. L2 errors of models tested on **full-resolution** datasets with millions of points on the surface and volume.

| Model | Relative L2 Error $\times 10^{-2}$ ($\downarrow$) | | | | | |
|---|---|---|---|---|---|---|
| | AhmedML | | SHIFT-SUV | | SHIFT-Wing | |
| | Surface | Volume | Surface | Volume | Surface | Volume |
| Transolver++ | Out of memory (A100 80GB GPU) | | | | | |
| SMART | $3.2938^{\pm 0.3285}$ | $5.3050^{\pm 0.3483}$ | $0.0175^{\pm 0.0001}$ | $6.2946^{\pm 0.0212}$ | $0.0559^{\pm 0.0002}$ | $14.7187^{\pm 0.0231}$ |

## L.2. Qualitative Results

### L.2.1. SHAPENETCAR DATASET

Figures 20 to 22 show the ground truth and predicted pressure and velocity fields for one random test sample from the ShapeNetCar dataset.

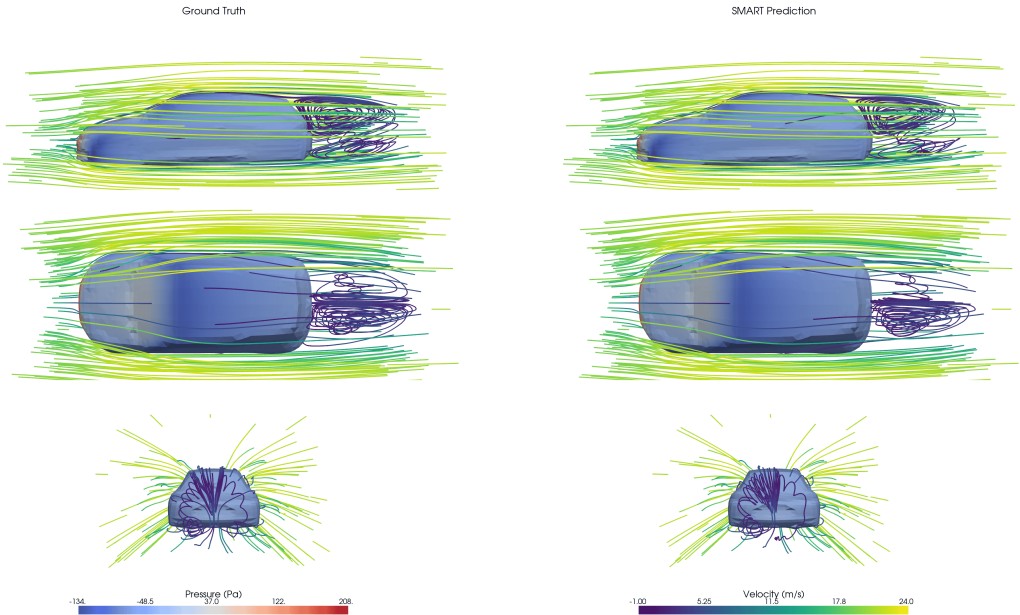

*Figure 20.* Ground truth and predicted surface pressure and velocity fields for one random test sample from the ShapeNetCar dataset. The velocity fields are represented as streamlines.

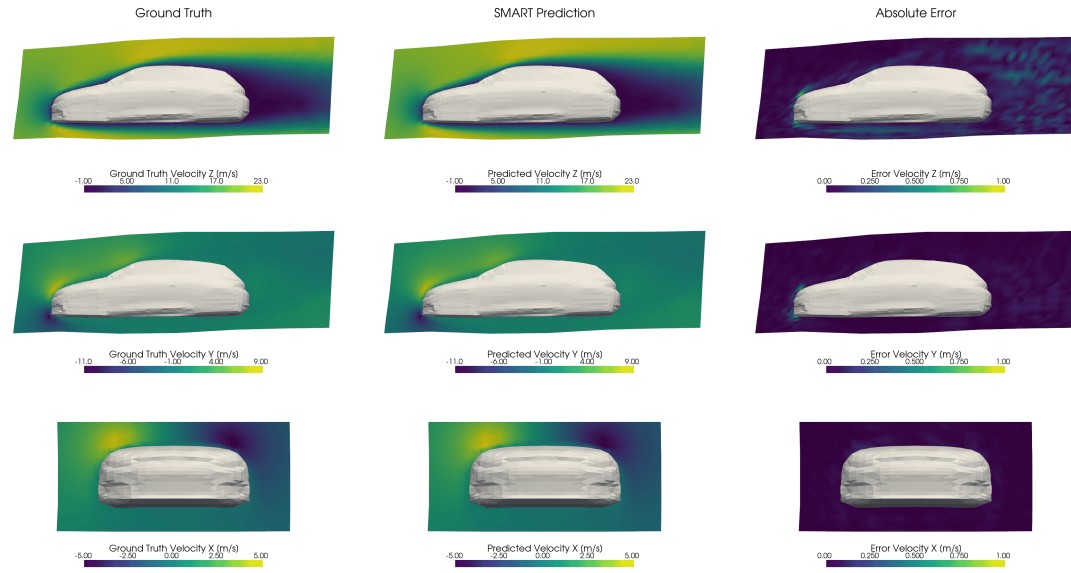

*Figure 21.* Ground truth and predicted velocity fields on 2D planes of one random test sample from the ShapeNetCar dataset. Each row shows one of the $x, y, z$ velocity components.

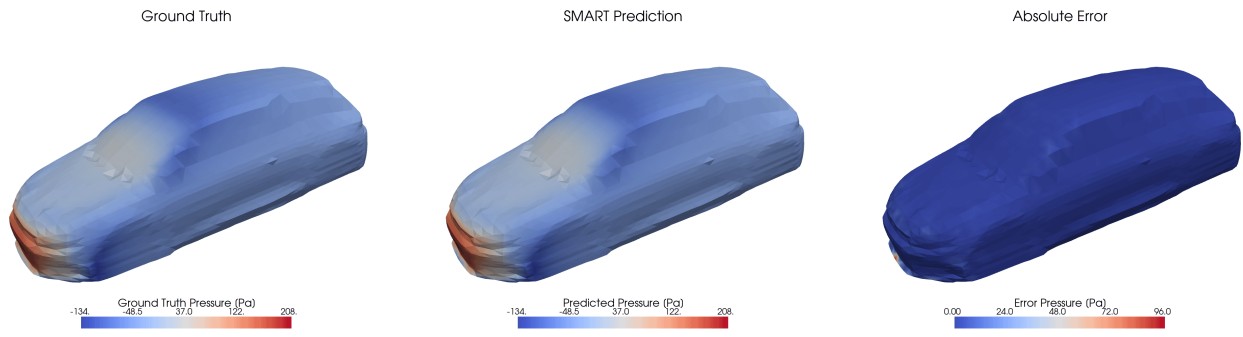

*Figure 22.* Ground truth and predicted surface pressure fields for one random test sample from the ShapeNetCar dataset.

### L.2.2. AHMEDML DATASET

Figures 23 to 25 show the ground truth and predicted pressure and velocity fields for one random test sample from the AhmedML dataset.

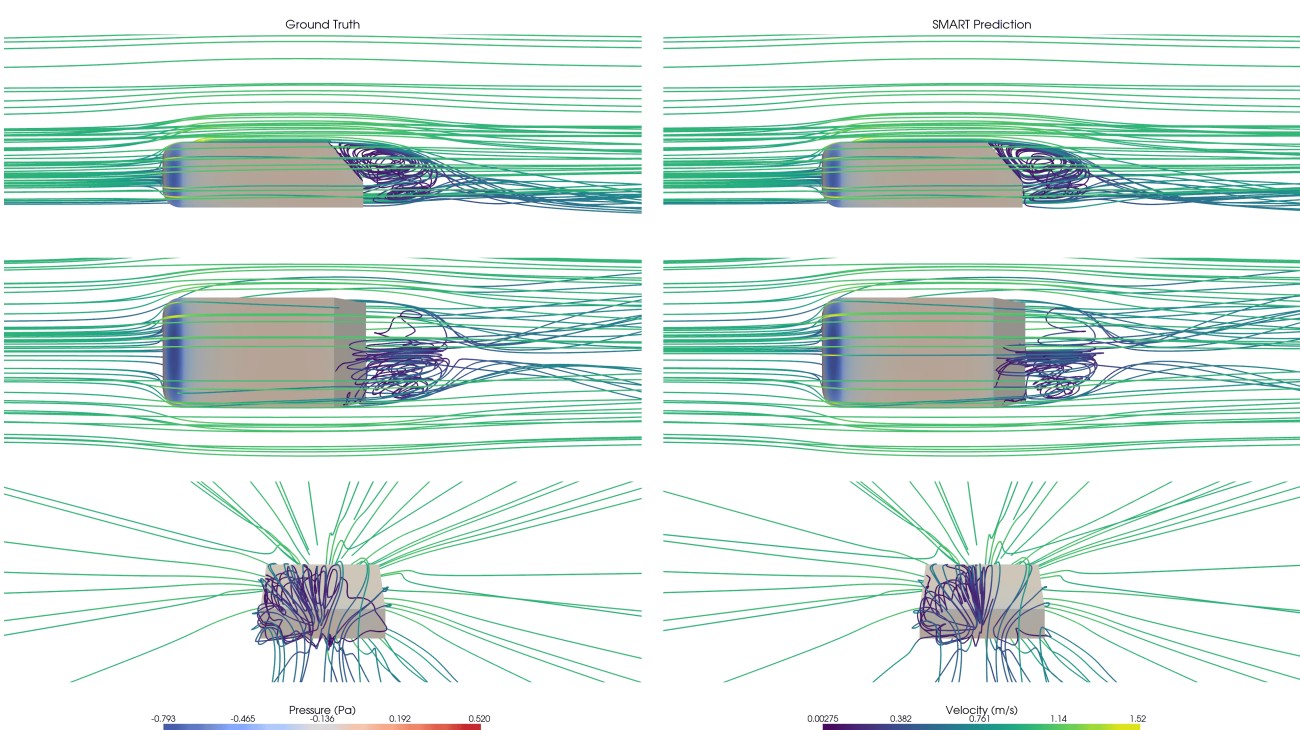

*Figure 23.* Ground truth and predicted surface pressure and velocity fields for one random test sample from the AhmedML dataset. The velocity fields are represented as streamlines.

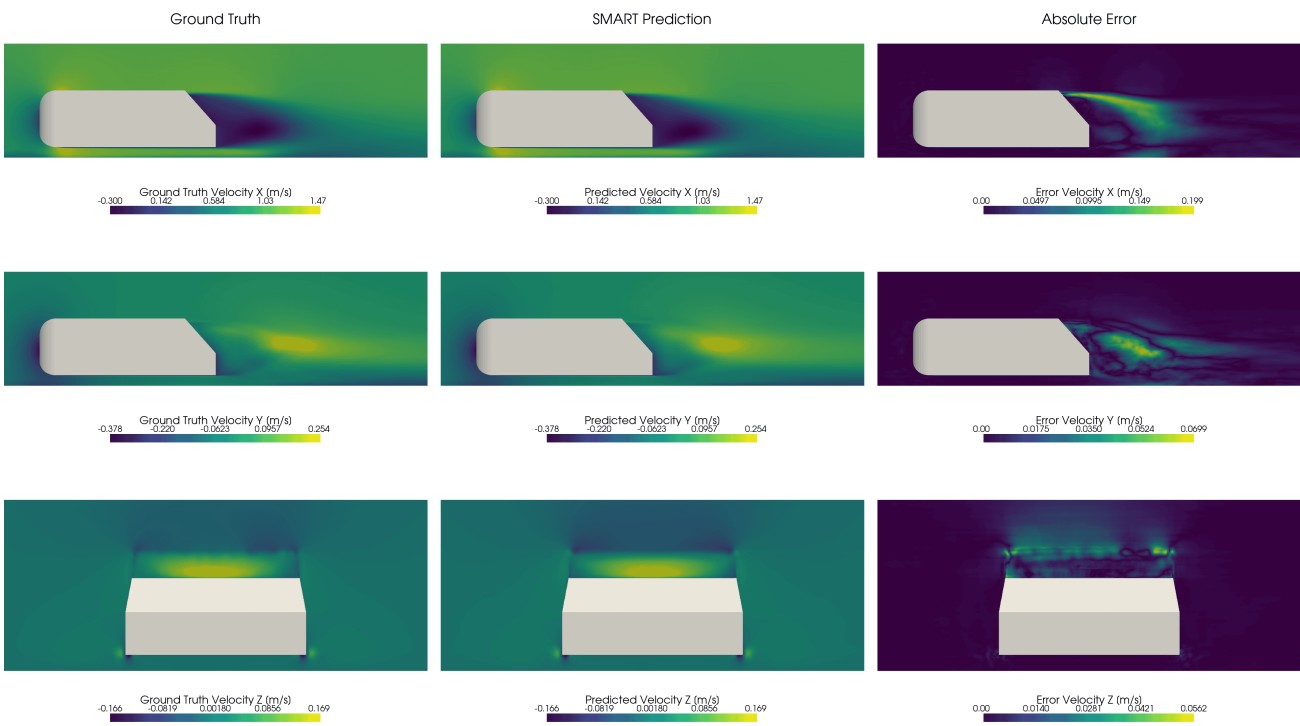

*Figure 24.* Ground truth and predicted velocity fields on 2D planes of one random test sample from the AhmedML dataset. Each row shows one of the $x, y, z$ velocity components.

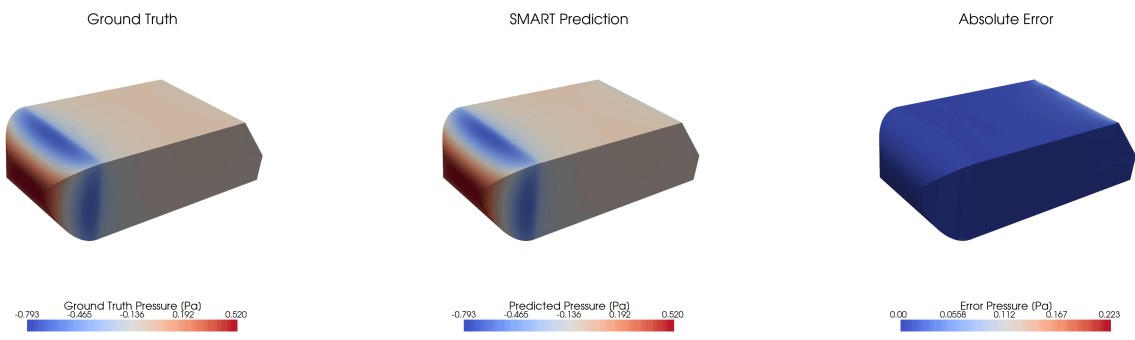

*Figure 25.* Ground truth and predicted surface pressure fields for one random test sample from the AhmedML dataset.

### L.2.3. SHIFT-SUV DATASET

Figures 26 to 28 show the ground truth and predicted pressure and velocity fields for one random test sample from the SHIFT-SUV dataset.

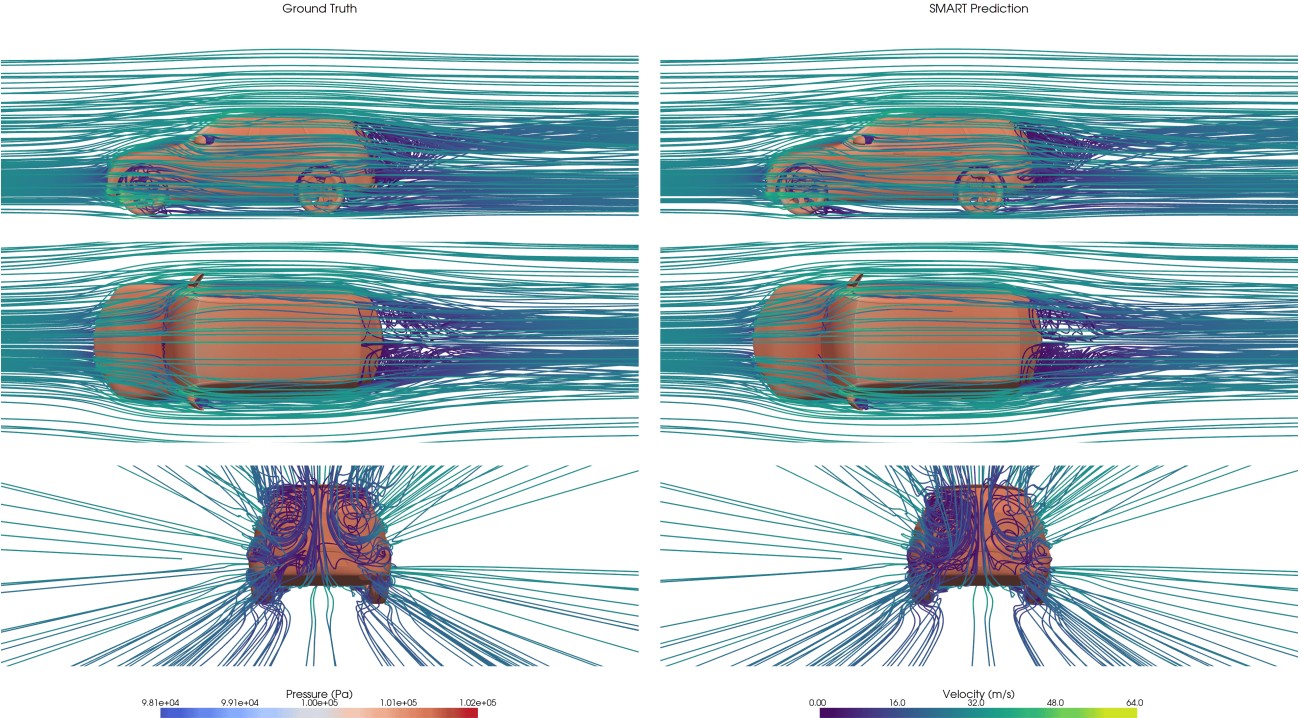

*Figure 26.* Ground truth and predicted surface pressure and velocity fields for one random test sample from the SHIFT-SUV dataset. The velocity fields are represented as streamlines.

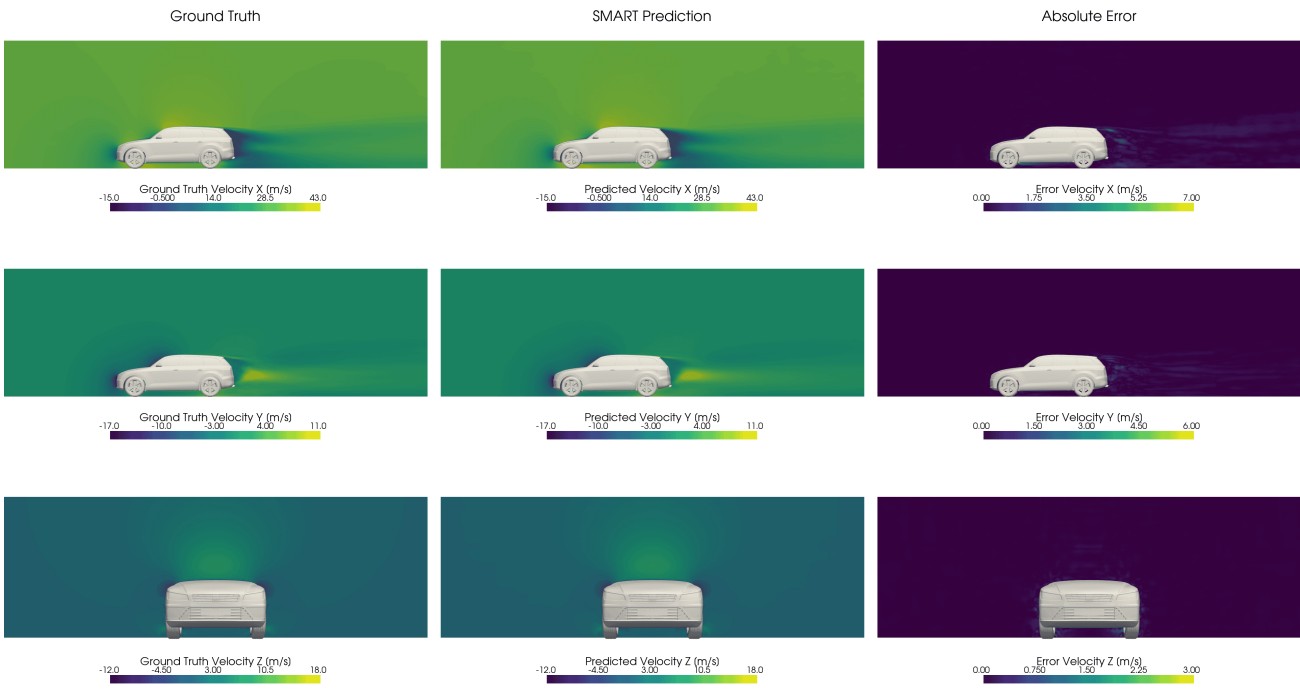

*Figure 27.* Ground truth and predicted velocity fields on 2D planes of one random test sample from the SHIFT-SUV dataset. Each row shows one of the $x, y, z$ velocity components.

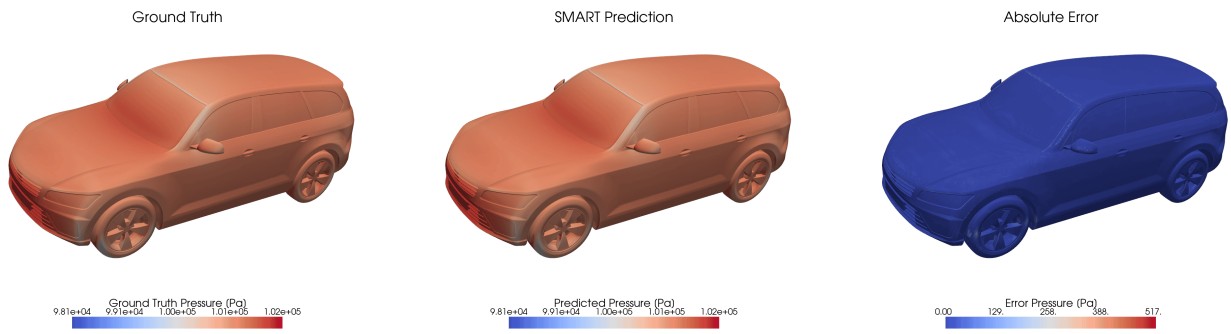

*Figure 28.* Ground truth and predicted surface pressure fields for one random test sample from the SHIFT-SUV dataset.

### L.2.4. SHIFT-WING DATASET

Figures 29 to 31 show the ground truth and predicted pressure and velocity fields for one random test sample from the SHIFT-Wing dataset.

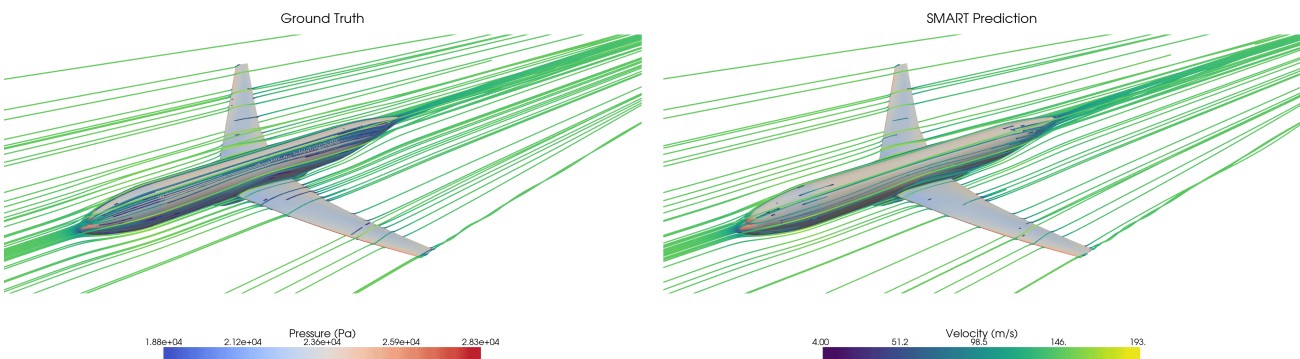

*Figure 29.* Ground truth and predicted surface pressure and velocity fields for one random test sample from the SHIFT-Wing dataset. The velocity fields are represented as streamlines.

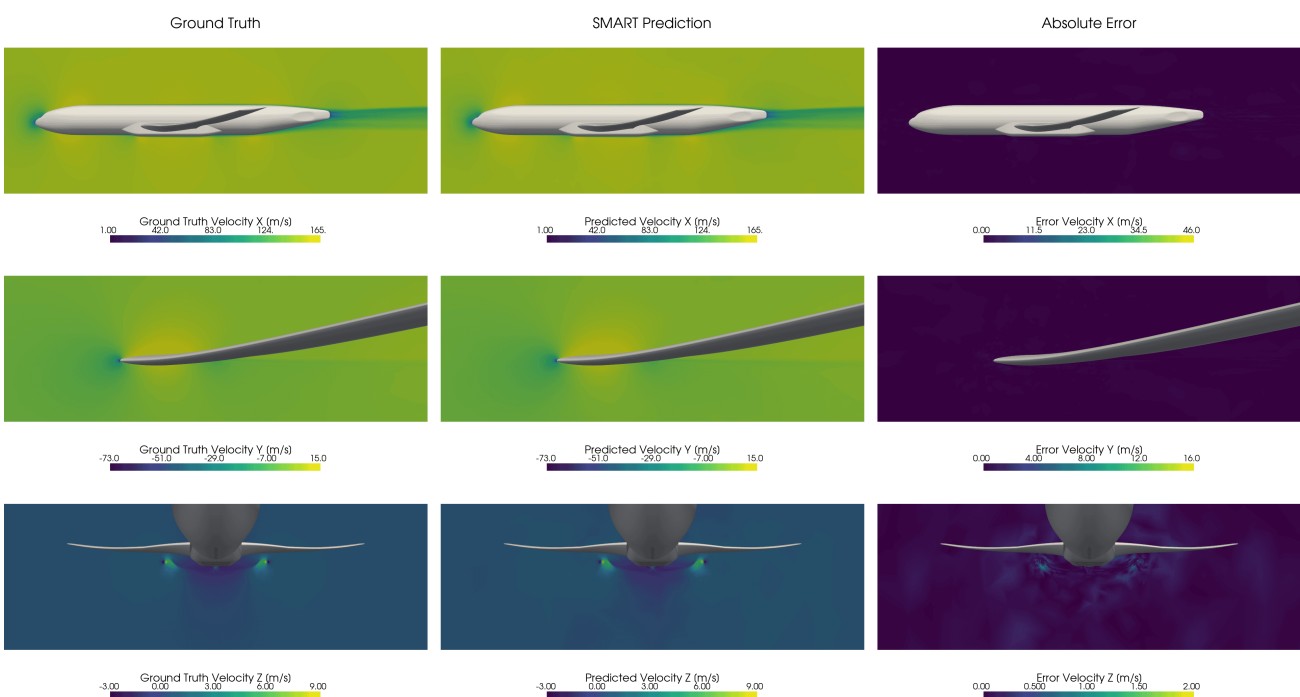

*Figure 30.* Ground truth and predicted velocity fields on 2D planes of one random test sample from the SHIFT-Wing dataset. Each row shows one of the $x, y, z$ velocity components.

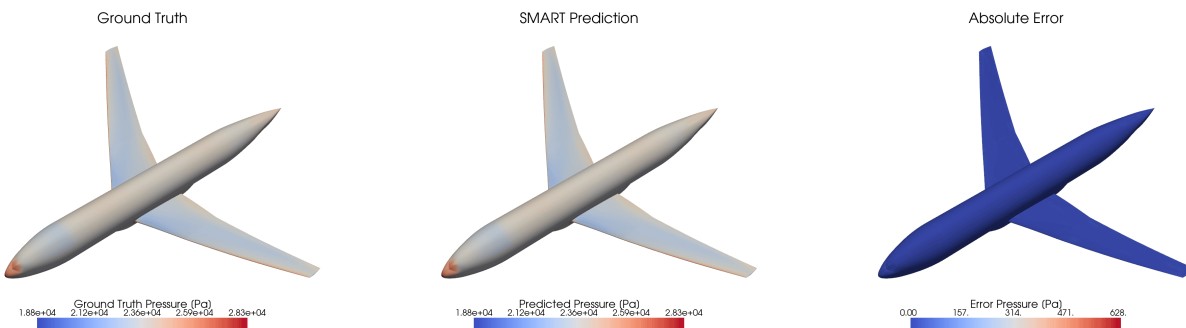

*Figure 31.* Ground truth and predicted surface pressure fields for one random test sample from the SHIFT-Wing dataset.

