# OpenReview forum: "SMART: Scalable Mesh‑free Aerodynamic Simulations from Raw Geometries using a Transformer‑based Surrogate Model"
_ICML.cc/2026/Conference — ICML 2026 regular_

### Official Review · Reviewer_iaGh · 2026-03-03

**Soundness:** 3
**Presentation:** 3
**Significance:** 3
**Originality:** 3
**Overall Recommendation:** 4
**Confidence:** 4

**Summary:**

This paper presents SMART, a neural surrogate model that predicts physical quantities at arbitrary query locations using only the point-cloud representation of the geometry, without the need for the simulation mesh. Simulation mesh can be computationally expensive to generate, and some some simulation cases adapt the simulation mesh during the simulation process, therefore the capability to predict physical quantities without the simulation mesh is an advantage over previous methods. The network employs an encoder-decoder architecture, with the geometry and simulation parameters encoded into a latent space with fixed dimension, and the decoder maps arbitrary spatial coordinates to the physical field through cross-layer interaction.

**Compliance With Llm Reviewing Policy:**

Affirmed.

**Final Justification:**

This is a solid paper and the rebuttal addresses my main concerns. I remain positive about this paper.

**Key Questions For Authors:**

In introduction you say "approaches that treat the simulation mesh solely as query points typically suffer from substantially higher error". What do you think is most important for SMART to mitigate this limitation?

**Limitations:**

Yes.

**Strengths And Weaknesses:**

Strengths:

- Simulation meshes can be expensive to obtain, while for neural surrogate models the incorporation of simulation mesh generally yields lower prediction error. This paper provides a model that can predict physical quantities without relying on the simulation mesh.
- The proposed model features a geometry encoder which evolve the geometry and parameter information with two cross-attention layers and a physics decoder which cross-attend to the geometry information layer by layer, which helps multi-scale physics learning.
- The architectural design allows flexible training, by sampling training points randomly from the original geometry, both for geometry encoding and spatial query.
- Experimental results demonstrate the superior performance of the proposed method.



Weaknesses:

- Although SMART does not rely on simulation mesh, the geometry encoder relies on the distribution of randomly sampled points. It may encounter a distribution shift when trained with uniformly sampled points and applied to a locally refined simulation mesh typical of industrial CFD.
- The ablation studies lacks comparison between different point sampling strategies.
- For models like Transolver, Transolver++ has provided solutions to train and inference on million-scale meshes, yet it is labelled as "OOM" in Table 3 for 3 of large-scale geometry datasets.

---

> ### Author Rebuttal · Authors · 2026-03-30
>
> Thank you for your effort in reviewing our work and your valuable feedback, which contributed to strengthening our submission. We appreciate your recognition of SMART's contributions, including its mesh‑independent prediction capability, hierarchical cross‑attention design, and strong empirical performance. We are happy to address the mentioned weaknesses and questions below.
>
> > **(W1)** Although SMART does not rely on simulation mesh, the geometry encoder relies on the distribution of randomly sampled points. It may encounter a distribution shift when trained with uniformly sampled points and applied to a locally refined simulation mesh typical of industrial CFD.
>
> If the distribution of the input geometry points to the encoder is changed, it may affect the model's performance in a negative way. However, SMART uses the CAD geometry mesh as input and not (locally refined) CFD meshes. As long as the model is applied to CAD geometry meshes as input, it does not affect the model negatively. The output query coordinates are independent of the input coordinates and independent of each other. This means that the model can be queried with arbitrary coordinates in the simulation domain, for instance, from a locally refined mesh.
>
> > **(W2)** The ablation studies lacks comparison between different point sampling strategies.
>
> We conducted an experiment on the SHIFT-SUV dataset with farthest-point-sampling (FPS) instead of uniform sampling. The results show that there is no significant difference between uniform and farthest-point sampling.
>
> | Sampling | Subsampled Surface Rel. L2 ($\times 10^{-2}$) | Subsampled Volume Rel. L2 ($\times 10^{-2}$) | Full Surface Rel. L2 ($\times 10^{-2}$) | Full Volume Rel. L2 ($\times 10^{-2}$) |
> |---|---|---|---|---|
> | Uniform | 0.0175 | 6.2627 | 0.0175 | 6.2946 |
> | FPS | 0.0174 | 6.2449 | 0.0175 | 6.2922 |
>
> > **(W3)** For models like Transolver, Transolver++ has provided solutions to train and inference on million-scale meshes, yet it is labelled as "OOM" in Table 3 for 3 of large-scale geometry datasets.
>
> Transolver++ enables inference on million-scale meshes (2.5M mesh points) by removing the over-parametrization of the model and by scaling the physics-attention mechanism across multiple GPUs. We have now included Transolver++, which also yields out-of-memory errors for all 3 large-scale datasets on a single A100 80GB GPU. If we linearly extrapolate the GPU memory consumption from Transolver++ reported in their paper (Figure 1a), about 20 A100 80GB GPUs would be required to run inference on the SHIFT-SUV dataset with 53M mesh points. For comparison, SMART requires only 7GB on this dataset.
>
> > **(Q1)** In introduction you say "approaches that treat the simulation mesh solely as query points typically suffer from substantially higher error". What do you think is most important for SMART to mitigate this limitation?
>
> We identify the use of a coarse geometry as a scaffold for the latent representation and the joint coupling of the encoder-decoder through the cross-layer geometry-physics update as the most important components to mitigate this limitation. The ablation study in Appendix K (page 29) shows that these components significantly reduce the errors, supporting our hypothesis.

---

> > ### Author Rebuttal · Reviewer_iaGh · 2026-04-01
> >
> > I thank the authors for providing a clear rebuttal. My concerns are resolved.

---

> > > ### Author Response · Authors · 2026-04-02
> > >
> > > Thank you for taking the time to review our work and your response. We are glad that we have resolved all of your concerns. All of your improvements will be included in the revised manuscript, including the comparison of the different point sampling strategies and the results for Transolver++.
> > >
> > > Given that all raised issues have been resolved, we kindly ask whether you might consider revising your score to reflect this, or inform us of any outstanding concerns or questions that need clarification.

---

### Official Review · Reviewer_gC68 · 2026-03-11

**Soundness:** 3
**Presentation:** 3
**Significance:** 3
**Originality:** 2
**Overall Recommendation:** 4
**Confidence:** 5

**Summary:**

This paper introduces SMART, a Transformer-based surrogate model designed for 3D aerodynamic simulations that operates directly on raw geometric point clouds. The framework addresses the significant computational bottleneck of traditional mesh-informed surrogates by proposing a mesh-free architecture. To achieve this, the authors use a Geometry Encoder that compresses raw geometries into a fixed-size set of latent tokens through iterative cross-attention and a Physics Decoder that maps arbitrary spatial queries to physical quantities such as surface pressure and velocity fields. The model uses Modulated Positional Encoding to capture multi-scale spatial variations and a Cross-Layer Geometry-Physics Update to set the physics predictions in the geometric representation. The authors evaluate the model on several datasets, including SHIFT-SUV and SHIFT-Wing, showing that SMART achieves accuracy comparable to state-of-the-art mesh-dependent models while being similarlry accurate to raw CAD geometries.

**Compliance With Llm Reviewing Policy:**

Affirmed.

**Final Justification:**

Overall, this is a solid paper with strong practical utility, and you have adequately addressed most of my primary concerns. I will have my score of 4 (Weak Accept).

**Key Questions For Authors:**

1. Given that to create such datasets can take a lot of time, it is important to know how the model performs on geometries with fundamentally different topologies than the training set? For example, if trained on bluff-body cars, could the model accurately predict the flow around a bikes?

2. How does the architecture distinguish between surface and volume queries in "hollow" geometries (e.g., a car with open windows)? Without an explicit mesh or manifold awareness, how do you ensure the decoder doesn't produce unphysical predictions for coordinates that are within the geometry?

3. Since a physics-informed loss is not employed, have you analyzed the divergence of the predicted velocity fields? When dealing with flows around objects, it is important to guarantee conservation of mass and momentum, especially at the wake regions, which can affect the drag and lift coefficient predictions.

4. What decides how many sampling points K to use for each geometry? How can one decide that a priori?

**Limitations:**

A primary concern is that the model does not inherently respect physical conservation laws, which can lead to unphysical flow patterns that L2 metrics may overlook. In addition, the uniform sampling is a limitation for capturing boundary layer physics and steep flow gradients, which are critical for accurate drag and lift predictions. Finally, the framework is not tested on out-of-distribution geometries, since creating these datasets and retraining can take a lot of time.

**Strengths And Weaknesses:**

# Strengths
### Soundness
The authors provide an extensive experimental evaluation across different datasets. Specifically, the use of the SHIFT-Wing dataset, where they use adaptively refined meshes, provides strong evidence for the model's ability to generalize across varying point densities, especially close to geometry surfaces.

### Presentation
The paper is well-structured, and the flow is easy to follow. The authors identify the mesh-generation bottleneck in current CFD workflows and position their work as a solution. The figures showing the encoder-decoder interaction and the cross-layer updates effectively communicate the model innovations. Also, the visualizations help in interpreting the results (both flow and pressure-based)

### Significance
The removal of the mesh requirement is significant for aerodynamic design. By operating on raw point clouds, the framework enables faster iteration cycles in the early design phase, where high-fidelity meshes are not yet available or required. The model independence in terms of queries also shows high scalability for applications at the industry level.

### Originality
The main originality lies in the integration of Modulated Positional Encoding with a mesh-free Transformer architecture. While components like cross-attention are standard, the specific application of a probing mechanism to a latent geometric scaffold for 3D physics is also something that has very recently emerged as an application.


# Weaknesses
### Soundness
The major concerns are the following. In high Re number flows, the physics are governed by thin boundary layers and steep gradients (e.g. velocity and pressure over an airfoil), and uniform sampling is likely to miss these critical small-scale flow features. It is known that not capturing these correctly could lead to significant errors in drag and lift coefficients. In addition, there is no discussion of physical conservation laws (conservation of mass and momentum). Without enforcing these constraints, the model may produce unphysical flow patterns in wake regions.

### Presentation
 While Section K.1 ablates K for a single dataset, the paper should have a more formal discussion or heuristic on how K should be chosen relative to geometric complexity (e.g. why the wing requires twice as many tokens as the SUV despite having fewer volume points).

### Significance
The framework's use is limited by its focus on steady-state predictions. As the authors acknowledge in the limitations, the model does not handle time-dependent flows. In fields where transient phenomena such as vortex shedding and peak loads are imporand, this restriction limits the model's applicability. Furthermore, the accuracy is limited by the diversity of the training set. The paper does not show the model ability to handle out-of-distribution geometries.

### Originality
The architecture has some overlap with existing models, such as AB-UPT. While the application to 3D aerodynamic fields is well-defined, the implemented components and cross-layer updates work more as improvements to these established architectures rather than a completely new model.

---

> ### Author Rebuttal · Authors · 2026-03-30
>
> Thank you for your thoughtful feedback and your questions that helped us to refine our submission. We appreciate that you consider our method a significant improvement to enable faster iteration cycles in design.
>
> > **(W1)** Sampling
>
> The physical field is resampled for each sample in each epoch. Thus, the model will observe the entire physical field in expectation over the training. Additionally, we compute drag and lift forces in Appendix L.1.5, which shows that SMART accurately predicts drag and lift forces. Please note that there was a mistake in our code for the drag on SHIFT-Wing, which only affects the raw forces and not the relation between predictions and ground truth. This suggests that uniform sampling works in many settings. However, we acknowledge that more sophisticated sampling strategies could improve the model, which we plan for future work.
>
> > Physical laws
>
> Incorporating explicit conservation constraints is a promising direction. We consider this complementary to the current work and plan to explore it for future work. We added it to the limitations section of the manuscript. Please also see our response to (Q3).
>
> > **(W2)** and **(Q4)** Size of K
>
> K is a task‑dependent hyperparameter that must be adapted to the geometric complexity, flow physics, and effective size of the simulation domain. In the SHIFT‑Wing dataset, the domain is substantially larger and the wake exhibits long, coherent trailing‑vortex structures. Capturing these 3D flow features requires a larger number of latent tokens. In contrast, the SHIFT‑SUV dataset represents a bluff‑body flow with a short, rapidly breaking turbulent wake, so fewer tokens are sufficient despite having more volume points. For these reasons, K scales with the spatial extent and the underlying flow structures, not merely with the number of grid points.
>
> > **(W3)** Steady-state
>
> Steady‑state predictions are essential for optimizing time‑averaged metrics like drag or lift, and most large‑scale datasets are built around steady‑state simulations. However, we plan to extend SMART for time-dependent simulations for future work. Adapting the model for time-dependent simulations is non-trivial and could include the time as an additional modulation parameter or using a latent time-stepping approach, which requires further research.
>
> > Out-of-distribution
>
> We followed the standard setup and trained and tested the models on geometries of the same type. However, the out-of-distribution behavior is interesting for practical use cases. Thus, we included a new experiment where we evaluate a SMART model, trained on SUV estate bodies, on SUV fastback bodies, which tests the out-of-distribution behavior. The following table shows the rel. L2 errors in $\times 10^{-2}$ on the SUV fastback dataset.
>
> | Training | Subsampled Surf | Subsampled Vol | Full Surf | Full Vol |
> |---|---|---|---|---|
> | Estate | 0.0338 | 17.5187 | 0.0339 | 17.5204 |
> | Fastback | 0.0184 | 6.5641 | 0.0186 | 6.5489 |
>
> The errors of SMART trained on estate and evaluated on fastback are increased compared to a SMART model that is trained on fastback. However, this is expected as the model learned the aerodynamics of estate SUVs. Thus, we hypothesize that SMART cannot generalize to completely new objects and that either finetuning on a few samples is required or that the model must be trained on diverse datasets with different objects to enable zero-shot generalization to new objects.
>
> > **(W4)** Overlap
>
> We acknowledge that SMART has an overlap with existing attention-based models. However, SMART specifically tackles simulations from the CAD geometry, uses the geometry as a shared latent representation for surface and volume predictions, and introduces novel components such as the geometry cross-attention and cross-layer update.
>
> > **(Q1)** Geometry generalization
>
> Please see our response for (W3) part 2.
>
> > **(Q2)** Surface and volume
>
> The model does not distinguish between surface and volume queries and processes both queries similarly. In the mentioned scenario with hollow windows, a surface and volume query with the same coordinates would produce the same output. In general, SMART implicitly learns the spatial boundaries.
>
> > Mesh awareness
>
> Even if the simulation mesh is provided to the model (e.g., AB-UPT), the model could be queried with coordinates that are within the geometry and may produce unphysical predictions. To avoid that, meaningful query coordinates must be provided to the SMART and baseline models.
>
> > **(Q3)** Divergence, drag and lift coefficients
>
> We computed the divergence of the predicted velocity fields on SHIFT-SUV and obtained a mean absolute divergence (averaged across field and samples) of $0.22 s^{-1}$. This demonstrates that SMART preserves the mass reasonably well. The drag and lift forces in Appendix L.1.5 show that SMART also accurately preserves drag and lift forces. Thus, SMART produces predictions that are reasonably good even without a physics-informed loss.

---

> > ### Author Rebuttal · Reviewer_gC68 · 2026-04-03
> >
> > Thanks to the reviewers for addressing most of my questions and concerns. Some comments:
> >
> > The OOD tests look promising but SUV Estate and Fasttrack have similar geometries, and it is hard to assess how well it generalizes to broader OOD regimes.

---

> > > ### Author Response · Authors · 2026-04-04
> > >
> > > Thank you for reviewing our work and for your response. We are glad to see that most of your concerns have been addressed. Your suggestions, including the explanation of the size K, the out-of-distribution experiment, and the divergence of the velocity field on SHIFT-SUV, will be included in the revised manuscript. Additionally, we will add a discussion of physical constraints to the limitations and future work sections.
> > >
> > > Given that all raised issues have been resolved, we kindly ask whether you might consider revising your score to reflect this, or inform us of any outstanding concerns or questions that need clarification.
> > >
> > > We appreciate the comment on the OOD tests and agree that broader geometric variation would be valuable. We chose the estate and fastback subsets of SHIFT‑SUV to ensure identical inlet velocities across both geometry types. We also considered using AhmedML and SHIFT‑SUV to broaden the geometric variation, but the inlet velocities differ substantially, which makes a direct comparison difficult. Nonetheless, we plan to evaluate the OOD behavior more rigorously and explore approaches to improve OOD generalization in future work.

---

### Official Review · Reviewer_VaP3 · 2026-03-13

**Soundness:** 3
**Presentation:** 4
**Significance:** 3
**Originality:** 3
**Overall Recommendation:** 4
**Confidence:** 4

**Summary:**

SMART proposes a mesh-free encoder-decoder architecture that takes a geometric point cloud and simulation parameters as inputs. The geometry encoder compresses these inputs into a shared latent space, while the physics decoder employs a cross-layer interaction mechanism to read the intermediate latent representations of the encoder at various depths, enabling the prediction of physical quantities at arbitrary query locations. Experimental results demonstrate that, while eliminating the reliance on computational meshes, SMART achieves prediction accuracy that rivals or even surpasses existing state-of-the-art mesh-dependent models.

**Compliance With Llm Reviewing Policy:**

Affirmed.

**Key Questions For Authors:**

1. What are the actual wall-clock inference time and FLOPs for SMART on an airplane model with 5 million query points?
2. How does the model automatically identify spatial boundaries relying solely on point cloud coordinates as inputs?
3. Can a trained SMART model only handle objects of the same class with parameterized deformations (for example, only predicting sedans), or can it generalize to novel objects with completely different topological structures?
4. For a 3D manifold with specific high-frequency features (such as extremely thin wing edges), how large must K be to mathematically guarantee that critical geometric information is not lost?

**Limitations:**

yes

**Strengths And Weaknesses:**

Strengths:
1. During the encoder phase, instead of using traditional randomly initialized learnable tokens, the model directly utilizes K real physical points uniformly sampled from the input point cloud as initial features, which elegantly injects physical spatial priors directly into the model.
2. The Cross-layer Geometry-Physics Update mechanism is a significant contribution. Each layer of the decoder reads not only the final output of the encoder but also interacts with the intermediate latent geometric features at the corresponding depth via cross-attention. This allows the geometric perception and the physical field prediction to evolve synchronously across multiple scales.
3. SMART strictly prohibits self-attention information exchange among query points. This design allows the model to partition spatial domains containing tens of millions of points into chunks for sequential evaluation during inference, thereby keeping GPU memory consumption constant. It successfully handles industrial simulation tasks with tens of millions of nodes, proving its strong potential for real-world industrial simulations.
4. The ablation studies are highly detailed and targeted. They systematically verify the effectiveness of the coarse geometry scaffold, the geometry cross-attention, and the cross-layer update mechanism, significantly strengthening the persuasiveness of the model design.

Weaknesses:
1. The encoder relies on uniform random sampling to construct geometric features. However, for extremely large-scale point clouds, pure random sampling might miss high-frequency geometric details. Replacing it with curvature-aware sampling or Farthest Point Sampling (FPS) could make the model more robust.
2. The SMART model maps coordinates to physical quantities directly in 3D space via cross-attention mechanisms, but it does not provide any theoretical proof akin to the Universal Approximation Theorem to mathematically guarantee the architecture's capability to fit the solutions of arbitrarily complex Navier-Stokes equations.
3. The paper demonstrates the effectiveness of SMART solely through empirical experiments (such as relative L2 error) and does not provide any Theoretical Bounds.

---

> ### Author Rebuttal · Authors · 2026-03-30
>
> Thank you for carefully reviewing our work and asking insightful questions, which helped us to improve our submission. We are grateful that you recognize our approach as a strategy with the potential for wide applicability in industrial simulation tasks. We address each of your concerns and questions individually below.
>
> > **(W1)** Uniform random sampling
>
> Uniform subsampling has the advantage of preserving the underlying structure of the point cloud. The geometry point cloud often provides a higher discretization in important regions. For example, the transition between the car body and the windshield on the SHIFT-SUV dataset has a higher discretization, which is preserved with uniform sampling and may help the model to better capture the pressure differences in this area. Furthermore, uniform sampling is cheaper on large point clouds compared to sophisticated sampling strategies such as farthest-point-sampling (FPS). Additionally, we conducted an experiment on the SHIFT-SUV dataset with FPS instead of uniform sampling. The table shows that there is no significant difference between uniform and FPS.
>
> | Sampling | Subsampled Surface Rel. L2 ($\times 10^{-2}$) | Subsampled Volume Rel. L2 ($\times 10^{-2}$) | Full Surface Rel. L2 ($\times 10^{-2}$) | Full Volume Rel. L2 ($\times 10^{-2}$) |
> |---|---|---|---|---|
> | Uniform | 0.0175 | 6.2627 | 0.0175 | 6.2946 |
> | FPS | 0.0174 | 6.2449 | 0.0175 | 6.2922 |
>
> > **(W2)** Theoretical proof
>
> Our approach is mainly based on the attention mechanism. For the attention mechanism, it has been shown that it is a special case of a neural operator layer and that the universal approximation theorem for neural operators also holds for attention-based architectures [1]. Consequently, mapping 3D coordinates to physical quantities with attention is a special application of a neural operator that satisfies the approximation theorem for neural operators. Thus, the theorems for neural operators also hold for the SMART model.
>
> > **(W3)** Theoretical bounds
>
> We refer to our answer to (W2). We included a justification in the manuscript that SMART satisfies the universal approximation theorem for neural operators.
>
>
> > **(Q1)** Inference time
>
> The wall-clock time on the airplane dataset, with 5 million query points is $17.11 \pm 0.0656$ s and 60,968 GFLOPS, measured on an A100 80GB GPU.
>
> > **(Q2)** Spatial boundaries
>
> The geometry point clouds are dense enough that spatial coordinates alone contain all the information that is needed to infer the spatial boundaries. The attention mechanism in the encoder allows the model to build a latent representation of the object and to learn/encode the surfaces and spatial boundaries of each object solely based on the coordinates.
>
> > **(Q3)** Generalize to novel objects
>
> If the SMART model is only trained on a specific class of objects (e.g., estate SUVs), the generalization to new objects is limited. We added a new experiment where we evaluate a SMART model, trained on SUV estate bodies, on SUV fastback bodies. The following table shows the results on the SUV fastback dataset.
>
> | Training Dataset | Subsampled Surface Rel. L2 ($\times 10^{-2}$) | Subsampled Volume Rel. L2 ($\times 10^{-2}$) | Full Surface Rel. L2 ($\times 10^{-2}$) | Full Volume Rel. L2 ($\times 10^{-2}$) |
> |---|---|---|---|---|
> | SUV Estate | 0.0338 | 17.5187 | 0.0339 | 17.5204 |
> | SUV Fastback | 0.0184 | 6.5641 | 0.0186 | 6.5489 |
>
> The rel. L2 errors of SMART trained on estate and evaluated on fastback are increased compared to a SMART model that is trained on fastback SUVs. However, this is expected as the model learned the aerodynamics of estate SUVs. Thus, we hypothesize that SMART cannot generalize to completely new objects and that either finetuning on a few samples is required or that the model must be trained on diverse datasets with different objects to enable zero-shot generalization to new objects.
>
> > **(Q4)** Size of K
>
> Providing a theoretical guarantee for the size of the latent geometries K is non-trivial, because K is a task‑dependent hyperparameter that potentially must be adapted to the geometric complexity, flow physics, and effective size of the simulation domain. For instance, in the SHIFT‑Wing dataset, the domain is large and the wake exhibits long, coherent trailing‑vortex structures. Capturing these persistent 3D flow features requires a larger number of latent tokens. In contrast, the SHIFT‑SUV dataset represents a bluff‑body flow with a short, rapidly breaking turbulent wake, so fewer tokens are sufficient. The ablation study in Appendix K.1 (page 30) investigates the effect of different K on the model's performance. A small K of 128 already yields errors smaller compared to most of the baselines, and increasing K consistently decreases the errors. This suggests that even a small K can capture the important geometric structures.
>
> References \
> [1] Kovachki et al. (2021). Neural operator: Learning maps between function spaces. http://arxiv.org/abs/2108.08481

---

> > ### Author Rebuttal · Reviewer_VaP3 · 2026-04-03
> >
> > The authors have addressed my concerns.

---

> > > ### Author Response · Authors · 2026-04-04
> > >
> > > Thank you for taking the time to review our work and for your response. We appreciate that your concerns have been addressed. We will integrate all of the suggested additions into the revised manuscript, including the ablation study of the sampling strategy, theoretical proof, inference time and FLOPS, out-of-distribution experiment, and clarification on how to choose the latent geometry size K.
> > >
> > > Given that all raised issues have been resolved, we kindly ask whether you might consider revising your score to reflect this, or inform us of any outstanding concerns or questions that need clarification.

---

### Official Review · Reviewer_wxMY · 2026-03-13

**Soundness:** 3
**Presentation:** 3
**Significance:** 3
**Originality:** 3
**Overall Recommendation:** 4
**Confidence:** 4

**Summary:**

This is a well-motivated paper that targets a real bottleneck in practical deployment: many strong surrogate models are only partially “mesh-free” in the sense that their predictions still depend strongly on the particular simulation mesh used during training or evaluation. The paper does a good job articulating why arbitrary independent queries matter, and the evaluation includes a useful rear-region query-shift test that directly probes this claim.

**Compliance With Llm Reviewing Policy:**

Affirmed.

**Final Justification:**

In general, the author's response addressed some of my concerns.

**Key Questions For Authors:**

1. The paper introduces a number of sensible components, but it is not yet clear which of them constitutes the real conceptual advance. A large portion of the pipeline resembles an effective synthesis of ideas already present in modern neural operator and transformer surrogate literature: arbitrary-query decoding, latent token compression, cross-attention from queries to latent states, and simulation-parameter modulation.

2. Query independence is arguably the paper’s defining claim, yet it is mostly justified by architectural description rather than by a precise statement of what is and is not invariant. In practice, the paper seems to mean that queries do not exchange information and do not alter latent geometry, which is a useful property, but this is weaker than a formal characterization of independence under query partitioning, ordering, batching, or distribution shift.

3. The geometry encoder relies on subsampled point-cloud representations and a coarse-geometry scaffold, but the paper does not thoroughly justify why uniform subsampling is the right choice versus geometry-aware alternatives such as curvature-aware or farthest-point strategies. This matters because aerodynamic quantities are often highly sensitive near localized geometric features. These are not minor implementation details; they are structurally important to the success of the approach.

**Limitations:**

yes

**Strengths And Weaknesses:**

1. The paper is unusually clear about the difference between supporting arbitrary query coordinates in principle and being truly independent of the simulation mesh in practice. That distinction is important and often glossed over in surrogate-modeling papers.
2. The decoder architecture is well matched to the query-independence claim: queries do not self-interact and instead pull information only from latent geometry through cross-attention.
3. The paper reports strong results across multiple aerodynamic datasets and includes nontrivial stress tests such as full-resolution evaluation and shifted query distributions.

---

> ### Author Rebuttal · Authors · 2026-03-30
>
> Thank you for your constructive feedback and thoughtful questions, which helped us to enhance our submission. We appreciate your recognition of SMART's role in overcoming the simulation-mesh bottleneck. We respond to each of your points individually below.
>
> > **(Q1)** The paper introduces a number of sensible components, but it is not yet clear which of them constitutes the real conceptual advance.
>
> SMART offers query independence through a decoder that uses cross-attention between the latent geometry and the query points. Moreover, SMART can accurately predict physical fields directly from a cheap geometry point cloud, enabled by a strong encoder and a tightly coupled encoder-decoder design. The ablation study in Appendix K (page 29) shows that encoding the input geometry point cloud into a coarse geometry, rather than into learned latent tokens, leads to improved performance. In addition, the cross-layer geometry-physics update provides a substantial further gain.
>
> > A large portion of the pipeline resembles an effective synthesis of ideas already present in modern neural operator and transformer surrogate literature [...]
>
> While there is an overlap with existing attention-based models, previous models, such as LNO [1], often rely on encoding the input into a set of learned latent tokens and only utilize the final output representation of the encoder. In contrast, SMART introduces novel components such as using the geometry as a scaffold for the latent space, geometry cross-attention, and the cross-layer geometry-physics update.
>
>
> > **(Q2)** Query independence is arguably the paper’s defining claim, yet it is mostly justified by architectural description rather than by a precise statement of what is and is not invariant. [...]
>
> Yes, query independence means that queries do not interact and do not alter the latent geometry. To investigate query independence, only the decoder's components need to be analyzed. Using only pointwise operations (e.g., pointwise positional encodings and MLPs) on the queries $P$ and cross-attention between the queries and latent geometries automatically ensures query independence. This implies that queries can be arbitrarily partitioned, ordered, or batched, since each query in $P$ is independent of the others. Formally, it holds $\forall p \in \Omega \quad \forall P =${$p_1, ..., p_{j-1}, p, p_{j+1}, ...$}$ \subseteq \Omega: \quad f_\theta(p, G, \boldsymbol{\xi}) = f_\theta(P, G, \boldsymbol{\xi})_j$ which means that querying the model with only one arbitrary query point $p$ yields the same output as querying $p$ together with an arbitrary query set $P$. We will clarify this in the final version of the paper.
>
> > **(Q3)** The geometry encoder relies on subsampled point-cloud representations and a coarse-geometry scaffold, but the paper does not thoroughly justify why uniform subsampling is the right choice versus geometry-aware alternatives such as curvature-aware or farthest-point strategies. [...]
>
> Uniform subsampling has the advantage of preserving the underlying structure of the point cloud. The geometry point cloud often provides a higher discretization in important regions. For example, the transition between the car body and the windshield on the SHIFT-SUV dataset has a higher discretization, which is preserved with uniform sampling and may help the model to better capture the pressure differences in this area. Note that the higher discretization is a property of CAD meshes and that the mesh is still not optimized for simulations. Furthermore, uniform sampling is cheaper on large point clouds compared to sophisticated sampling strategies such as farthest-point-sampling (FPS). We added this to the manuscript to provide clarification. Additionally, we conducted an experiment on the SHIFT-SUV dataset with FPS instead of uniform sampling. The table shows that there is no significant difference between uniform and farthest-point sampling.
>
> | Sampling | Subsampled Surface Rel. L2 ($\times 10^{-2}$) | Subsampled Volume Rel. L2 ($\times 10^{-2}$) | Full Surface Rel. L2 ($\times 10^{-2}$) | Full Volume Rel. L2 ($\times 10^{-2}$) |
> |---|---|---|---|---|
> | Uniform | 0.0175 | 6.2627 | 0.0175 | 6.2946 |
> | FPS | 0.0174 | 6.2449 | 0.0175 | 6.2922 |
>
> References \
> [1] Wang, T., & Wang, C. (2024). Latent Neural Operator for solving forward and inverse PDE problems. In arXiv [cs.LG]. http://arxiv.org/abs/2406.03923

---

> > ### Author Rebuttal · Reviewer_wxMY · 2026-04-02
> >
> > Thanks for the detailed reply from the author. The author's reply addressed my concerns.

---

> > > ### Author Response · Authors · 2026-04-02
> > >
> > > We appreciate your thorough review and your response. We are pleased to see that we have addressed your concerns. All suggested improvements, including the clarification of query independence as well as the additional ablation study and explanation of the sampling strategy, will be incorporated into the revised manuscript.
> > >
> > > Given that all raised issues have been resolved, we kindly ask whether you might consider revising your score to reflect this, or inform us of any outstanding concerns or questions that need clarification.

---

### Decision · Program_Chairs · 2026-04-30

**Decision:**

Accept (regular)

**Comment:**

The paper introduces a transformer-based neural surrogate model for 3D aerodynamic simulations that operates directly on raw geometry point clouds, without relying on predefined simulation meshes. The motivation is to remove the costly mesh-generation step required by existing approaches. The model encodes geometry and simulation parameters into a latent representation and uses a query-independent decoder with cross-attention to predict physical quantities at arbitrary spatial locations. This design enables scalable inference on large domains. Experiments on multiple aerodynamic datasets show that the method achieves accuracy competitive with mesh-based approaches, while improving flexibility for large scale applications.

The reviewers agree that the paper addresses an important and practical problem, and that the proposed approach is empirically strong. The main concerns relate to the clarity of the core conceptual contribution given its overlap with existing architectures, limitations of the uniform sampling strategy used by the authors, and questions regarding physical consistency.

In response, the authors provided detailed clarifications of the methodology, added ablations including alternative sampling strategies, and included additional experiments on scalability and physical metrics. Overall, the rebuttal satisfactorily addressed most of the reviewers’ concerns, and there is a consensus in favor of acceptance.